# Riboflavin metabolism shapes FSP1-driven ferroptosis resistance

Vera Skafar[1], Izadora de Souza[1], Biplab Ghosh[2,3], Ancely Ferreira dos Santos [1], Florencio Porto Freitas [1], Zhiyi Chen[1], Shibo Sun [1], Merce Donate Castillo [1], Palina Nepachalovich[4], Lars Seufert [5], Sebastian Bothe[1], Juliane Tschuck [6], Apoorva Mathur [7], Ariane Nunes-Alves [7], Jannik Buhr [8], Camilo Aponte-Santamaría[8], Werner Schmitz[9], Matthias Mack [5], Martin Eilers [9], Ralf Bargou[10], Milena Chaufan [11], Mayher Kaur[11], Mario Palma [11], Jessalyn M. Ubellacker [11], Ulrich Elling [12], Hellmut G. Augustin[2,3], Kamyar Hadian [6], Svenja Meierjohann[13], Bettina Proneth [14], Marcus Conrad [14,15], Maria Fedorova [4], Hamed Alborzinia[16,17] & José Pedro Friedmann Angeli [1,10] ✉

Membrane protection against oxidative insults is achieved by the concerted action of glutathione peroxidase 4 (GPX4) and endogenous lipophilic antioxidants such as ubiquinone and vitamin E. More recently, ferroptosis suppressor protein 1 (FSP1) was identified as a critical ferroptosis inhibitor, acting via the regeneration of membrane-embedded antioxidants. Yet, regulators of FSP1 are largely uncharacterized, and their identification is essential for understanding the mechanisms buffering phospholipid peroxidation and ferroptosis. Here we report a focused CRISPR–Cas9 screen to uncover factors influencing FSP1 function, identifying riboflavin (vitamin B$_2$) as a modulator of ferroptosis sensitivity. We demonstrate that riboflavin supports FSP1 stability and the recycling of lipid-soluble antioxidants, thereby mitigating phospholipid peroxidation. Furthermore, we show that the riboflavin antimetabolite roseoflavin markedly impairs FSP1 function and sensitizes cancer cells to ferroptosis. Our findings provide a rational strategy to modulate the FSP1–antioxidant recycling pathway and underscore the therapeutic potential of targeting riboflavin metabolism, with implications for understanding the interaction of nutrients, as well as their contributions to a cell's antioxidant capacity.

Cellular membranes are crucial structural components that act as dynamic barriers. Membrane lipid constituents are nevertheless vulnerable to oxidative damage, in a process known as lipid peroxidation (LPO)[1]. LPO disrupts membrane integrity and executes ferroptosis, a type of regulated cell death implicated in a growing number of (patho)physiological processes, including cancer, neurodegeneration and ischaemia-reperfusion injury[2].

Ferroptosis is predominantly inhibited by glutathione peroxidase 4 (GPX4)[3,4], which reduces phospholipid hydroperoxides using glutathione. Additionally, radical-trapping antioxidants such as ubiquinone (CoQ10), vitamin E and vitamin K are crucial in suppressing the propagation of lipid peroxidation[5]. More recently, ferroptosis suppressor protein 1 (FSP1) has emerged as a key player in resisting ferroptosis[6,7]. Unlike GPX4, FSP1 regenerates quinone-like antioxidants such as ubiquinone and vitamin K (ref. [8]), using NAD(P)H as an electron donor, thereby halting the radical chain reaction required for ferroptosis execution. Although FSP1's enzymatic mechanism is well understood, actionable factors regulating its function are

largely uncharacterized. Identifying these regulators is vital for comprehending how cells maintain membrane redox homeostasis and withstand stress-induced ferroptosis, potentially revealing therapeutic strategies for pathological conditions in which ferroptosis plays an important role.

Here we use a CRISPR–Cas9 screen to identify regulators of FSP1 function, revealing riboflavin (vitamin B$_2$) to be a modulator of ferroptosis sensitivity. Riboflavin, best known as a precursor for flavin adenine dinucleotide (FAD) and flavin mononucleotide (FMN), is essential for redox biology. We show that riboflavin directly supports the stability and activity of FSP1, enhancing its capacity to recycle lipid-soluble antioxidants and to mitigate phospholipid peroxidation. Unlike other vitamins that act as direct radical-trapping antioxidants, riboflavin uniquely facilitates enzymatic recycling and is positioned upstream in the cascade, promoting ferroptosis resistance and preserving membrane integrity.

By revealing the role of riboflavin in regulating FSP1 and lipophilic antioxidant recycling, our findings expand our understanding of ferroptosis regulation and provide a framework for therapeutic strategies targeting the FSP1–antioxidant axis in cancer and other ferroptosis-associated diseases.

## Results

### Focused CRISPR-based screen uncovers regulators of FSP1

We previously demonstrated that cells lacking GPX4 can survive and proliferate indefinitely when FSP1 activity is robust, either through naturally elevated FSP1 expression or enforced overexpression[6] (Fig. 1a). Building on this observation, we generated FSP1-dependent HT1080 cells where GPX4 was deleted in an FSP1-overexpressing background (HT1080$^{GPX4KO/FSP1OE}$; Fig. 1b). HT1080$^{GPX4KO/FSP1OE}$ cells readily undergo cell death upon FSP1 inhibition, which can be rescued by co-treatment with the ferroptosis inhibitor liproxstatin-1 (Lip-1; Fig. 1a,c).

We reasoned that this cellular model could be combined with CRISPR-based genetic screening to identify factors contributing to FSP1 function. We thus performed a focused CRISPR screen (targeting ~3,000 potentially druggable genes; a complete list of genes is provided in Supplementary Table 1), comparing conditions with and without Lip-1. We posited that genetic perturbations impairing FSP1 expression or activity would induce ferroptosis, which Lip-1 should rescue (Fig. 1d)[6]. Through this approach, we identified several genes whose loss significantly impacted FSP1-dependent ferroptosis resistance, with the two top hits being *SCD1* and *RFK*, encoding for stearoyl-CoA desaturase-1 (SCD1) and riboflavin kinase (RFK), respectively (Fig. 1e and Supplementary Table 2). Loss of SCD1 is known to increase ferroptosis sensitivity by raising the polyunsaturated fatty acid-to-monounsaturated fatty acid (PUFA/MUFA) ratio[9]. Consistent with this, pharmacological inhibition of SCD1 readily triggered ferroptosis in HT1080$^{GPX4KO/FSP1OE}$ cells (Extended Data Fig. 1a,b), probably due to an elevated pool of oxidizable substrates overwhelming the protective capacity of FSP1. These findings suggest that cells with a high PUFA/MUFA ratio show limited benefit conferred by FSP1 activity.

Given these results, we focused on RFK and its role in regulating FSP1. RFK is a key enzyme that phosphorylates riboflavin to generate flavin mononucleotide (FMN), a central step in the production of flavin adenine dinucleotide (FAD)[10]—a cofactor essential for the activity of flavoproteins such as FSP1[11]. The dependence of FSP1 on FAD implies a direct link to RFK, but the extent to which RFK deficiency affects FSP1 levels and ferroptosis resistance is unknown. Using A375 cells, in which FSP1 confers strong protection against GPX4 inhibitors (GPX4i) including RSL3 and ML210, we found that CRISPR-mediated deletion of *RFK* leads to a robust loss of RFK expression and a decrease in FSP1 protein levels (Fig. 1f). This reduction renders RFK-deficient cells highly sensitive to GPX4i, highlighting a

previously uncharacterized dependency of ferroptosis resistance on RFK (Fig. 1g and Extended Data Fig. 1c). We obtained similar results using HT1080$^{GPX4KO/FSP1OE}$ cells, where we observed that the absence of RFK impairs viability and induces ferroptosis (Extended Data Fig. 1d–g). Our findings establish RFK as an actionable upstream regulator of FSP1 functionality and ferroptosis resistance.

### FAD deficiency disrupts FSP1 function and promotes ferroptosis susceptibility

Interestingly, we observed that increased sensitivity to GPX4 inhibitors (GPX4i) in cells transduced with single guide RNA (sgRNA) targeting RFK was lost over time. FAD biosynthesis is a multistep process beginning with riboflavin uptake via members of the SLC52A family of solute carriers (SLC52A1, SLC52A2 and SLC52A3). Once inside the cell, riboflavin is phosphorylated by RFK to generate FMN, which is subsequently adenylated by flavin adenine dinucleotide synthase (FADS, encoded by *FLAD1*) to produce FAD (Fig. 2a). Consistently, analysis of the DepMap database (www.depmap.org) revealed that RFK loss is poorly tolerated by most cell types (Fig. 2b). This intolerance was evident in our cultures, where unedited cells rapidly outcompeted RFK-deficient cells (Extended Data Fig. 2a). These challenges impeded further experiments and prompted us to explore whether other enzymes generating the flavin cofactors could be targeted.

Notably, unlike RFK, loss of FADS—the enzyme responsible for the final step in FAD biosynthesis—appears to be better tolerated by cells, probably because essential FMN-dependent proteins remain functional. We therefore disrupted FADS in A375 cells and HT1080$^{GPX4KO/FSP1OE}$ cells and generated single clones thereof (Fig. 2c and Extended Data Fig. 2b–j), which allowed us to establish an isogenic pair of FADS-proficient and -deficient cell lines (Fig. 2c,d). Using this model, we found that loss of FADS leads to markedly increased sensitivity to GPX4 inhibition (Fig. 2e), as well as to lipid peroxidation (Fig. 2f). We observed similar effects in the HT1080$^{GPX4KO/FSP1OE}$ model (Extended Data Fig. 2h–j).

Loss of flavin cofactors impairs the stability of the flavoproteome and exerts a broad metabolic effect, including impacting lipid metabolism[12]. We thus aimed to determine whether the effect—loss of redox homeostasis—is FSP1-specific or is more generalizable. A whole-proteome analysis of the FADS isogenic pair confirmed a reduced abundance of flavoproteins. Yet, interestingly, FSP1 and NQO1 are the most strongly depleted flavoproteins in FADS-deficient cells (Fig. 2g and Supplementary Table 3), an effect independent of messenger RNA (mRNA) abundance (Extended Data Fig. 2k). Building on this observation, we validated the effect in other cell lines, showing that genetic loss of FADS leads to a substantial loss of FSP1 and is accompanied by increased sensitivity to ferroptosis (Extended Data Fig. 2l,m).

Molecular dynamic simulations of FSP1 with and without FAD were carried out, and root-mean-square deviation (RMSD) analysis revealed that the absence of FAD increases protein backbone instability (Extended Data Fig. 2n). Root-mean-square fluctuation (RMSF) profiles revealed more significant residue fluctuations, especially in residues 282–300, which interact with FAD (Extended Data Fig. 2o), following our previous report[13]. Overall, these findings establish that FAD is essential not only for activity but also for FSP1 stability.

Using genetic and pharmacological approaches, we found that loss or inhibition of FSP1 in FADS-deficient cells causes no further sensitization to lipid peroxidation and ferroptosis induced by GPX4i (Fig. 2h–k and Extended Data Fig. 2p,q). Finally, loss of FADS appeared to sensitize cells to GPX4i specifically (Fig. 2h–k): other ferroptosis inducers, such as erastin and L-buthionine sulfoximine (BSO), or other tested cytotoxic agents encompassing a diverse array of mechanisms of action (Extended Data Fig. 2p,q), exert no enhanced effects in the absence of FADS. These findings establish FADS as a key regulator of FSP1 activity and ferroptosis sensitivity.

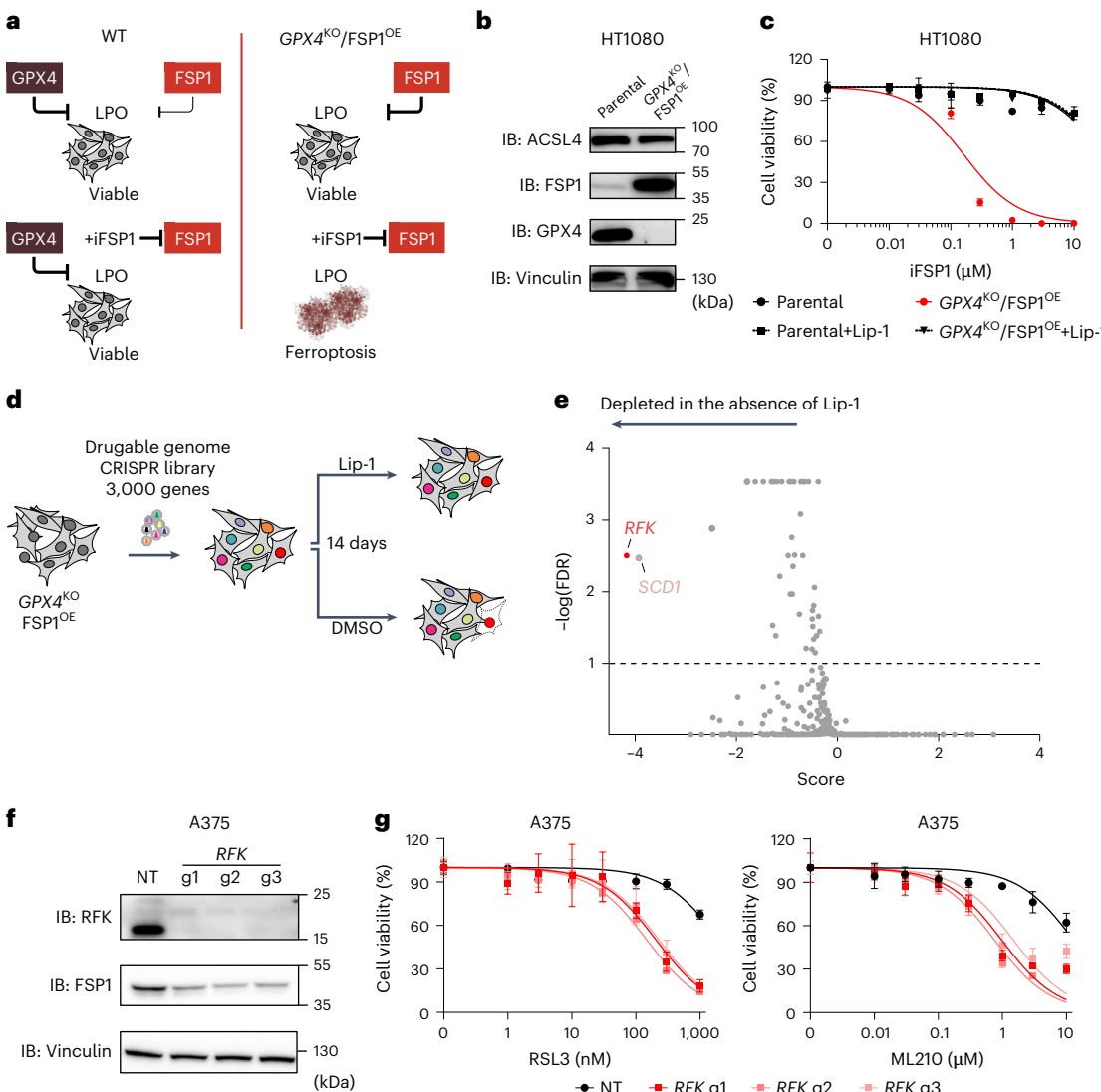

**Fig. 1 | Identification of factors supporting FSP1 function. a**, Schematic representation of the FSP1-dependent model used to identify factors supporting FSP1 function. Upper left: the primary protective system against lipid peroxidation (LPO) is the enzyme GPX4, complemented by FSP1. Lower left: cells treated with an FSP1 inhibitor (iFSP1) can rely on GPX4 activity to survive. Upper right: FSP1-dependent HT1080 cells (HT1080$^{GPX4KO/FSP1OE}$). In HT1080 cells, as in many others, knocking out *GPX4* induces ferroptosis due to insufficient endogenous FSP1 levels to compensate for GPX4 loss. However, cell survival can be rescued by overexpressing FSP1. Lower right: upon pharmacological inhibition of FSP1 (iFSP1 treatment), cells undergo ferroptosis, as they solely rely on FSP1 function for survival. **b**, Immunoblot (IB) analysis of ACSL4, FSP1, GPX4 and vinculin in HT1080 parental and HT1080$^{GPX4KO/FSP1OE}$ cells. The experiment was performed once. **c**, Dose-dependent toxicity of an FSP1 inhibitor (iFSP1) in HT1080 parental and HT1080$^{GPX4KO/FSP1OE}$ cell lines. Cell viability was monitored using alamarBlue after 24 h of treatment. Where indicated, cells were treated with the ferroptosis inhibitor Lip-1 (500 nM). Data plotted are mean ± s.d. of triplicates from one representative of three independent experiments.

**d**, Schematic representation of the screening strategy used to identify factors supporting FSP1 function in the previously described FSP1-dependent cellular model (**a**). HT1080$^{GPX4KO/FSP1OE}$ cells were transduced with a guide RNA (gRNA) library targeting ~3,000 genes and selected over seven days in the presence of the ferroptosis inhibitor Lip-1 (500 nM). Subsequently, the cells proliferated with or without Lip-1 supplementation for an additional 14 days. **e**, Graphical representation of the results from two independently performed CRISPR screens. The plot depicts the score calculated using the MaGeCK package (*x* axis) and the −log(false discovery rate, FDR) (*y* axis). *RFK* and *SCD1* genes were identified as potentially robust candidates to modulate FSP1 function. **f**, IB analysis of RFK, FSP1 and vinculin in A375 cells transduced with either a non-targeting control (NT) or three different *RFK*-targeting sgRNAs. The data shown are from one representative of two independent experiments. **g**, Dose-dependent toxicity of RSL3 and ML210 in A375 cells transduced with either NT or three different *RFK*-targeting sgRNAs. Cell viability was monitored after 72 h of treatment. Data plotted are mean ± s.d. of triplicates from one representative of three independent experiments.

## Riboflavin availability as a central determinant of ferroptosis resistance

Having established a functional link between intracellular flavin metabolism and ferroptosis resistance via FSP1, we hypothesized that the availability of riboflavin—the main precursor for FAD—can influence ferroptosis sensitivity. To test this, we examined proteomic changes in cells cultured under riboflavin-deficient conditions (Fig. 3a, Extended Data Fig. 3a–c and Supplementary Table 3). As in

FADS-deficient cells, we observed an overall loss of flavoproteins. However, the changes appeared more extensive, presumably because FMN-containing proteins are also affected (Extended Data Fig. 3d,e). Notably, FSP1 emerged as one of the most downregulated proteins after 96 h of riboflavin deprivation (Fig. 3a,b), underscoring the critical role of FAD in FSP1 stability. Riboflavin withdrawal was also accompanied by upregulation of NRF2 target genes (for example, *AKR1Cs*, *TXNRD1*, *GCLM*).

Culturing cells without riboflavin for 72 h markedly increased their susceptibility to lipid peroxidation (Fig. 3c and Extended Data Fig. 3f) and enhanced cell death upon GPX4 inhibition (GPX4i) (Fig. 3d and Extended Data Fig. 3g). These effects are not limited to A375 cells: three additional cancer cell lines from different tissue origins exhibited similar responses (Fig. 3b,d). Consistent with an FSP1-dependent mechanism, combining GPX4i with an FSP1 inhibitor (iFSP1) strongly sensitized cells under riboflavin-replete conditions, but had negligible effects in the absence of riboflavin (Fig. 3d). We further corroborated these effects in FSP1-deficient A375 and MDA-MB-231 cell lines. Upon GPX4 inhibition, only cells cultured with riboflavin displayed further ferroptosis sensitization following FSP1 loss (Fig. 3e,f). These results were recapitulated in HT1080$^{GPX4KO/FSP1OE}$ and A375 cells, where genetic deletion of the main riboflavin transporter, namely SLC52A2, disrupted FSP1 function and sensitized cells to GPX4i (Extended Data Fig. 4a–i). These findings indicate that, under riboflavin-poor conditions, cells become ferroptosis-sensitive primarily through an FSP1-dependent pathway.

To directly demonstrate that riboflavin supports protection against LPO, we analysed the epilipidome of cells treated with GPX4i under riboflavin-replete and riboflavin-deficient conditions (Fig. 3g). Our detailed analysis revealed a rapid and specific accumulation of oxidized phosphatidylethanolamine (PE) species[14] upon GPX4 inhibition in riboflavin-starved cells. Critically, treatment with Lip-1 fully reversed the formation of oxidized lipids. Together, these results establish a direct link between riboflavin and membrane antioxidant capacity.

Given our results under riboflavin-depleted conditions, we next asked whether moderate variations in riboflavin concentration would similarly affect sensitivity to GPX4i. Notably, human plasma riboflavin concentrations typically range from 10 to 20 nM (ref. 15), whereas standard cell culture media such as RPMI and Dulbecco's modified Eagle medium (DMEM) contain around 500 and 1,000 nM, respectively. More advanced media, such as Plasmax[16] and human plasma-like medium (HPLM)[17], also include substantially higher riboflavin levels, at 300 and 500 nM, respectively. We found that physiological riboflavin levels (≤20 nM) markedly reduced FSP1 expression, whereas concentrations above 100 nM were sufficient to stabilize FSP1 and confer ferroptosis resistance (Extended Data Fig. 4j). Altogether, our studies demonstrate that riboflavin availability is a central determinant of membrane repair capacity and determines FSP1 antioxidant recycling capacity.

## Riboflavin analogues disrupt FSP1 activity and promote ferroptosis

Our results suggest that pharmacologically targeting riboflavin metabolism, to disrupt its FSP1-protective branch, would sensitize cancer cells to ferroptosis. No selective inhibitors for any of the proteins involved in riboflavin uptake (SLC52A2) or its action towards FMN (RFK) and FAD (FADS) have been described. Nonetheless, bacteria from the genus *Streptomyces* (for example, *S. davaonensis* and *S. cinnabarinus*) produce the riboflavin antimetabolite roseoflavin, which exerts antimicrobial activity by binding to and disrupting riboflavin riboswitches. Roseoflavin has shown promise as a broad-spectrum antibiotic, but few studies have explored it as an anticancer agent. In eukaryotic cells, roseoflavin is thought to follow a similar metabolic route as riboflavin, being transported, phosphorylated and adenylated (Fig. 4a)[18]. The dimethylamino group on C8 of the isoalloxazine ring leads to the formation of an altered flavin cofactor that is believed to disrupt normal flavin-mediated electron-transfer reactions.

Using our HT1080$^{GPX4KO/FSP1OE}$ cells, we investigated whether roseoflavin influences FSP1-mediated ferroptosis protection. Notably, at physiologically relevant levels of riboflavin, roseoflavin induced ferroptosis within the single-digit nanomolar range (Fig. 4b). Moreover, roseoflavin affected the response to GPX4i only in FSP1-expressing cells (Fig. 4c and Extended Data Fig. 5a,b), and no sensitization was detected in FSP1-deficient cells (Fig. 4c and Extended Data Fig. 5c), indicating that the effect of roseoflavin is specific. Additionally, roseoflavin restored FSP1 levels when cells were treated under low and physiological riboflavin conditions (Fig. 4d and Extended Data Fig. 5c), suggesting that the effect is probably on-target. The effect of roseoflavin was broadly reproduced in a larger panel of cell lines (Fig. 4e), where treatment can restore FSP1 levels (Fig. 4f). These actions reflect the ability of roseoflavin adenine dinucleotide (roFAD), like FAD, to stabilize FSP1; however, the modification of the isoalloxazine group destroys its oxidoreductase function.

Finally, we explored the mechanism of action of roseoflavin. To this end, we first generated A375 cells stably expressing either mock (control) or FSP1-Flag and performed immunoprecipitation (IP) assays to isolate and purify the Flag-tagged protein (Fig. 5a). Upon successful purification, we measured the enzymatic activity in a cell-free system using NADH as the electron donor and monitored FSP1 activity by following resazurin reduction through fluorescence (Fig. 5b). Importantly, there was no activity in the mock condition, and the signal was abolished by iFSP1, thus confirming the specificity of the assay.

**Fig. 2 | FAD deficiency disrupts FSP1 function and promotes ferroptosis susceptibility. a**, Schematic representation of FMN and FAD biosynthesis from riboflavin (RbF). Riboflavin is first phosphorylated to form FMN in a reaction catalysed by RFK (encoded by *RFK*). FMN is then adenylated to form FAD in a reaction catalysed by FADS (encoded by *FLAD1*). **b**, CRISPR dependency scores of *RFK*, *FLAD1* and *SLC52A1-3* perturbations across a panel of human cancer cell lines (https://depmap.org/portal/, version 23Q2). Data are represented as a violin plot. The median, as well as the upper and lower quartiles, are indicated by dashed lines. **c**, IB analysis of FLAD1, RFK and vinculin in A375 parental, A375 *FLAD1*$^{KO}$ single clone 1 (C1) and A375 *FLAD1*$^{KO}$ C1 cells stably overexpressing either an empty vector (mock) or Flag-FADS (addback). The experiment was performed once. **d**, Relative quantification of FAD in A375 parental and A375 *FLAD1*$^{KO}$ C1 cells stably overexpressing either an empty vector (mock) or Flag-FADS. Data plotted are mean ± s.d. of triplicates from one representative of two independent experiments. **e**, Dose-dependent toxicity of RSL3 and ML210 in A375 parental, A375 *FLAD1*$^{KO}$ C1 and A375 *FLAD1*$^{KO}$ C1 cells stably overexpressing either an empty vector (mock) or Flag-FADS. Cell viability was monitored using alamarBlue after 72 h of treatment. Data plotted are mean ± s.d. of triplicates from one representative of two independent experiments. **f**, Lipid peroxidation evaluated by C11-BODIPY 581/591 staining of A375 parental, A375 *FLAD1*$^{KO}$ C1 and A375 *FLAD1*$^{KO}$ C1 cells stably overexpressing either an empty vector (mock) or Flag-FADS. Cells were treated with DMSO, RSL3 (200 nM) or RSL3 (200 nM) + Lip-1 (500 nM) for 6 h. Representative plots of one of two independent experiments.

**g**, Volcano plot of DEPs between A375 *FLAD1*$^{KO}$ C1 and A375 *FLAD1*$^{KO}$ C1 cells stably overexpressing Flag-FADS. Quantified proteins are plotted based on their fold change (FC: *FLAD1*$^{KO}$ C1/*FLAD1*$^{KO}$ C1 + Flag-FADS$^{OE}$). Statistical significance was assessed by an unpaired two-sided *t*-test, and values were deemed significant if *P* < 0.05, and minimum fold change was 1.5. The statistical significance of the respective ratios is plotted on the *y* axis (*n* = 5 technical replicates). **h**, Dose-dependent toxicity of RSL3 in A375 parental and A375 *FLAD1*$^{KO}$ cells in the absence or presence of an FSP1 inhibitor (iFSP1, 2 μM). Cell viability was monitored after 72 h of treatment. Data plotted are mean ± s.d. of triplicates from one representative of three independent experiments. **i**, Dose-dependent toxicity of RSL3 and ML210 in A375 parental, A375 *FLAD1*$^{KO}$ C1 and A375 *FLAD1*$^{KO}$ C1 transduced with either NT or *FSP1*-targeting sgRNAs (*FSP1*$^{KO}$). Cell viability was monitored after 72 h of treatment. Data plotted are mean ± s.d. of triplicates from one representative of two independent experiments. **j**, IB analysis of ACSL4, FSP1, GPX4 and vinculin in A375 parental, A375 *FLAD1*$^{KO}$ C1, A375 *FLAD1*$^{KO}$ C1 cells stably overexpressing an empty vector (mock) or Flag-FADS and A375 *FLAD1*$^{KO}$ C1 transduced with either NT or *FSP1*-targeting sgRNAs (*FSP1*$^{KO}$). The experiment was performed once. **k**, Lipid peroxidation evaluated by C11-BODIPY 581/591 staining of A375 *FLAD1*$^{KO}$ C1 cells transduced with either NT or *FSP1*-targeting sgRNAs (*FSP1*$^{KO}$). Cells were treated with DMSO, RSL3 (200 nM) or RSL3 (200 nM) + Lip-1 (500 nM) for 6 h. Representative plots of one of two independent experiments. Panel **a** created with BioRender.com.

To assess whether roseoflavin can be metabolized, incorporated into FSP1, and its activity modulated, we combined modelling and functional assays. Molecular dynamics (MD) simulations of FSP1 and the cofactors FAD, 6-OH-FAD and roFAD (Extended Data Fig. 5d) suggested that roFAD binds to FSP1, although the resulting complex appears to be less stable. We next examined the functional outcome by starving cells of riboflavin for 72 h, then refeeding the cells with riboflavin or roseoflavin. As expected, flavin starvation destabilized FSP1, whereas

supplementation with either compound stabilized the protein, confirming their conversion to FAD or roFAD and proper incorporation into the enzyme. Strikingly, IP assays showed that, although roseoflavin supplementation stabilized FSP1, the enzyme was catalytically inactive (Fig. 5c,d), in contrast to the activity restored by riboflavin.

We also extended our proteomics analysis and confirmed that the expression of FSP1 and other flavoproteins is restored upon refeeding cells with either riboflavin (Fig. 5e and Supplementary Table 3)

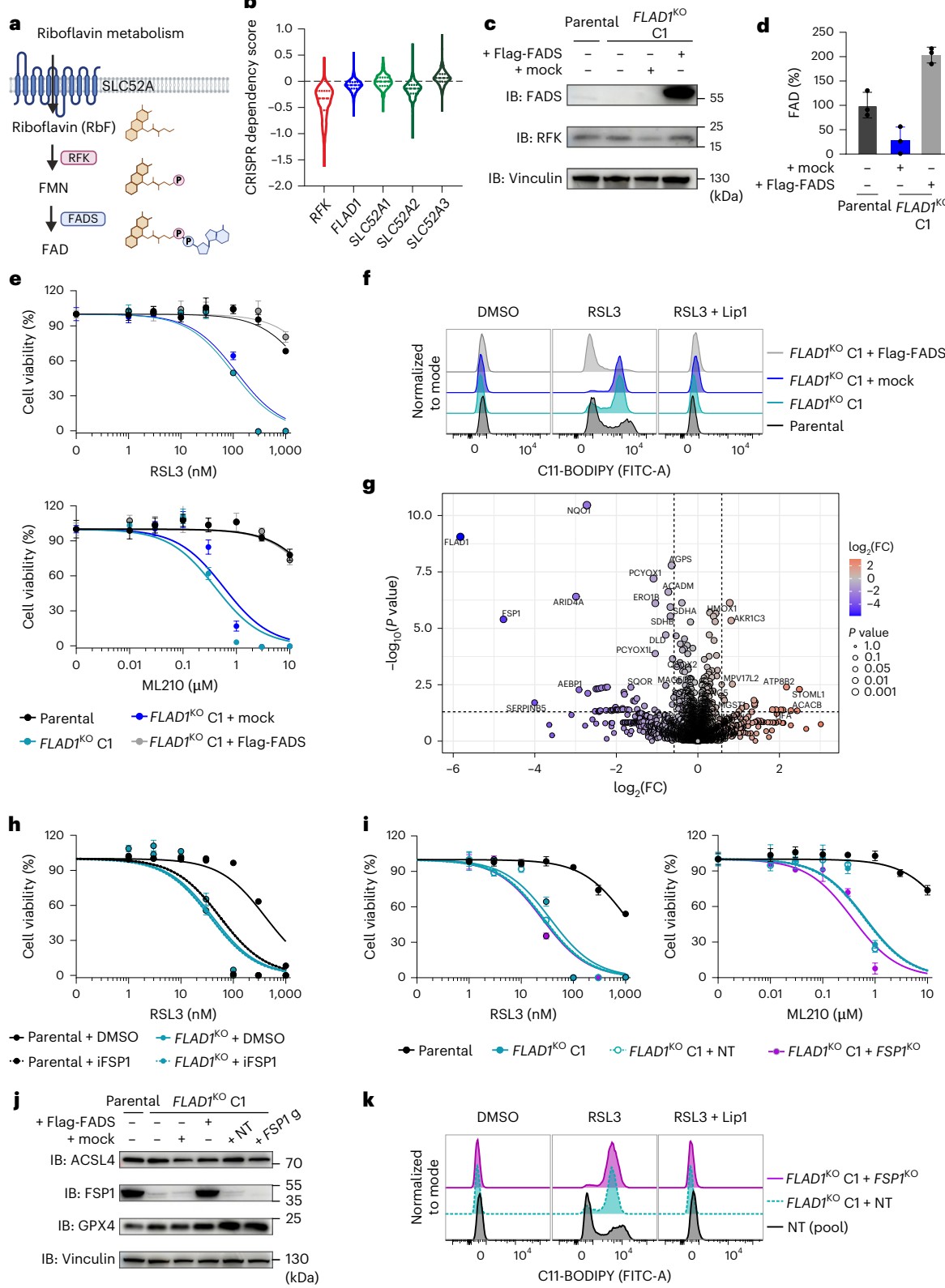

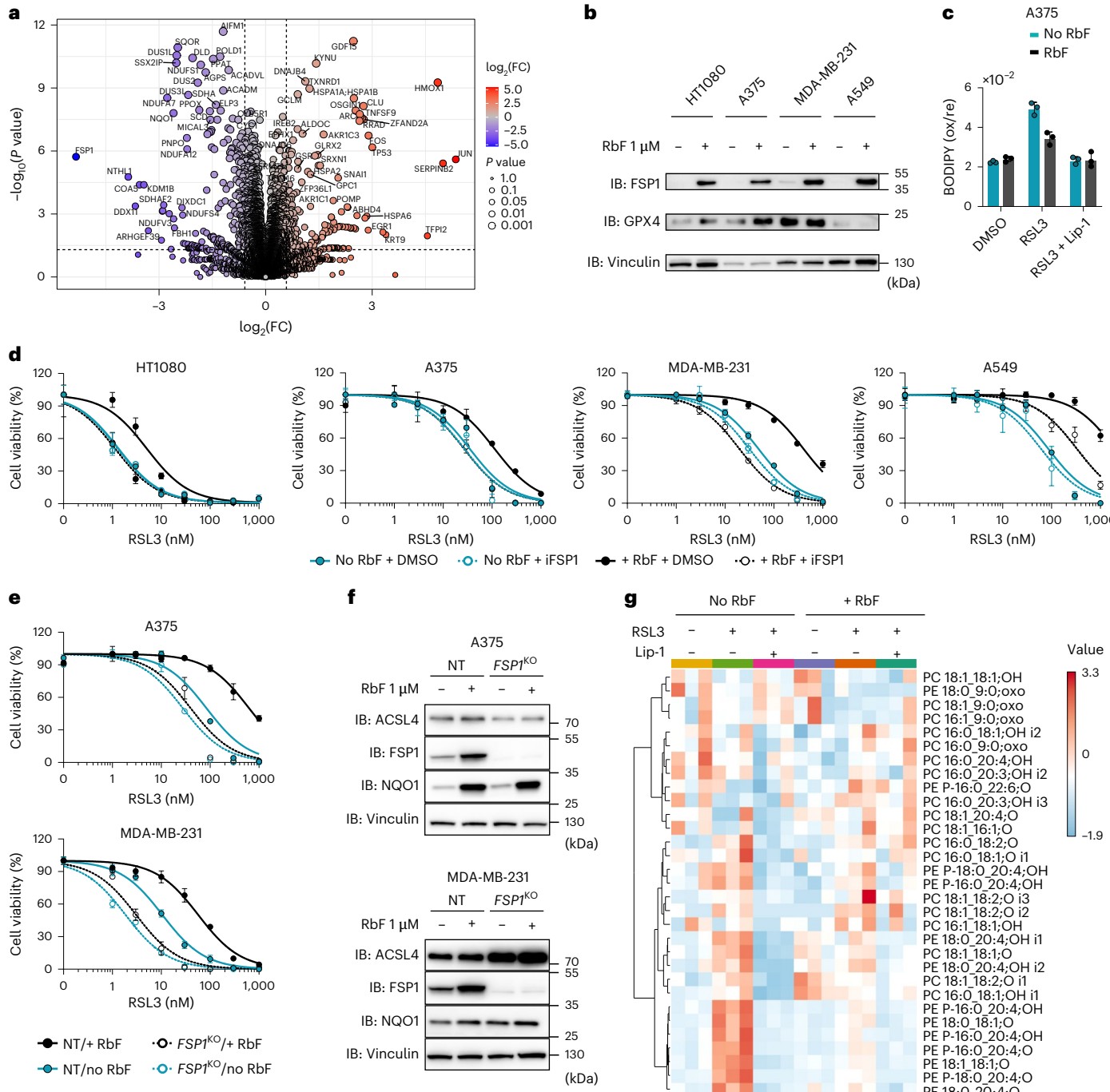

**Fig. 3 | Riboflavin availability as a central determinant of ferroptosis resistance.** **a**, Volcano plot of quantified proteins showing their change in A375 parental cells cultured in riboflavin-deficient medium for 96 h. Proteins are plotted based on their FC (riboflavin deficient/normal). Statistical significance was assessed by an unpaired two-sided *t*-test, and values were deemed significant if $P < 0.05$, and the minimum fold change was 1.5. The statistical significance of the respective ratios is plotted on the *y* axis ($n = 5$ technical replicates). **b**, IB analysis of FSP1, GPX4 and vinculin in HT1080, A375, MDA-MB-231 and A549 parental cell lines after 96 h of growth in riboflavin-deficient medium or medium supplemented with 1 μM riboflavin. Data shown are from one representative of two independent experiments. **c**, Lipid peroxidation evaluated by C11-BODIPY 581/591 staining of an A375 parental cell line cultured for 72 h in riboflavin-deficient medium or medium supplemented with 1 μM riboflavin and after treatment with DMSO, RSL3 (200 nM) or RSL3 (200 nM) + Lip-1 (500 nM) for 6 h. Data plotted are mean ± s.d. of triplicates from one representative of two independent experiments. **d**, Dose-dependent toxicity of RSL3 in the absence or presence of an FSP1 inhibitor (iFSP1, 3 μM) in HT1080, A375, MDA-MB-231 and A549 parental cell lines pre-cultured

in riboflavin-deficient medium or medium supplemented with 1 μM riboflavin for 48 h. Cell viability was monitored using alamarBlue after 96 h of treatment. Data plotted are mean ± s.d. of triplicates from one representative of three independent experiments. **e**, Dose-dependent toxicity of RSL3 in the absence or presence of an FSP1 inhibitor (iFSP1, 3 μM) in A375 and MDA-MB-231 cells transduced with either NT or *FSP1*-targeting sgRNAs pre-cultured in riboflavin-deficient medium or medium supplemented with 1 μM riboflavin for 48 h. Cell viability was monitored after 96 h of treatment. Data plotted are mean ± s.d. of triplicates from one representative of two independent experiments. **f**, IB analysis of ACSL4, FSP1, NQO1 and vinculin in A375 and MDA-MB-231 cells transduced with either NT or *FSP1*-targeting sgRNAs cultured in riboflavin-deficient medium or medium supplemented with 1 μM riboflavin for 96 h. Data shown are from one representative of two independent experiments. **g**, Epilipidomics analysis of A375 parental cells pre-cultured in riboflavin-deficient medium or medium supplemented with 1 μM riboflavin for 72 h after treatment with DMSO, RSL3 (200 nM) or RSL3 (200 nM) + Lip-1 (500 nM) for 6 h. The heatmap shows three technical replicates from one independent experiment.

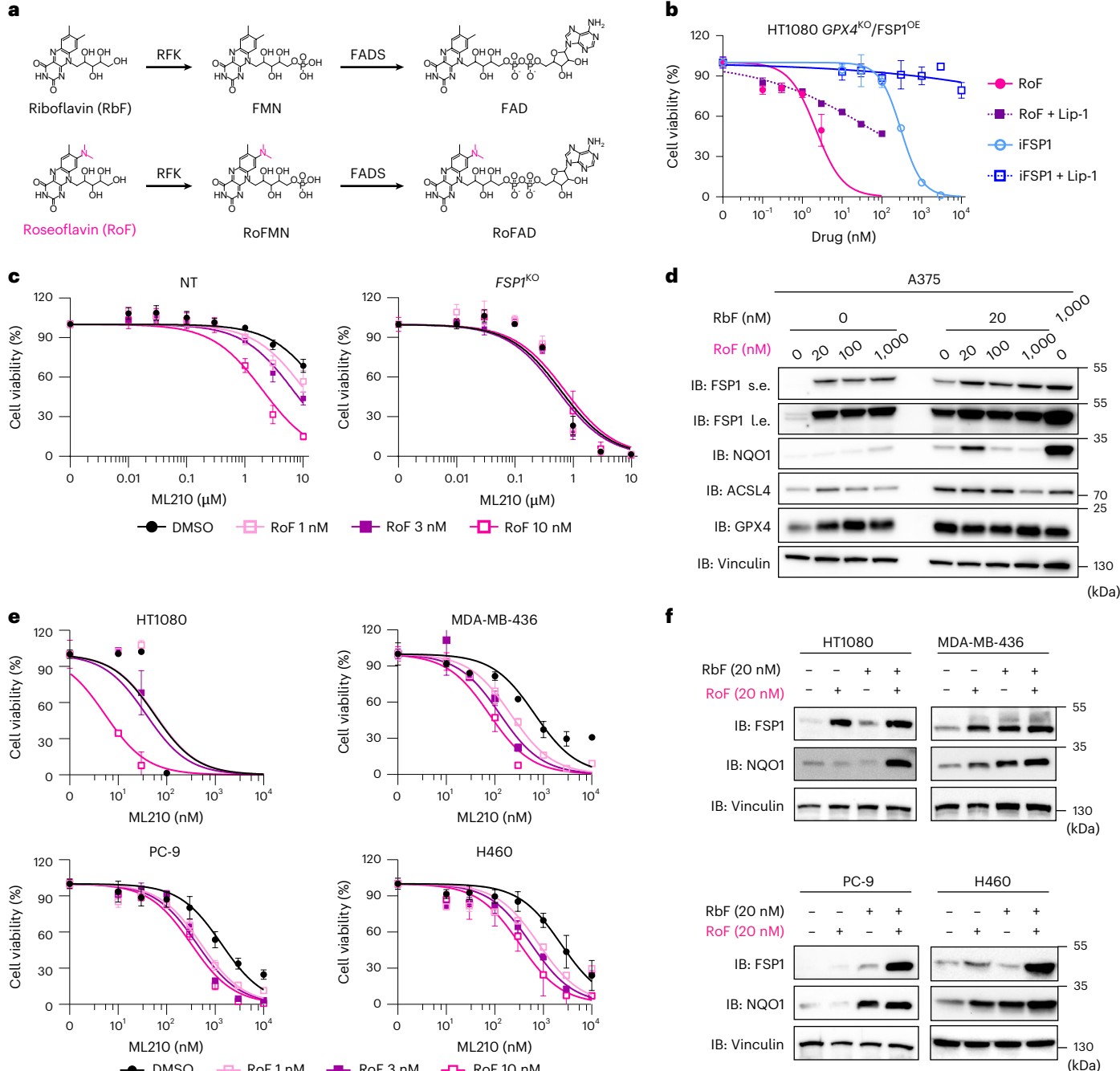

**Fig. 4 | The riboflavin antimetabolite roseoflavin promotes ferroptosis.**
**a**, Schematic representation of the metabolism of riboflavin (RbF) and its analogue roseoflavin (RoF) by riboflavin kinase (encoded by *RFK*) and FADS (encoded by *FLAD1*). **b**, Dose-dependent toxicity of roseoflavin and iFSP1 in the absence or presence of Lip-1 (500 nM) in HT1080$^{GPX4KO/FSP1OE}$ cells. Cell viability was monitored using alamarBlue after 48 h of treatment. Cells were cultured in low-riboflavin medium (20 nM). Data plotted are mean ± s.d. of triplicates from one representative of two independent experiments. **c**, Dose-dependent toxicity of the GPX4 inhibitor ML210 in A375 cells transduced with either NT or *FSP1*-targeting sgRNAs (*FSP1*$^{KO}$) that were pre-treated with roseoflavin (1, 3 and 10 nM) for 48 h. Cell viability was monitored after 48 h of ML210 treatment. Cells were cultured in low-riboflavin medium (20 nM). Data plotted are mean ± s.d.

of triplicates from one representative of two independent experiments. **d**, IB analysis of FSP1, NQO1, ACSL4, GPX4 and vinculin treated with roseoflavin (0, 20, 100 and 1,000 nM) for 96 h in the absence or presence of riboflavin (20 nM). Data shown are from one representative of two independent experiments. s.e., short exposure; l.e., long exposure. **e**, Dose-dependent toxicity of the GPX4 inhibitor ML210 in HT1080, MDA-MB-436, PC-9 and H460 parental cell lines pre-treated with roseoflavin (1, 3 and 10 nM) for 48 h. Cell viability was monitored after 48 h of ML210 treatment. Data plotted are mean ± s.d. of triplicates from one representative of three independent experiments. **f**, IB analysis of FSP1, NQO1 and vinculin in HT1080, MDA-MB-436, PC-9 and H460 parental cell lines treated with roseoflavin (20 nM) for 48 h in the absence or presence of riboflavin (20 nM). Data shown are from one representative of two independent experiments.

or roseoflavin (Fig. 5f) following 96 h of riboflavin deprivation. Notably, some proteins associated with a stress response (NRF2 targets) remained upregulated in roseoflavin-treated cells (Extended Data Fig. 5e,f), suggesting that roseoflavin-derived metabolites can stabilize some

flavoproteins (Fig. 6a), but fail to restore their full functionality and flavin homeostasis.

In parallel, we confirmed that roseoflavin restores FSP1-GFP expression in A375 cells cultured under riboflavin-deficient conditions

by cell sorting and fluorescence analysis. Notably, inhibition of the proteasome with MG132 prevented the loss of FSP1 in the absence of flavin cofactors (Fig. 6b).

Substitutions on the isoalloxazine ring are known to change flavin redox properties substantially[19]. These changes in the electronic density of the ring offer a plausible explanation for why roFAD stabilizes, but inactivates FSP1.

Based on this, we hypothesized that the electron flow within roFAD would be less favourable.

As roFAD is not commercially available, we could not directly measure its reactivity, but we analysed the reactivity of the roFAD precursor, roseoflavin, which retains the isoalloxazine ring. Monitoring NADH oxidation at 340 nm revealed that riboflavin, but not roseoflavin, oxidized NADH (Fig. 6c). Similarly, only riboflavin facilitated resazurin reduction (Fig. 6d).

Although these assays were conducted in aqueous solution and must be interpreted with caution, as the protein environment can further modulate the redox potential of flavins, this combined evidence supports a mechanism in which roseoflavin-derived metabolites incorporate into FSP1 but cannot accept electrons from NAD(P)H, thereby inactivating the enzyme.

Altogether, these results demonstrate that FSP1 activity can be effectively inhibited by riboflavin antimetabolites, offering additional opportunities to target ferroptosis protective mechanisms in cancer cells (Fig. 6e).

## Discussion

We have revealed a critical role for riboflavin in governing ferroptosis via FSP1. Previous work established FSP1 as an essential safeguard against lipid peroxidation, complementing the activity of GPX4; however, the upstream factors regulating FSP1 functionality remained largely unknown. By systematically dissecting FAD biosynthesis, we demonstrate that multiple nodes in the riboflavin–FMN–FAD pathway directly influence the stability of FSP1 and its enzymatic capacity to recycle lipophilic antioxidants. The FAD cofactor is essential for the structural stabilization of FSP1, and its absence leads to misfolding and

ultimately degradation via the ubiquitin–proteasome pathway (UPS). The E3 ligase RNF8 appears to be involved in the degradation of apoFSP1 by the UPS[20]. In particular, we highlight FADS as a key enzyme in the last step of FAD production and reveal that its depletion specifically compromises FSP1, with broad implications for ferroptosis sensitivity and cell viability. These findings enhance our understanding of how cells depend on riboflavin metabolism to preserve membrane integrity under oxidative stress.

Crucially, we observed that standard tissue culture media supply riboflavin at concentrations far exceeding physiological plasma levels, potentially obscuring the impact of the genetic and non-genetic factors impacting riboflavin availability and ultimately ferroptosis sensitivity. Culturing cells under physiological or riboflavin-deficient conditions revealed pronounced changes in FSP1 expression and ferroptosis outcomes, underscoring that moderate fluctuations in riboflavin can alter redox balance. The discrepancy between riboflavin concentrations in vitro and in human plasma is therefore highly relevant for translational research, as a lower availability of riboflavin in vivo may attenuate FSP1-mediated protection against ferroptosis. This insight raises important considerations for mechanistic studies and potential

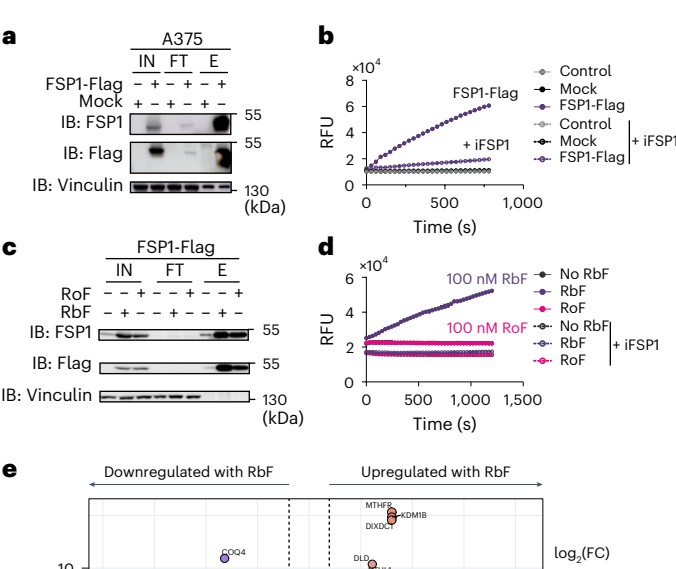

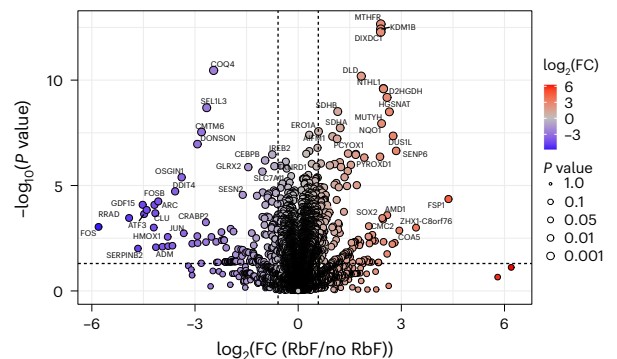

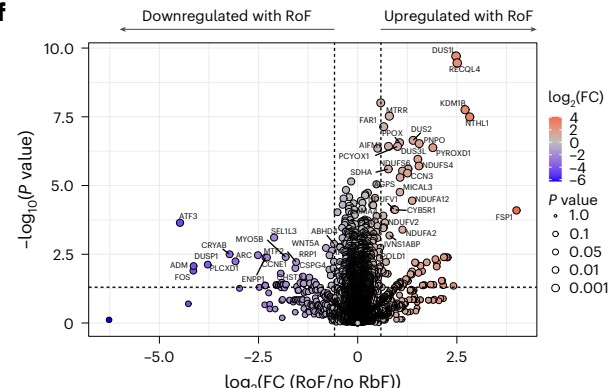

**Fig. 5 | The riboflavin antimetabolite roseoflavin disrupts FSP1 activity.**
**a**, IB analysis of FSP1, flag and vinculin in input (IN), flow-through (FT) and elution (E) fractions from a representative IP assay. Data shown are from one representative of two independent experiments. **b**, Time courses following resazurin reduction (100 μM) by fluorescence in the presence of NADH (200 μM) and elution fractions from IP (mock and FSP1-flag) in the absence or presence of iFSP1 (3 μM) in phosphate buffered saline (PBS). RFU, relative fluorescence unit. The data plotted are from one representative experiment of two independent experiments. **c**, IB analysis of FSP1, flag and vinculin in IN, FT and E fractions from IP. Cells were cultured in riboflavin-deficient medium for 72 h and subsequently supplemented with either riboflavin (100 nM) or roseoflavin (100 nM) for an additional 24 h. Data shown are from one representative of two independent experiments. **d**, Time courses following resazurin reduction (100 μM) by fluorescence in the presence of NADH (200 μM) and the E fractions shown in **c** in the absence or presence of iFSP1 (3 μM) in PBS. Data plotted are mean ± s.d. of duplicates from one representative of two independent experiments. **e**, Volcano plot of quantified proteins showing their change in A375 parental cells cultured in medium supplemented with 100 nM riboflavin for 24 h relative to riboflavin-deficient medium. Proteins are plotted based on their FC (riboflavin/ no riboflavin). Statistical significance was assessed by an unpaired two-sided $t$-test, and values were deemed significant if $P < 0.05$, and the minimum fold change was 1.5. The statistical significance of the respective ratios is plotted on the $y$ axis ($n = 5$ technical replicates). **f**, Volcano plot of quantified proteins showing their change in A375 parental cells cultured in medium supplemented with 100 nM roseoflavin for 24 h relative to riboflavin-deficient medium. Proteins are plotted based on their FC (roseoflavin/no riboflavin). Statistical significance was assessed by an unpaired two-sided $t$-test, and values were deemed significant if $P < 0.05$, and the minimum fold change was 1.5. The statistical significance of the respective ratios is plotted on the $y$ axis ($n = 5$ technical replicates).

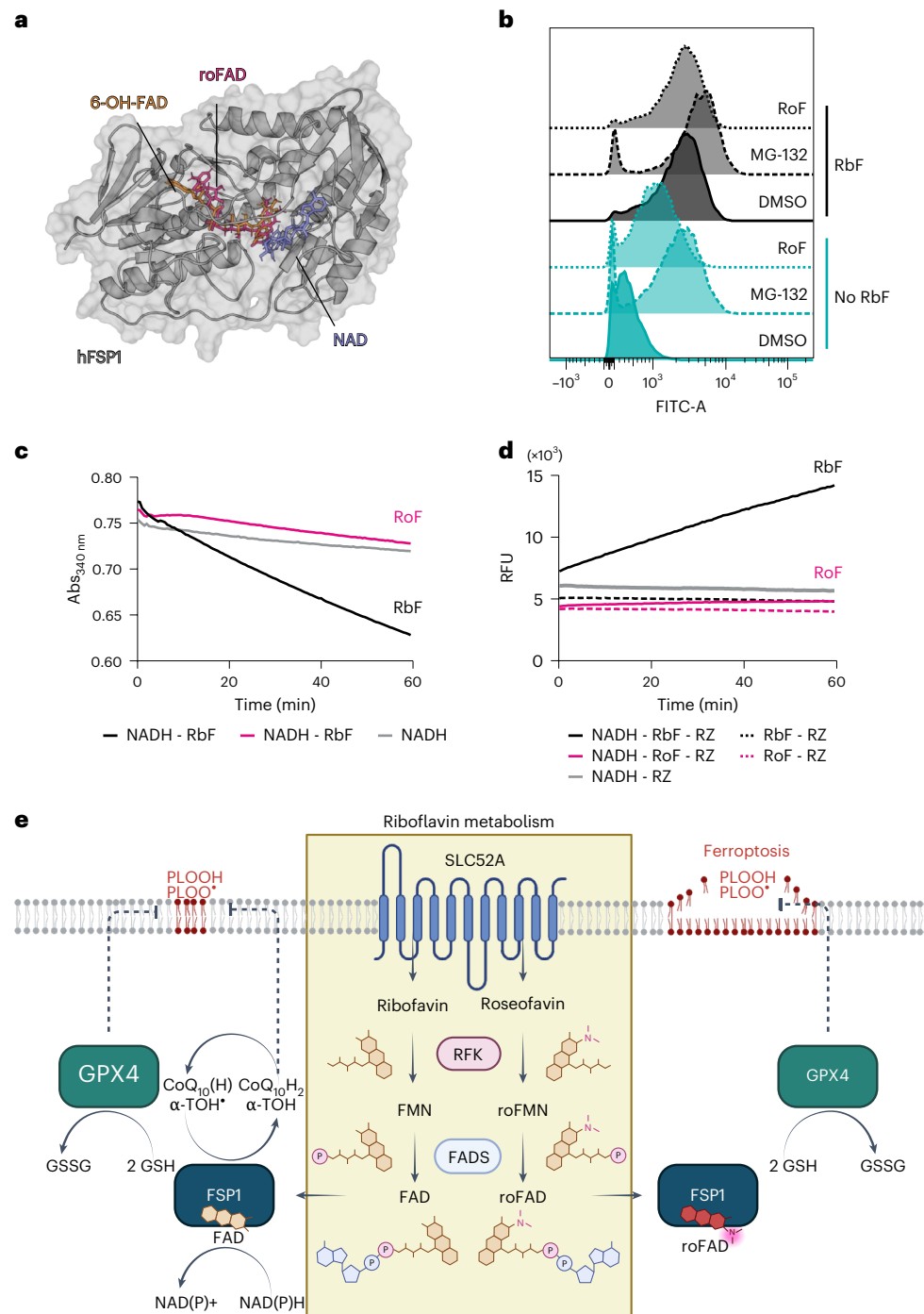

**Fig. 6 | Roseoflavin binds to FSP1 and prevents its degradation by the proteasome. a**, Representative frame of human FSP1 (hFSP1) in complex with ligands from MD simulations. FSP1 is shown in a cartoon representation (grey) in complex with 6-OH-FAD, roFAD and NAD (depicted in stick representation in orange, magenta and blue, respectively). The starting structure was based on PDB 8WIK (FSP1 in complex with 6-OH-FAD). **b**, Fluorescence histograms of A375 overexpressing FSP1-GFP cultured in riboflavin-deficient or 1 μM riboflavin-supplemented medium for 96 h, followed by treatment with DMSO, MG132 (1 μM) or roseoflavin (100 nM) for 24 h. Data shown are from one representative of two independent experiments. **c**, Time courses following NADH oxidation (200 μM) by absorbance at 340 nm in the presence of either riboflavin (25 μM) or roseoflavin (25 μM). The data plot is from one representative experiment of two independent experiments. **d**, Representative time courses following resazurin (RZ) reduction (25 μM) by fluorescence in the presence of the indicated reagents (initial concentrations: 200 μM NADH, 25 μM riboflavin, 25 μM roseoflavin). The data plot is from one representative experiment of two independent experiments. **e**, Riboflavin metabolism supports FSP1 function and promotes ferroptosis resistance. This is therefore a process that riboflavin analogues, such as roseoflavin, can modulate. Panel **e** created in BioRender.com.

clinical contexts in which dietary riboflavin or altered metabolism may drive vulnerability to ferroptosis-related diseases[21].

Interestingly, a parallel can be drawn between the regulation of FSP1 by riboflavin and the regulation of GPX4 by selenium. It is well-established that selenium availability dictates GPX4 protein translation, with selenium metabolism pathways attracting substantial interest as potential modulators of ferroptosis[22–24]. Similarly, our findings suggest that riboflavin—by serving as a precursor for FAD—determines

the abundance and activity of FSP1 independently of mRNA levels. Importantly, both GPX4 and FSP1 highlight how two crucial regulators of ferroptosis cannot be understood solely by analysing the abundance of their mRNA. Instead, the availability of specific micronutrients—selenium for GPX4 and riboflavin for FSP1—is required for their proper translation and functionality. This establishes a broader principle where micronutrients and their metabolic pathways modulate protein functionality and ferroptosis susceptibility, suggesting important opportunities for therapeutic intervention.

Beyond elucidating a role for endogenous riboflavin metabolism, we highlight the value of riboflavin antimetabolites, such as roseoflavin, as potent modulators of FSP1 function. Because of its structural similarity to riboflavin, roseoflavin probably shares efficient tissue distribution and demonstrates activity in the nanomolar range in physiologically relevant riboflavin concentrations—substantially lower than current FSP1 inhibitors. An additional advantage of roseoflavin lies in its uptake and metabolism, which depend on SLC52A2, RFK and FADS. Mutations in these proteins that could affect resistance would also compromise riboflavin uptake and metabolism, ultimately preventing the production of essential cofactors such as FAD and FMN, as recently exemplified in *Plasmodium falciparum*[25]. This dual impact makes it unlikely for cells to selectively develop resistance to roseoflavin without severely impairing riboflavin metabolization and FAD production. In contrast, specific inhibitors of FSP1 are more prone to resistance mechanisms, such as mutations in FSP1 itself or activation of compensatory pathways, as these do not disrupt the broader metabolic network. The reliance of roseoflavin on essential and conserved metabolic pathways could thus provide an important therapeutic advantage.

Altogether, our results show that intracellular flavin metabolism is pivotal to FSP1's protective function against phospholipid peroxidation, operating independently of GPX4. This framework constitutes a previously underappreciated approach for enhancing ferroptosis in cancer cells and other contexts where FSP1 supports survival. Moreover, our work unveils riboflavin's role in the FSP1-driven recycling of lipophilic antioxidants, offering fundamental insights into the complex interactions of nutrients, with important implications for understanding inconsistent outcomes in preclinical and clinical studies of antioxidant therapies[26–28].

## Online content

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

[1]Rudolf Virchow Zentrum (RVZ), Center for Integrative and Translational Bioimaging, University of Würzburg, Würzburg, Germany. [2]European Center for Angioscience, Medical Faculty Mannheim, Heidelberg University, Heidelberg, Germany. [3]Division of Vascular Oncology and Metastasis, German Cancer Research Center Heidelberg (DKFZ–ZMBH Alliance), Heidelberg, Germany. [4]Center of Membrane Biochemistry and Lipid Research, University Hospital and Faculty of Medicine Carl Gustav Carus of TU Dresden, Dresden, Germany. [5]Department of Biotechnology, Institute for Technical Microbiology, Hochschule Mannheim, Mannheim, Germany. [6]Research Unit Signaling and Translation, Helmholtz Zentrum München, Neuherberg, Germany. [7]Institute of Chemistry, Technische Universität Berlin, Berlin, Germany. [8]Heidelberg Institute for Theoretical Studies, Heidelberg, Germany. [9]Department of Biochemistry and Molecular Biology, Theodor Boveri Institute, Biocenter, University of Würzburg, Würzburg, Germany. [10]Comprehensive Cancer Center Mainfranken, University Hospital Würzburg, Würzburg, Germany. [11]Department of Molecular Metabolism, Harvard T.H. Chan School of Public Health, Boston, MA, USA. [12]Institute of Molecular Biotechnology of the Austrian Academy of Science (IMBA), Vienna, Austria. [13]Department of Pathology, University of Würzburg, Würzburg, Germany. [14]Institute of Metabolism and Cell Death, Helmholtz Zentrum München (HMGU), Neuherberg, Germany. [15]Translational Redox Biology, TUM Natural School of Sciences, Technical University of Munich, Garching, Germany. [16]Heidelberg Institute for Stem Cell Technology and Experimental Medicine (HI-STEM GmbH), Heidelberg, Germany. [17]Division of Stem Cells and Cancer, German Cancer Research Center (DKFZ), Heidelberg, Germany. ✉e-mail: pedro.angeli@uni-wuerzburg.de

## Methods

### Chemicals

Lip-1 (Sigma, cat. no. SML1414), riboflavin (Sigma, cat. no. R9504), roseoflavin (Sigma, cat. no. SML1583), RSL3 (Sigma, cat. no. SML2234), ML210 (Sigma, cat. no. SML0521), CAY10566 (MedChemExoress, cat. no. HY-15823), erastin (Selleckchem, cat. no. S724203), BSO (Sigma cat. no. B25159), TRi-1 (MedChemExpress, cat. no. HY-125006), etoposide (MedChemExpress, cat. no. HY-13629), PLX4032 (Selleckchem, cat. no. S1267), auranofin (Sigma, cat. no. A6733), camptothecin (MedChemExpress, cat. no. HY-16560), protamine sulfate (Sigma, cat. no. P3369), bortezomib (PS-341, MedChemExpress, cat. no. HY-10227) and C11-BODIPY (581/591) (Invitrogen, cat. no. D3861) were used in this study.

### Cell lines

The human cancer cell lines HT1080, A375, MDA-MB-231, MDA-MB-436, A549 and H460 were purchased from ATCC (cat. nos. CCL-121, CRL-1619, CRM-HTB-26, HTB-130, CRM-CCL-185 and HTB-177, respectively) and PC-9 cells were purchased from Sigma (cat. no. 90071810). Cells were cultured in high-glucose DMEM GlutaMax (Gibco, cat. no. 31966-021) with 10% fetal bovine serum (FBS; Gibco, ref. A5256701) and 1% penicillin-streptomycin (Gibco, cat. no. 15140122). *GPX4*, *FSP1*, *RFK*, *FLAD1* and *SLC52A2* knockout cells were maintained with Lip-1 (500 nM). Cells were cultured at 37 °C, 5% $CO_2$, and routinely tested for *Mycoplasma* contamination.

### Lentivirus production and transduction

HEK293T cells were used to produce replication-incompetent lentivirus with a third-generation system (pCMV-VSV-G, pRSV_Rev and pMDLg/pRRE plasmids, Addgene cat. nos. 8454, 12253 and 12251). Cells (700,000 well$^{-1}$ in six-well plates) were co-transfected (3 µg DNA, 1:1:1:1, X-tremeGENE HP, Roche, cat. no. 06366236001), and viral supernatants were collected 48 h later, filtered (0.45 µm), and used to transduce target cells (100,000 well$^{-1}$) in medium with protamine sulfate (8 µg ml$^{-1}$) and Lip-1 (500 nM), followed by antibiotic selection.

### CRISPR–Cas9 screen

HT1080 *GPX4*$^{KO}$ FSP1-Flag$^{OE}$ Cas9 cells were transduced with a lentiviral human VBC CRISPR library (15,062 sgRNAs targeting 3,007 genes, 5 sgRNAs per gene) at 500× coverage (26 million cells, 1,000 µg DNA) with Lip-1 (500 nM). After 48 h, cells were selected with G418 (1 mg ml$^{-1}$) for seven days, collected at day 0, and split into dimethyl sulfoxide (DMSO) or Lip-1 (500 nM) conditions for an additional 14 days. Genomic DNA was isolated (Qiagen, cat. no. 69504), polymerase chain reaction (PCR)-amplified, sequenced (Illumina MiniSeq High-Output, 25 M reads), and analysed using the MaGeCK pipeline and visualized with MaGeCKFlute.

### CRISPR–Cas9-mediated gene knockout

sgRNAs were selected using the VBC score (https://www.vbc-score.org/) and cloned via annealed oligonucleotides (Eurofins Genomics) into BsmBI-digested lentiCRISPRv2-puro or lentiCRISPRv2-blast (Addgene, cat. nos. 98290 and 98293, respectively) and selected for 4–7 days. Knockout efficiency was confirmed by immunoblotting (when antibodies were available) and genomic DNA sequencing; *FLAD1* knockout cells were additionally validated by relative FAD measurements. Cells were used as pools unless otherwise noted. The oligonucleotide sequences used are listed in Supplementary Table 4.

### Stable protein overexpression

HT1080 and A375 cells overexpressing FSP1, FADS or SLC52A2 were generated via lentiviral transduction. Coding sequences (CCDS7297.1, CCDS1078.1, CCDS6423.1) were codon-optimized (IDT), synthesized as gBlocks, and cloned into p442-IRES-blast or p442-IRES-neo vectors using ClonExpress II (Vazyme, cat. no. C112). Plasmids were then delivered to target cells via lentivirus as described above.

### Cell viability assays

Cells were seeded at 1,000 cells well$^{-1}$ on 96-well plates and, after 4–6 h, treated with the following compounds: RSL3, ML210, iFSP1, Lip-1, erastin, BSO, TRi-1, etoposide, PLX4032, auranofin, camptothecin, bortezomib and roseoflavin. Cell viability was assessed 72–96 h after treatment using the alamarBlue method as an indicator of viable cells. Viability was estimated by measuring fluorescence using a 540/35-nm excitation filter and 590/20-nm emission on a Spark microplate reader (Tecan).

### Live-cell imaging and proliferation analysis

Cell proliferation was assessed using the IncuCyte Live-Cell Analysis System S3/SX1 (Sartorius). Plates were placed in the IncuCyte incubator, and phase-contrast images were automatically acquired every 6 h over a period of 5–7 days. Image acquisition was performed at ×10 magnification, and cell confluence was quantified using the IncuCyte integrated analysis software according to the manufacturer's guidelines.

### Cell lysis and immunoblotting

Cells were lysed in RIPA buffer with protease inhibitors (Roche, cat. no. 11697498001) and centrifuged at 20,000g for 30 min at 4 °C. Protein concentrations were measured by BCA assay (Thermo Fisher, cat. no. 23235) and samples heated in sodium dodecyl sulfate (SDS) loading buffer (70–90 °C, 10 min). Proteins (20 µg) were separated on 12% SDS–polyacrylamide gel electrophoresis (PAGE) gels, transferred to nitrocellulose/polyvinylidene fluoride membranes, blocked (5% milk in Tris-buffered saline with Tween 20 for 1 h at room temperature (r.t.)), and incubated overnight at 4 °C with the following primary antibodies: FSP1 (1:10, clone 6D8-11, Helmholtz Munich), GPX4 (1:1,000, Abcam cat. no. ab125066 or 1:500, Proteintech cat. no. 67763-1-Ig), ACSL4 (1:500, Santa Cruz cat. no. sc-271800), FADS (1:250, Santa Cruz cat. no. sc-376819), RFK (1:250, Santa Cruz cat. no. sc-398830 or 1:500, Abbexa cat. no. abx124688), NQO1 (1:5,000, Santa Cruz cat. no. sc-32793), Flag (1:1,000, Sigma cat. no. F3165), β-actin (1:5,000, Sigma cat. no. A5441) and vinculin (1:1,000, Santa Cruz cat. no. sc-73614). After washing, the membranes were incubated with horseradish peroxidase-conjugated secondary antibodies (1:3,000, Cell Signaling cat. nos. 7074, 7076 and 7077) for 2 h at r.t. and detected using enhanced chemiluminescence (BioRad 107-5061) on an Amersham ImageQuant 800 system (Cytiva).

### Genotyping

*FLAD1* and *SLC52A2* single-clone knockouts were confirmed by genotyping. Genomic DNA was extracted via proteinase K digestion, and sgRNA-targeted regions were PCR-amplified and Sanger-sequenced to identify WT and knockout alleles. Indels were analysed using CRISP-ID[29]. For *FLAD1* knockout cells, PCR products were cloned into pJET1.2 (CloneJET PCR Cloning Kit, Thermo Fisher, cat. no. K1231), and five colonies were PCR-amplified and Sanger-sequenced. The oligonucleotide sequences for the primers used are listed in Supplementary Table 4.

### Preparation of riboflavin-deficient medium

DMEM without riboflavin (PAN Biotech, cat. no. P04-03584) was supplemented with GlutaMax (Gibco, cat. no. 35050-061), 10% dialysed FBS (see below), 1% penicillin-streptomycin (Gibco, cat. no. 15140122), 15 mg l$^{-1}$ phenol red (Sigma, cat. no. P0290) and riboflavin (1 µM, unless otherwise noted). Riboflavin stock (20 mg l$^{-1}$, Sigma, cat. no. R9504) was freshly prepared in PBS and sterilized through a 0.22-µm filter.

### Dialysis of FBS

FBS was dialysed to remove small molecules, including riboflavin, using cellulose tubing (Sigma, cat. no. D9527, 14-kDa cutoff) against PBS (1:10 vol/vol) with buffer changes every 24 h for five cycles at 4 °C under constant stirring. The dialysed serum was sterile-filtered (0.22 µm), aseptically aliquoted, and stored at −20 °C until use.

## Riboflavin, FMN and FAD measurements

Water-soluble metabolite levels were analysed by liquid chromatography–mass spectrometry (LC–MS). For sample preparation, $1 \times 10^6$ cells were collected, washed with PBS, and rapidly frozen in liquid nitrogen. The metabolites were extracted with 500 µl of ice-cold MeOH/$H_2O$ (80:20, vol/vol) containing 0.01 µM lamivudine and 1 µM each of $D_2$-glucose, $D_4$-succinate, $D_5$-glycine and $^{15}N$-glutamate (Sigma-Aldrich). After centrifugation, the supernatants were evaporated using a rotary evaporator (Savant, Thermo Fisher Scientific). Dried extracts were redissolved in 150 µl of 5 mM $NH_4OAc$ in $CH_3CN/H_2O$ (50:50, vol/vol), and 20 µl of supernatant was transferred to LC vials.

Metabolites were separated on an XBridge Premier BEH Amide UPLC column (2.5 µm, 100 × 2.1 mm; Waters) using a DIONEX Ultimate 3000 UHPLC system (Thermo Fisher Scientific) at 45 °C and 200 µl min$^{-1}$. Solvent A consisted of 5 mM $NH_4OAc$ in $CH_3CN/H_2O$ (40:60, vol/vol) and solvent B consisted of 5 mM $NH_4OAc$ in $CH_3CN/H_2O$ (95:5, vol/vol). The linear gradient started after 2 min at 100% B, decreased to 10% B over 23 min, was held for 16 min, and returned to 100% B within 2 min, followed by a 7-min pre-run equilibration at 100% B before each injection.

Mass spectrometry was performed on a high-resolution Q Exactive instrument equipped with a HESI probe (Thermo Fisher Scientific) in alternating positive and negative full MS modes (69–1,000 $m/z$, 70-K resolution). The ESI source parameters were as follows: sheath gas = 30, auxiliary gas = 1, sweep gas = 0, aux gas heater = 120 °C, spray voltage = 3 kV, capillary temperature = 320 °C and S-lens RF level = 50. Extracted ion chromatograms and signal quantification were performed in TraceFinder V 5.1 (Thermo Fisher Scientific) by integrating peaks corresponding to the calculated monoisotopic masses (MIM ± H$^+$ ± 3 mMU).

## RNA extraction, cDNA synthesis and reverse transcription quantitative PCR

Total RNA was isolated from cell pellets using TRIzol (Invitrogen, cat. no. 15596026) following the manufacturer's instructions. Complementary DNA (cDNA) was synthesized with a HiScript III 1st Strand Kit (Vazyme, cat. no. R312) and hexamer primers. Reverse transcription quantitative PCR was performed on a Mastercycler ep realplex (Eppendorf) or CFX Connect (Biorad) system using SYBR Green. Gene expression was normalized to ACTB using the ΔΔCt method. Oligonucleotide sequences for the primers used are listed in Supplementary Table 4.

## Lipid peroxidation assays using C11-BODIPY (581/591)

A total of 50,000 cells well$^{-1}$ were seeded in six-well plates and treated the next day with DMSO, RSL3 (200 nM) or RSL3 (200 nM) + Lip-1 (500 nM) for 6 h. Cells were then stained with C11-BODIPY (1 µM, Invitrogen, cat. no. D3861) for 30 min at 37 °C, collected, and resuspended in PBS + 2% FBS for flow cytometry (FACS Canto II, BD Biosciences). FITC/PE signals were recorded (≥10,000 events sample$^{-1}$) and analysed using FlowJo. The ratio of FITC/PE (oxidized/reduced ratio) was calculated as follows: (median FITC-A fluorescence – median FITC-A fluorescence of unstained samples)/(median PE-A fluorescence – median PE-A fluorescence of unstained samples).

## Epilipidomics analysis

A375 parental cells (500,000 cells) were seeded on 15-cm dishes in standard DMEM. After 4 h, the cells were cultured for 72 h in either riboflavin-deficient medium or medium supplemented with riboflavin (1 µM). Subsequently, the cells were treated for 6 h with DMSO, RSL3 (200 nM) or RSL3 + Lip-1 (500 nM), collected, washed with PBS, snap-frozen in liquid nitrogen, and stored at −80 °C until analysis.

Lipids were extracted following the Folch protocol[30]. All solvents contained 1 µg ml$^{-1}$ butylated hydroxytoluene and were cooled on ice before use. The cell pellets were washed, centrifuged, and resuspended in 50 µl of $H_2O$ in 2-ml tubes. SPLASH LIPIDOMIX (5 µl; Avanti

Polar Lipids) was added, and the samples were vortexed for 10 s and kept on ice for 15 min. Ice-cold MeOH (365 µl) and $CHCl_3$ (740 µl) were sequentially added, each followed by 10 s of vortexing. The samples were incubated for 1 h at 4 °C in a rotary shaker (40 r.p.m.). Phase separation was induced with 225 µl of $H_2O$, followed by vortexing (10 s) and centrifugation (2,000g, 10 min). The lower phase (770 µl) was collected and dried in a vacuum concentrator (40 mbar, 20 °C). Lipids from the upper phase were re-extracted with 400 µl $CHCl_3$:MeOH (2:1, vol/vol), vortexed (10 s) and 100 µl of $H_2O$ was added to promote phase separation, followed by vortexing (10 s), centrifugation and collection of the lower phase (380 µl), which was combined with the first extract and dried as above.

Dried lipid extracts were reconstituted in 125 µl i-PrOH, centrifuged (10,000g, 5 min), then 120 µl was transferred to glass vials for LC–MS analysis. The remaining upper phase was dried (20 mbar, 20 °C) and used for total protein quantification. Group-specific quality control (gQC) samples were prepared by pooling 40 µl of individual samples; total quality control (tQC) samples were obtained by mixing 40 µl of each gQC. Reversed-phase LC (RPLC) was performed on a Vanquish Horizon system (Thermo Fisher Scientific) equipped with an Accucore C30 column (150 × 2.1 mm, 2.6 µm, 150 Å). Lipids were separated at 50 °C and 0.3 ml min$^{-1}$ using solvent A (MeOH/$H_2O$, 1:1, vol/vol) and solvent B (i-PrOH/MeCN/$H_2O$, 85:15:5, vol/vol/vol), both containing 5 mM $NH_4HCOO$ and 0.1% (vol/vol) HCOOH. The gradient was as follows: 0–20 min, 10–86% B (curve 4); 20–22 min, 86–95% B (curve 5); 22–26 min, 95% B isocratic; 26–26.1 min, 95–10% B (curve 5); 26.1–31.1 min, 10% B re-equilibration.

Mass spectrometry was performed on an Orbitrap Exploris 240 instrument (Thermo Fisher Scientific) equipped with a heated electrospray ionization (HESI) source and EASY-IC for lock mass correction. The HESI parameters were as follows: sheath gas = 40, aux gas = 10, sweep gas = 1, spray voltage = 3.5 kV (positive) or 2.5 kV (negative), ion transfer tube = 300 °C, vaporizer = 370 °C, S-lens RF = 35%. EASY-IC lock mass correction was set to RunStart. For oxidized lipid identification, 6-µl gQC samples were injected and analysed in polarity-switching mode (0–22.5-min negative; 22.5–39.9-min positive) using the semi-targeted data-dependent acquisition mode (stDDA) with six MS/MS scans per cycle. Full scans (MS1) settings were as follows: 60,000 resolution ($m/z$ 200), scan range = 500–980 $m/z$ (negative) or 480–1,000 $m/z$ (positive), automatic gain control (AGC) standard, maximum injection time auto, 1 microscan. StDDA MS2 scans used a 1.5-$m/z$ precursor selection isolation window, 30,000 resolution ($m/z$ 200), stepped higher-energy collisional dissociation (HCD; normalized collision energy (nCE) 22–32–43% negative; 32–43–54% positive), AGC $1 \times 10^5$, 200-ms max injection, 2 microscans. The following filters were applied before MS2 scans: dynamic exclusion after five times if occurring within 6 s, exclusion duration of 5 s, mass tolerance of ±5 ppm, isotope exclusion; allowed charge state of 1; targeted mass inclusion using the mass list of in silico predicted oxidized lipids (667 individual $m/z$ values split into three LC–MS runs), inclusion mass tolerance of ±5 ppm. Data were acquired in profile mode.

Oxidized lipids were identified using LPPtiger2[31,32] (Fedorova Lab). For relative quantification of oxidized lipids identified in stDDA runs, retention time–scheduled parallel reaction monitoring (PRM) was performed in polarity-switch mode (0–22-min negative; 22–40-min positive) at 15,000 resolution ($m/z$ 200), AGC $1 \times 10^5$ and 100-ms max injection. The isolation window for precursor selection was 1.5 $m/z$. HCD nCE for every target was chosen based on optimization PRM runs of gQC samples (fixed nCE from 15 to 40%). Data were acquired in profile mode. PRM data were processed with Skyline v.24.1.0.199 (MacCoss Lab)[33], considering fragment anions of oxidized fatty acyl chains as the quantifier. Peak areas were normalized by appropriate lipid species from the SPLASH LIPIDOMIX Mass Spec Standard (for example, LPC(18:1(d7)), LPE(18:1(d7)), PC(15:0/18:1(d7)) and PE(15:0/18:1(d7))), and protein concentration measured for the corresponding sample.

Normalized peak areas were further log-transformed and autoscaled using the MetaboAnalyst online platform (https://www.metaboanalyst.ca; Xia Lab)[34].

## Proteomic analysis

For each condition, $1 \times 10^6$ cells were collected, washed with PBS, snap-frozen in liquid nitrogen, and stored at −80 °C. For proteomics, cell pellets were lysed in RIPA buffer supplemented with protease and phosphatase inhibitors (Roche, cat. nos. 11697498001 and 4906845001, respectively) and 1% (vol/vol) benzonase (Merck, cat. no. E1014). Lysates were sonicated and centrifuged at 17,000g for 2 h at 4 °C. Protein concentrations were determined by BCA assay (Thermo Fisher, cat. no. 23235).

Tryptic digestion (input: 10 μg) was performed using an AssayMAP Bravo liquid handling system (Agilent Technologies) running the autoSP3 protocol according to ref. 35. Peptides were vacuum-dried and stored at −20 °C until LC−MS/MS analysis.

The LC−MS/MS analysis was carried out on an Ultimate 3000 UPLC system directly connected to an Orbitrap Exploris 480 mass spectrometer (both Thermo Fisher Scientific) for a total of 120 min, injecting an equivalent of 1 μg of peptide. The peptides were online desalted on a trapping cartridge (Acclaim PepMap300 C18, 5 μm, 300-Å-wide pore; Thermo Fisher Scientific) for 3 min using a 30 μl min$^{-1}$ flow of 0.05% trifluoroacetic acid (TFA) in water. The analytical multistep gradient (300 nl min$^{-1}$) was performed using a nanoEase MZ Peptide analytical column (1.7 μm, 300 Å, 75 μm × 200 mm; Waters). Solvent A was 0.1% formic acid in water and solvent B was 0.1% formic acid in acetonitrile. The gradient was as follows: 4–30% B over 102 min, ramp to 78% B, hold for 2 min, return to 2% B, and equilibrate for 10 min.

Data were acquired in data-independent acquisition (DIA) mode. Full scans were recorded at 120-K resolution (380–1,400 $m/z$; AGC target of 300%; max injection time (IT) of 45 ms) followed by 47 MS2 windows covering the mass range 400–1,000 $m/z$ with variable width and overlapping 1-Da (30-K resolution (AGC target of 1,000%; max IT of 54 ms; HCD collision energy of 28%). Each sample run was followed by a 40-min wash to prevent carryover. Instrument performance was monitored regularly (approximately every 48 h) using a standard reference sample and an in-house Shiny QC application. Sample preparation and LC−MS/MS acquisition were performed in block-randomized order[36].

DIA RAW files were analysed using Spectronaut software (Biognosys, v.19.1.240724.62635)[37] in directDIA+ (deep) library-free mode. Default settings were applied with the following adaptations: precursor PEP cutoff = 0.01; protein Q-value cutoff (run) = 0.01; protein PEP cutoff = 0.01. Quantification parameters included Proteotypicity Filter = Only Protein Group Specific, Protein LFQ Method = MaxLFQ and Quantification Window = Not Synchronized (SN 17). Data were searched against the UniProt human reference proteome (20,597 entries; release 9 February 2024) and the MaxQuant contaminant database (246 entries; 22 December 2022). All experimental conditions were included in the Spectronaut analysis set-up.

## Statistical analysis of proteomics data

Statistical analysis was performed using MetaboAnalyst software[38], with Spectronaut output files as input. After Spectronaut analysis, the dataset comprised the identified proteins, along with LFQ intensities. The data were filtered to remove features matched with contaminant or reverse sequences. The missing values were replaced with limits of detection (1/5 of minimum positive values of corresponding variables) and data were log$_{10}$-transformed and used for further statistical analysis. The FC threshold was taken to be 1.5, and the RAW $P$-value cutoff was 0.05 to get the differentially expressed proteins (DEPs). A volcano plot and heatmap representing the expression of top DEPs was also obtained from statistical analysis in MetaboAnalyst. The Metascape tool[39] was used to obtain statistically enriched terms

(GO/Kyoto Encyclopedia of Genes and Genomes pathway (KEGG) terms, canonical pathways) for all DEPs. All analyses were performed in five technical replicates.

## MD simulations

The crystal structure of the human FSP1 complex (PDB 8WIK)[40] was used as the starting point to prepare four systems: NAD−6-OH-FAD, FSP1−NAD−FAD, FSP1−NAD−roFAD and FSP1−NAD. The FAD and roFAD structures were generated by substituting the 6FA ligand (6-OH-FAD) present in PDB 8WIK. The protonation states of all residues at pH 7.4 were determined using the Protein Preparation Wizard tool in Maestro (Schrödinger v.2024.2)[41,42]. Ligand parameters for NAD$^+$ were taken from the Manchester AMBER parameter database[43]. Partial charges for FAD and roFAD were determined using AmberTools23 (ref. 44) via the restrained electrostatic potential (RESP) method. A quantum-mechanical calculation was performed to obtain the electrostatic potentials using Gaussian 09 (ref. 45), Hartree−Fock, the 6-31G* basis set and full optimization. Bond, angle, torsion and van der Waals parameters were generated using the general AMBER force field (GAFF)[46] for cofactors and the AMBER-ILDN force field[47] for the protein. Three independent simulations were performed for each system using GROMACS 2024.2[48]. The systems were solvated with TIP3P water molecules[49] with a minimum margin of 10 Å, and Na$^+$/Cl$^-$ ions were added to ensure system neutrality at an ion concentration of 150 mM. Energy minimization was conducted for 5,000 steps using the steepest descent method with positional restraints of 1,000 kJ mol$^{-1}$ nm$^{-2}$ applied to heavy atoms of the protein and cofactors.

The systems were then equilibrated with an NVT ensemble to achieve constant temperature at 300 K using the velocity rescaling thermostat[50], followed by equilibration to a constant pressure of 1 bar using the cell rescaling barostat[51]. Next, positional restraints were gradually lifted from the heavy atoms of the protein and the cofactors (500, 100 and 10 kJ mol$^{-1}$ nm$^{-2}$). Production runs of 500 ns were performed with a timestep of 2 fs. The temperature coupling was achieved with the velocity rescaling at a time constant of 0.1 ps, and pressure coupling was achieved with cell rescaling at a compressibility of $4.5 \times 10^{-5}$ bar$^{-1}$ and time constant of 5 ps. The LINCS algorithm[52] was used to constrain covalent hydrogen bonds, and SETTLE[53] was applied to constrain the solvent bond lengths. The particle mesh Ewald (PME)[54,55] method was used to calculated the electrostatic forces with a real-space cutoff of 1.2 nm, PME order of 4 and a Fourier grid spacing of 1.2 Å. A cutoff of 1.2 nm was used for calculation of the van der Waals interactions. Data analyses−including RMSD and RMSF−were performed using GROMACS 2024.2 (ref. 48).

## Immunoprecipitation

For IP of Flag-tagged proteins, equal amounts of total protein (4 mg) were incubated with 50 μl of Pierce Anti-DYKDDDDK magnetic agarose beads (Invitrogen, cat. no. A36797) overnight at 4 °C with gentle rotation. Beads were washed three times with lysis buffer and then eluted with 100 μl of 1.5 mg ml$^{-1}$ 3× Flag peptide (MedChemExpress, cat. no. HY-P0319) overnight at 4 °C with rotation. The input (IN), flow-through (FT) and eluted (E) fractions were analysed by immunoblotting. Eluted fractions were desalted with Zeba spin desalting columns (Thermo Fisher, cat. no. 89889) and used for downstream enzymatic assays. For negative controls, lysates from mock-transfected cells were processed in parallel under identical conditions.

## Enzymatic activity assays

NADH (200 μM), FSP1 (500 nM) and resazurin (100 μM) were mixed in 200 μl of PBS in a 96-well plate at room temperature. Resazurin reduction was monitored by fluorescence ($\lambda_{ex} = 540$ nm, $\lambda_{em} = 590$ nm) every 30 s on a Spark microplate reader (Tecan). Where indicated, iFSP1 (3 μM) was added to inhibit the reaction.

### Reaction of riboflavin and roseoflavin with NADH

NADH (200 µM), riboflavin (25 µM) or roseoflavin (25 µM) were mixed on a 96-well plate in a final volume of 200 µl in PBS at room temperature. NADH oxidation was followed by absorbance at 340 nm, recorded every 30 s on a Spark microplate reader (Tecan). Alternatively, the reaction was followed by resazurin reduction (25 µM) by fluorescence ($\lambda_{ex}$ = 540 nm and $\lambda_{em}$ = 590 nm).

### Human cancer cell line datasets

Human cancer cell line datasets were obtained from the DepMap portal (https://depmap.org/portal/, v.23Q2).

### Statistics and reproducibility

All experiments (except those described otherwise in the legend) were performed independently at least twice. Graphs were generated using GraphPad Prism v.10 (GraphPad Software) if not stated otherwise.

### Sample size determination

No statistical methods were used to predetermine sample sizes, but our sample sizes are similar to those reported in previous publications[6,8].

### Assumptions for statistical tests

The data distribution was assumed to be normal, but this was not formally tested.

### Randomization

Samples for each proteomics experiment were randomized before data acquisition and analysis.

### Blinding strategy

Data collection and analysis were not performed blind to the conditions of the experiments.

### Data exclusion

In very rare cases, single values of technical triplicates were excluded from the analysis due to cell clumps/uneven plating.

### Materials availability

All unique/stable reagents generated in this study are available from the lead contact with a completed materials transfer agreement.

### Reporting Summary

Further information on research design is available in the Nature Portfolio Reporting Summary linked to this Article.

## Data availability

Mass spectrometry data have been deposited in ProteomeXchange with the primary accession code PXD061038 (https://www.ebi.ac.uk/pride/archive/projects/PXD061038). All other data supporting the findings of this study are available from the corresponding author on reasonable request. Source data are provided with this paper.

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

## Acknowledgements

J.P.F.A. acknowledges support from the Junior Group Leader programme of the Rudolf Virchow Center, University of Würzburg, and additional support from the Deutsche Forschungsgemeinschaft (DFG; FR 3746/3-1, FR 3746/5-1, FR 3746/6-1, CRC205 [INST 269/886-1]); TRR 387/1 [514894665] to J.P.F.A.), the EU-H2020 (ERC-Consolidator Grant, DeciFERR to J.P.F.A.), the Deutsche Jose Carreras Leukämie Stiftung (DJCLS 01R/2022 to J.P.F.A.) and the São Paulo Research Foundation (2023/04397-4 to J.P.F.A.). A.M. and A.F.N.A. acknowledge funding from the DFG under Germany's Excellence Strategy (EXC 2008/1-390540038 – UniSysCat). J.M.U. acknowledges support from the Ludwig Center at Harvard. M. Conrad acknowledges support from the DFG (CO 291/7-1, Priority Program SPP 2306 (CO 291/9-1, #461385412; CO 291/10-1, #461507177) and the CRC TRR 353 (CO 291/11-1; #471011418)), the European Research Council (ERC) under the European Union's Horizon 2020 research and innovation programme (grant agreement no. GA 884754) and the German Federal Ministry of Education and Research (BMBF) FERROPATH (01EJ2205B) (to M. Conrad and B.P.). M.F. acknowledges support from 'Sonderzuweisung zur Unterstützung profilbestimmender Struktureinheiten' by the SMWK to TUD, TG70 by Sächsische Aufbaubank and SMWK, co-financed with tax funds on the basis of the budget passed by the Saxon State Parliament (to M.F.), DFG (FE 1236/5-1, FE 1236/8-1 to M.F.) and Bundesministerium für Bildung und Forschung (031L0315A, DEEP_HCC and 01EJ2205A, FERROPath to M.F.). H.A. acknowledges support from the DFG Priority Program SPP 2306 (AL 1533/5-1 to H.A.). We also acknowledge the German Cancer Research Center (DKFZ) Proteomics Core Facility.

## Author contributions

Conceptualization was provided by J.P.F.A. Methodology was provided by V.S., A.F.d.S., F.P.F., Z.C., S.S., L.S., S.B., M.M., H.A. and J.P.F.A. Formal analysis was carried out by V.S., I.d.S., B.G., A.F.d.S., F.P.F., Z.C., M.D., P.N., J.T., A.M., A.F.N.A., W.S., S.M., M.F. and H.A. Investigations were performed by V.S., I.d.S., B.G., A.F.d.S., F.P.F., Z.C., S.S., M.D., P.N., L.S., S.B., J.T., A.M., A.F.N.A., J.B., C.A.-S., W.S., M. Chaufan, M.K., M.P. and B.P. Visualization was provided by V.S., I.d.S., B.G., A.F.d.S., F.P.F., Z.C., S.S., P.N. and J.P.F.A. Resources were provided by M.M., M.E., R.B., J.M.U., U.E., H.G.A., K.H., S.M., M. Conrad, M.F., H.A. and J.P.F.A. The original draft was written by J.P.F.A. Review and editing was performed by all authors. Project administration was carried out by J.P.F.A. and supervision by A.F.N.A., M.M., J.M.U., H.G.A., K.H., S.M., M. Conrad, M.F., H.A. and J.P.F.A. Funding acquisition was carried out by A.F.N.A., B.P., M. Conrad, M.F., H.A. and J.P.F.A.

## Funding

## Competing interests

M. Conrad and B.P. are co-founders and shareholders of ROSCUE Therapeutics GmbH. The remaining authors declare no competing interests.

## Additional information

**Extended data** is available for this Paper at https://doi.org/10.1038/s41556-025-01856-x.

**Correspondence and requests for materials** should be addressed to José Pedro Friedmann Angeli.

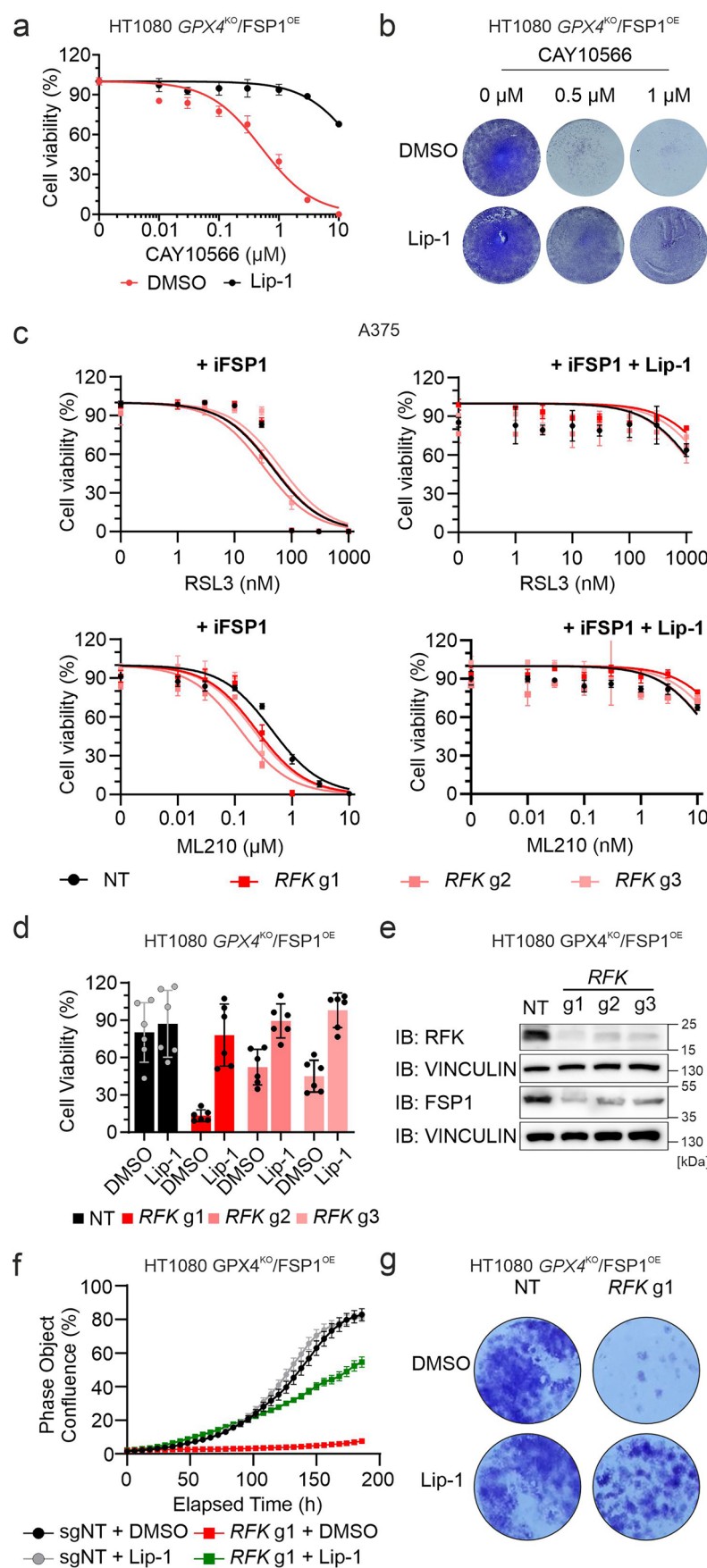

**Extended Data Fig. 1 | See next page for caption.**

**Extended Data Fig. 1 | Identification of factors supporting FSP1 function.**
**a**, Dose-dependent toxicity of the SCD1 inhibitor CAY10566 in the absence
or presence of the ferroptosis inhibitor Lip-1 (500 nM) in HT1080$^{GPX4KO/FSP1OE}$
cells. Cell viability was monitored using Alamar blue after 144 h of treatment.
Data plotted are mean ± SD of triplicates from one representative of three
independent experiments **b**, Dose-dependent toxicity of CAY10566 in the
absence or presence of Lip-1 (500 nM) in HT1080 $GPX4^{KO}$/FSP1$^{OE}$ cells. Cell
viability was monitored by crystal violet staining after 6 days of treatment. Data
shown is one representative of two independent experiments. **c**, Dose-dependent
toxicity of RSL3 in the presence of iFSP1 (3 μM) and/or Lip-1 (500 nM) in A375
cells transduced with either a non-targeting control (NT) or three different
$RFK$-targeting sgRNAs. Cell viability was monitored using Alamar blue after 72 h
of treatment. Data plotted are mean ± SD of triplicates from one representative
of two independent experiments. **d**, Cell viability of HT1080$^{GPX4KO/FSP1OE}$ cells

transduced with either a non-targeting control (NT) or three different $RFK$-
targeting sgRNAs in the absence or presence of Lip-1 (500 nM). Cell viability was
monitored after 96 h. Data plotted are mean ± SD from one representative of
three independent experiments. **e**, Immunoblot (IB) analysis of RFK, FSP1 and
vinculin in HT1080$^{GPX4KO/FSP1OE}$ cells transduced with either a non-targeting control
(NT) or three different $RFK$-targeting sgRNAs. Data shown is one representative
of two independent experiments. **f**, Growth curves showing the percentage of
confluence over time for HT1080$^{GPX4KO/FSP1OE}$ cells transduced with either a non-
targeting control (NT) or an $RFK$-targeting sgRNAs in the absence or presence of
Lip-1 (500 nM). Data plotted are mean ± SEM. The error was calculated per image.
The experiment was performed once. **g**, Cell viability of HT1080$^{GPX4KO/FSP1OE}$ cells
transduced with either a non-targeting control (NT) or an $RFK$-targeting sgRNAs
in the absence or presence of Lip-1 (500 nM). Cell viability was monitored by
crystal violet staining after 96 h. The experiment was performed once.

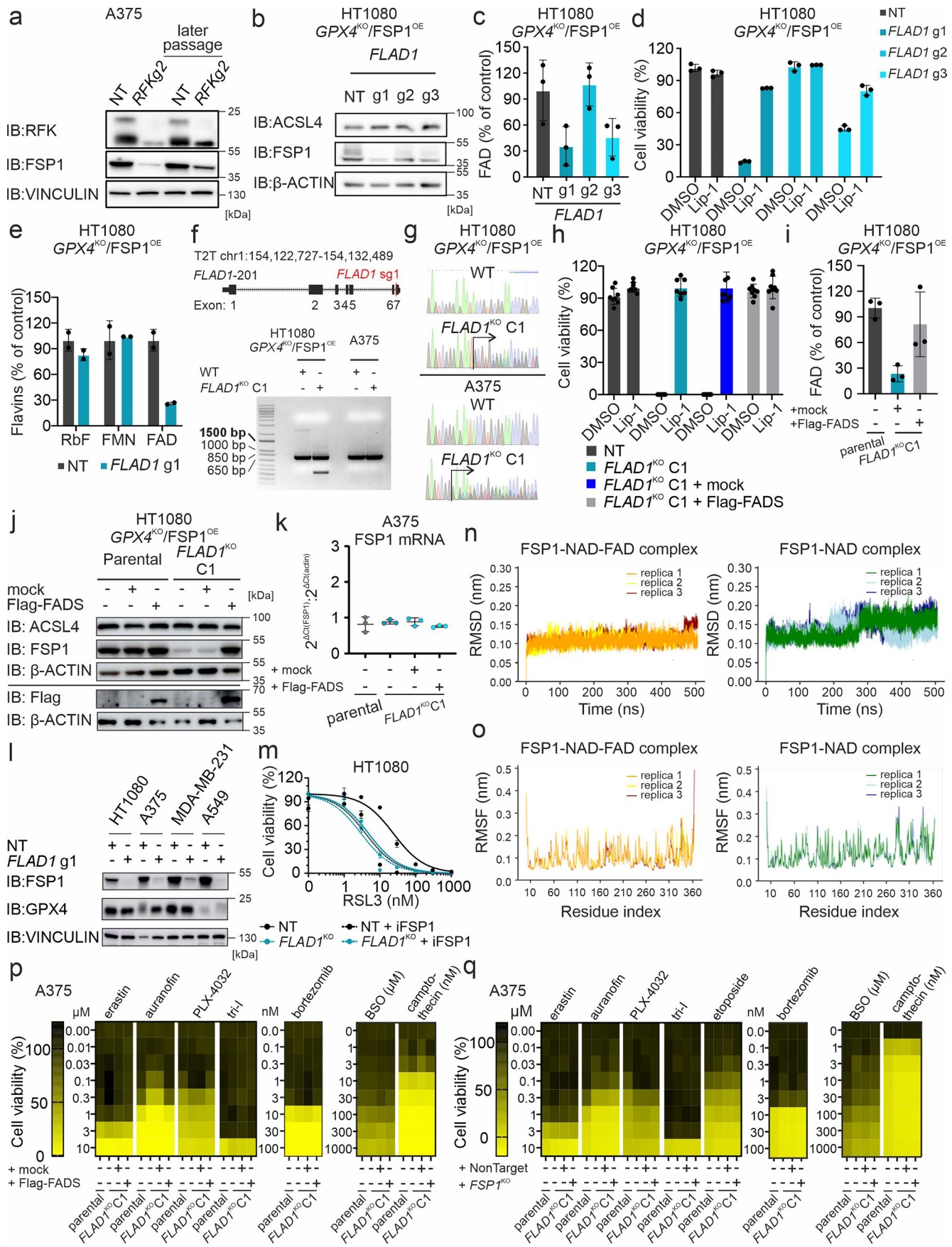

**Extended Data Fig. 2 | See next page for caption.**

**Extended Data Fig. 2 | FAD deficiency disrupts FSP1 function and promotes ferroptosis susceptibility. a**, Immunoblot (IB) analysis of RFK, FSP1 and vinculin in A375 cells transduced with either a non-targeting (NT) or *RFK*-targeting sgRNAs collected at an early passage or after two weeks of passaging (later passage). The experiment was performed once. **b**, Immunoblot (IB) analysis of FSP1, ACSL4 and β-actin in HT1080 *GPX4*^KO^/FSP1^OE^ cells transduced with either a non-targeting control (NT) or three different *FLAD1*-targeting sgRNAs. The experiment was performed once. **c**, Relative quantification of FAD in HT1080 *GPX4*^KO^/FSP1^OE^ cells transduced with either a non-targeting control (NT) or three different *FLAD1*-targeting sgRNAs. Data plotted are mean ± SD of triplicates from one independent experiment. **d**, Cell viability of HT1080 *GPX4*^KO^/FSP1^OE^ cells transduced with either a non-targeting control (NT) or three different *FLAD1*-targeting sgRNAs in the absence or presence of Lip-1 (500 nM). Cell viability was monitored using Alamar blue after 96 h. Data plotted are mean ± SD of triplicates from one representative of two independent experiments. **e**, Relative quantification of riboflavin (RoF), flavin mononucleotide (FMN) and flavin adenine dinucleotide (FAD) in HT1080 *GPX4*^KO^/FSP1^OE^ cells transduced with either a non-targeting control (NT) or an *FLAD1*-targeting sgRNAs (FLAD1 sgRNA1). Data plotted are mean ± SD of triplicates from one independent experiment. **f**, Genotyping of wild-type (WT) and *FLAD1*^KO^ cells. The scheme shows the *FLAD1* gene, indicating exons and the target site of *FLAD1* sgRNA1 (red) in exon 7. The PCR products spanning the *FLAD1* sgRNA1 cut site in WT and FLAD1^KO^ (single clone, C1) from HT1080 *GPX4*^KO^/FSP1^OE^ and A375 cells were resolved on a 1% agarose gel. The experiment was performed once. **g**, Sanger sequencing chromatograms of PCR products confirmed the presence of wild-type (WT) and mutant alleles in the *FLAD1* locus in HT1080 *GPX4*^KO^/FSP1^OE^ and A375 cells. The experiment was performed once. **h**, Cell viability of HT1080 *GPX4*^KO^/FSP1^OE^ non-target (NT), *FLAD1*^KO^ single clone 1 (C1), and *FLAD1*^KO^ C1 cells stably overexpressing either an empty vector (mock) or Flag-FADS (addback) in the absence or presence of Lip-1 (500 nM). Cell viability was monitored after 24 h. Data plotted are mean ± SD of triplicates from one representative of two

independent experiments. **i**, Relative quantification of FAD in HT1080 *GPX4*^KO^/FSP1^OE^ cells "parental", *FLAD1*^KO^ C1 and *FLAD1*^KO^ C1 cells stably overexpressing either an empty vector (mock) or Flag-FADS (addback). Data plotted are mean ± SD of triplicates from one independent experiment. **j**, Immunoblot (IB) analysis of ACSL4, FSP1, Flag-tag and β-actin in HT1080 *GPX4*^KO^/FSP1^OE^ "parental" and *FLAD1*^KO^ C1 cells stably overexpressing an empty vector (mock) or Flag-FADS (addback). The experiment was performed once. **k**, Relative mRNA levels of FSP1 measured by quantitative RT-PCR in A375 parental, *FLAD1*^KO^ single clone 1 (C1) and *FLAD1*^KO^ C1 overexpressing either an empty vector (mock) or Flag-FADS (addback). Data plotted are mean ± SD of triplicates from one independent experiment. **l**, Immunoblot (IB) analysis of FSP1, GPX4 and vinculin in HT1080, A375, MDA-MB-231 and A549 cells transduced with either a non-targeting (NT) or *FLAD1*-targeting sgRNA (*FLAD1* sgRNA1). The experiment was performed once. **m**, Dose-dependent toxicity of RSL3 in the absence or presence of iFSP1 (3 μM) in HT1080 cells transduced with either a non-targeting (NT) or *FLAD1*-targeting sgRNA (*FLAD1* sgRNA1). Cell viability was monitored using Alamar blue after 96 h. Data plotted are mean ± SD of triplicates from one representative of two independent experiments. **n**, Root mean square deviation (RMSD) of FSP1 backbone in FSP1-NAD-FAD and FSP1-NAD complexes for three 500 ns-MD simulations. **o**, Root mean square fluctuation (RMSF) of FSP1 residues in FSP1-NAD-FAD and FSP1-NAD complexes. The absence of FAD leads to more instability in the protein structure. **p**, Dose-dependent toxicity of the indicated cytotoxic compounds in A375 parental, A375 *FLAD1*^KO^ single clone 1 (C1) and A375 *FLAD1*^KO^ C1 cells stably overexpressing either an empty vector (mock) or Flag-FADS (addback). Cell viability was monitored using Alamar blue after 72 h. Data plotted are mean of triplicates from one representative of two independent experiments. **q**, Dose-dependent toxicity of the indicated cytotoxic compounds in A375 parental, A375 *FLAD1*^KO^ single clone (C1) and A375 *FLAD1*^KO^ C1 transduced with either a non-targeting control (NT) or a FSP1-targeting sgRNAs (*FSP1*^KO^). Cell viability was monitored using Alamar blue after 72 h. Data plotted are mean of triplicates from one representative of two independent experiments.

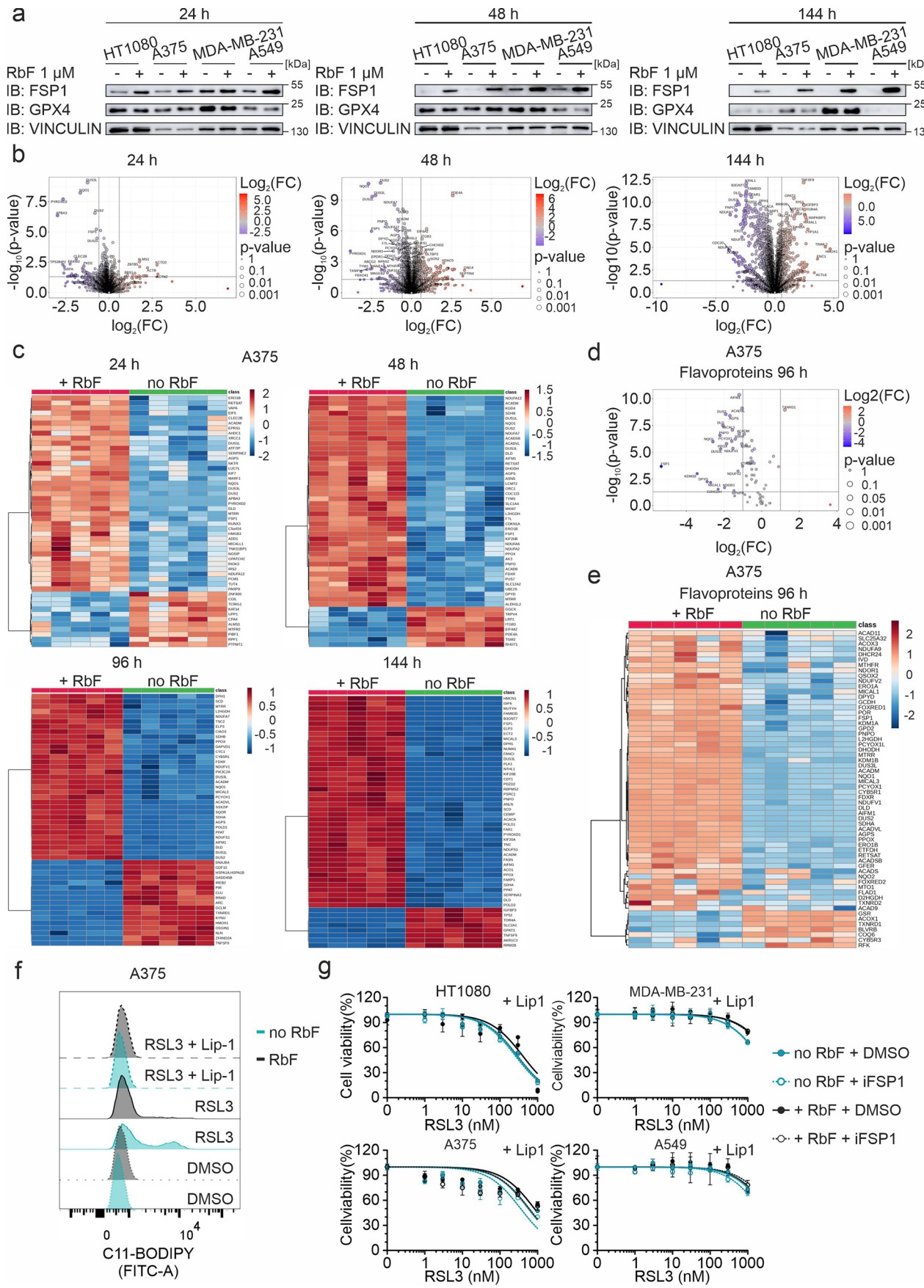

Extended Data Fig. 3 | See next page for caption.

**Extended Data Fig. 3 | Riboflavin availability as a central determinant of ferroptosis resistance. a**, Immunoblot (IB) analysis of FSP1, GPX4 and vinculin in HT1080, A375, MDA-MB-231 and 549 parental cell lines after 24, 48 and 144 h of growth in riboflavin-deficient medium or supplemented with 1 µM of riboflavin. Data shown is one representative of two independent experiments. **b**, Volcano plots of quantified proteins showing their change in A375 parental cells cultured in riboflavin-deficient medium for 24, 48 and 144 h. Proteins are plotted based on their fold change (FC: riboflavin deficient/normal). Statistical significance was assessed by unpaired two-sided t-test, and values were deemed significant if $p < 0.05$, and minimum fold change: 1.5. The statistical significance of the respective ratios is plotted on the y-axis (n = 5 technical replicates). **c**, Heatmaps of quantified proteins showing their change in A375 parental cells cultured in riboflavin-deficient medium for 24, 48, 96 and 144 h (FC: riboflavin deficient/ normal). Statistical significance was assessed by unpaired two-sided t-test, and values were deemed significant if $p < 0.05$, and minimum fold change: 1.5. The heatmap shows five technical replicates from one independent experiment. **d**, Volcano plot of quantified flavoproteins showing their change in A375 parental cells cultured in riboflavin-deficient medium for 96 h (FC: riboflavin deficient/normal). Statistical significance was assessed by unpaired two-sided t-test, and values were deemed significant if $p < 0.05$, and minimum fold change: 1.5. The statistical significance of the respective ratios is plotted on the y-axis (n = 5 technical replicates). **e**, Heatmap of quantified flavoproteins showing their change in A375 parental cells cultured in riboflavin-deficient medium for 96 h (FC: riboflavin deficient/normal). The heatmap shows five technical replicates from one independent experiment. Statistical significance was assessed by unpaired two-sided t-test, and values were deemed significant if $p < 0.05$, and minimum fold change: 1.5. **f**, Lipid peroxidation evaluated by C11-BODIPY 581/591 staining of A375 parental cell line cultured for 72 h in riboflavin-deficient medium or supplemented with 1 µM of riboflavin and after treatment with DMSO, RSL3 (200 nM) or RSL3 (200 nM) + Lip-1 (500 nM) for 6 h. Representative plots of one of two independent experiments. **g**, Dose-dependent toxicity of RSL3 in the presence of Lip-1 (500 nM) and iFSP1 (3 µM, when indicated) in HT1080, A375, MDA-MB-231 and A549 parental cell lines cultured in riboflavin-deficient medium or supplemented with 1 µM of riboflavin for 48 h. Cell viability was monitored using Alamar blue after 96 h of treatment. Data plotted are mean ± SD of triplicates from one representative of three independent experiments.

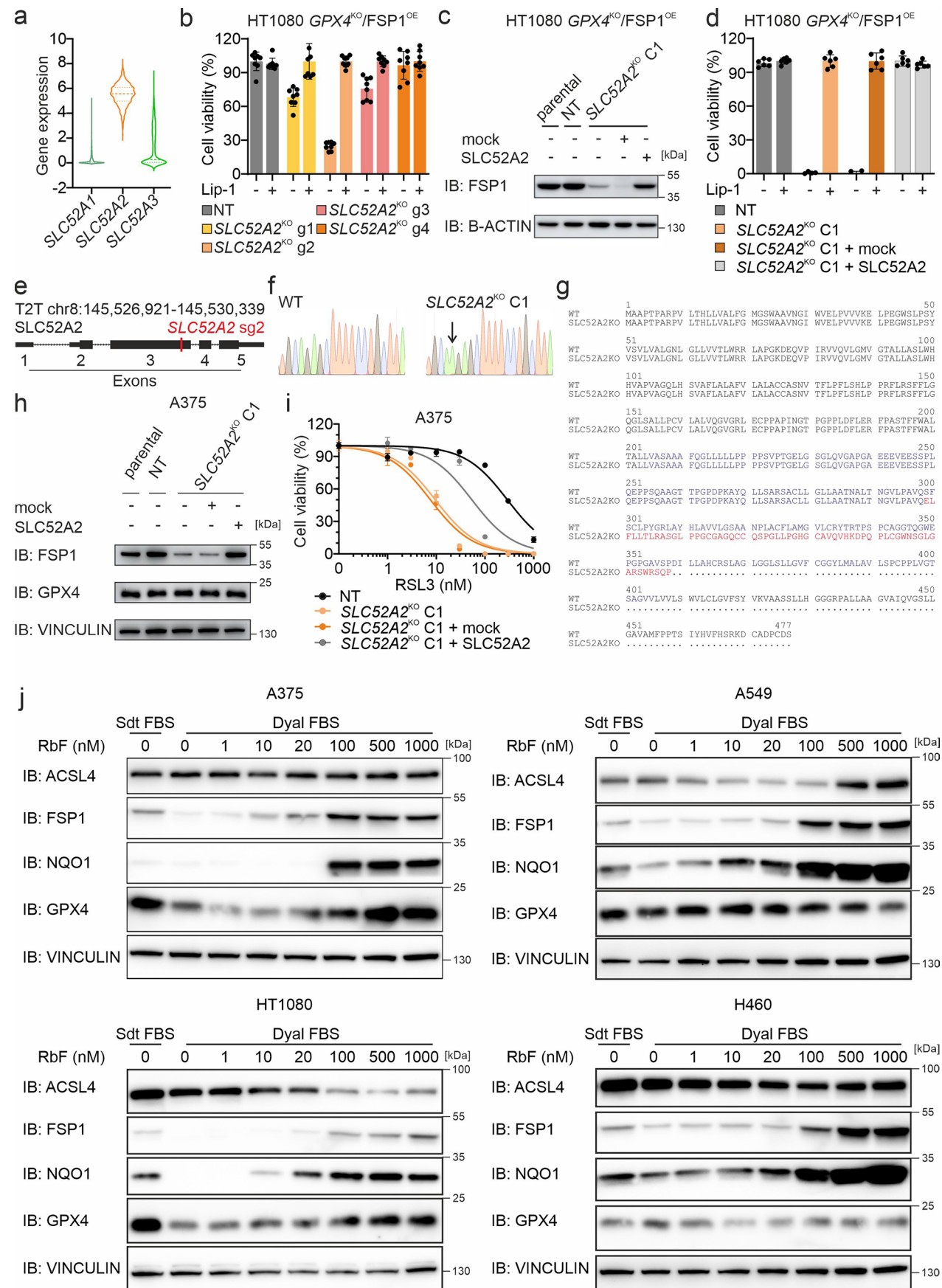

**Extended Data Fig. 4 | See next page for caption.**

**Extended Data Fig. 4 | Disruption of the riboflavin transporter SLC52A2 induces ferroptosis. a**, Transcript expression levels of the riboflavin transporters SLC52A1, SLC52A2 and SLC52A3 for a panel of human cancer cell lines from Dependency Map public 23Q2 dataset (https://depmap.org/portal/, version 23Q2). Data are presented as a violin plot. The median, as well as the upper and lower quartiles, are indicated by dashed lines. **b**, Cell viability of HT1080 $GPX4^{KO}$/FSP1$^{OE}$ cells transduced with either a non-targeting control (NT) or four different $SLC52A2$-targeting sgRNAs in the absence or presence of Lip-1 (500 nM). Cell viability was monitored using Alamar blue after 96 h. Data plotted are mean ± SD from one representative of two independent experiments. **c**, Immunoblot (IB) analysis of FSP1 and β-actin in HT1080 $GPX4^{KO}$/FSP1$^{OE}$ "parental", non-target control (NT), $SLC52A2^{KO}$ single clone 1 (C1) and $SLC52A2^{KO}$ C1 cells stably overexpressing an empty vector (mock) or SLC52A2 (addback). The data shown is one representative of two independent experiments. **d**, Cell viability of HT1080 $GPX4^{KO}$/FSP1$^{OE}$ cells non-target control (NT), $SLC52A2^{KO}$ single clone 1 (C1) and SLC52A2$^{KO}$ C1 cells stably overexpressing an empty vector (mock) or SLC52A2 (addback). Cell viability was monitored using Alamar blue after 96 h. Data plotted are mean ± SD from one representative of two independent experiments. **e**, Genotyping of wild-type (WT) and $SLC52A2^{KO}$ cells. The scheme shows the

$SLC52A2$ gene, indicating exons and the target site of $SLC52A2$ sgRNA2 (red) in exon 3. **f**, Sanger sequencing chromatograms of PCR products confirmed the presence of WT and mutant alleles in the $SLC52A2$ locus in HT1080 $GPX4^{KO}$/FSP1$^{OE}$ and A375 cells. **g**, Schematic representation of the sequencing results obtained from the PCR product (in blue) covering the edited region (in red) in comparison with the wild-type product. **h**, Immunoblot analysis (IB) of FSP1, GPX4 and vinculin in A375 parental, non-target control (NT), $SLC52A2^{KO}$ single clone 1 (C1) and $SLC52A2^{KO}$ single clone 1 (C1) stably overexpressing an empty vector (mock) or SLC52A2 (addback). Data shown is one representative of two independent experiments. **i**, Dose-dependent toxicity of RSL3 in A375 non-target control (NT), $SLC52A2^{KO}$ single clone 1 (C1) and $SLC52A2^{KO}$ C1 stably overexpressing an empty vector (mock) or SLC52A2 (addback). Cell viability was monitored using Alamar blue after 96 h. Data plotted are mean ± SD of triplicates from one representative of two independent experiments. **j**, Immunoblot analysis (IB) of ACSL4, FSP1, NQO1, GPX4 and vinculin in A375, A549, HT1080 and H460 parental cell lines cultured in medium supplemented with either standard FBS or dialysed FBS and different concentrations of riboflavin (RbF). Data shown is one representative of two independent experiments.

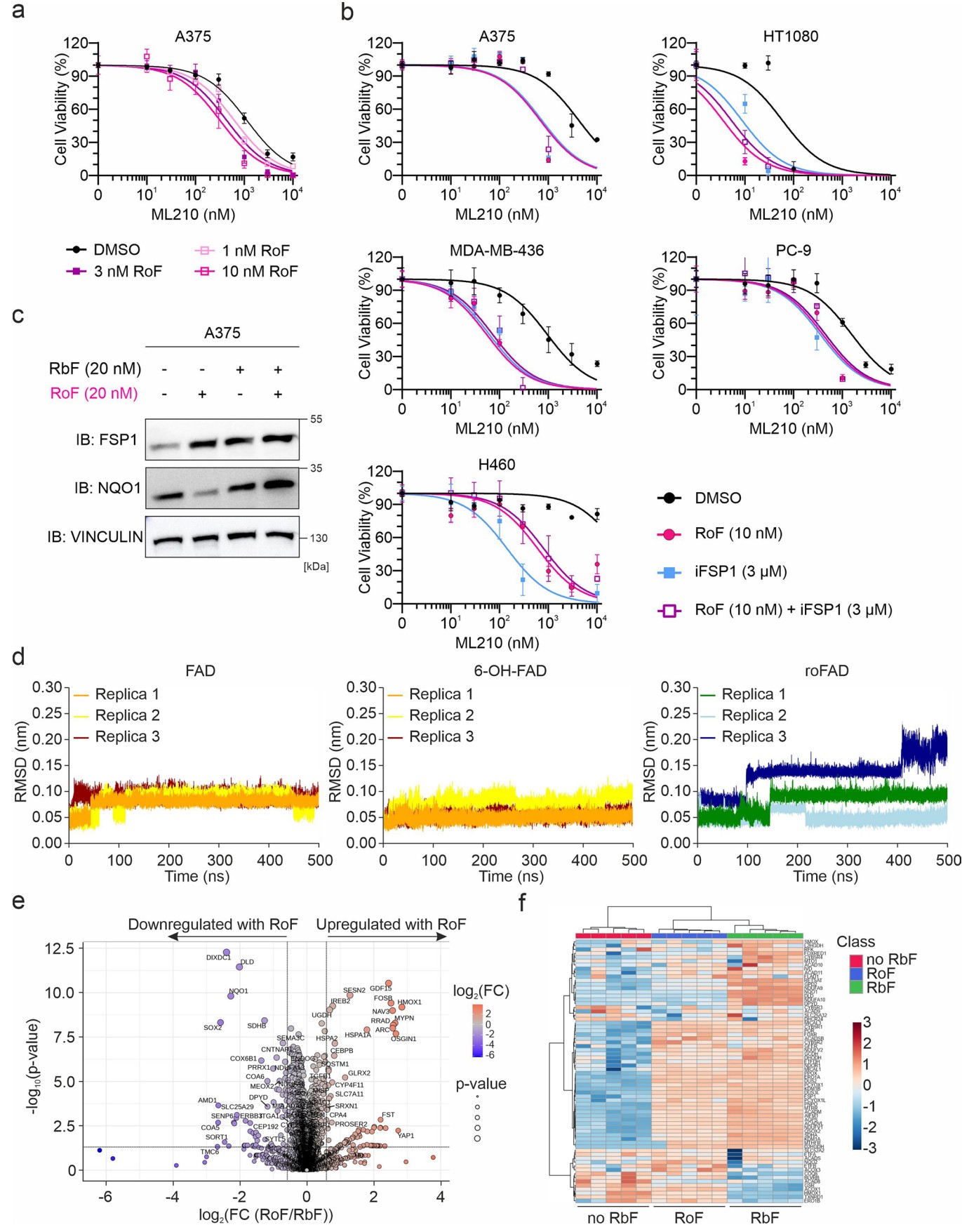

**Extended Data Fig. 5 | See next page for caption.**

**Extended Data Fig. 5 | The riboflavin analogue roseoflavin disrupts FSP1 activity and promotes ferroptosis. a**, Dose-dependent toxicity of ML210 in A375 parental cell line pre-treated with roseoflavin (RoF, 0, 1, 3 and 10 nM) for 48 h. Cell viability was monitored using Alamar blue after 48 h of treatment with ML210. Cells were cultured in low-riboflavin medium (20 nM). Data plotted are mean ± SD of triplicates from one representative of two independent experiments. **b**, Dose-dependent toxicity of ML210 in A375, HT1080, MDA-MB-436, PC-9 and H460 parental cell lines pre-treated with roseoflavin (10 nM) and/or iFSP1 (3 μM) for 48 h. Cell viability was monitored after 48 h of treatment. Cells were cultured in low-riboflavin medium (20 nM). Data plotted are mean ± SD of triplicates from one representative of two independent experiments. **c**, Immunoblot (IB) analysis of FSP1, NQO1 and vinculin in A375 parental cell line treated with roseoflavin (RoF, 20 nM) for 96 h in the absence or presence of riboflavin (RbF 20 nM). Data shown is one representative of two independent experiments. **d**, Root mean square deviation (RMSD) of FSP1 backbone in FSP1-NAD-FAD, FSP1-NAD-6-OH-FAD and FSP1-NAD-roFAD complexes for three 500 ns-MD simulations. **e**, Volcano plots of quantified proteins showing their change in A375 parental cells that were re-feeded with roseoflavin or riboflavin (100 nM) followed 96 hours of riboflavin deprivation. Proteins are plotted based on their fold change (FC: roseoflavin/riboflavin). Statistical significance was assessed by unpaired two-sided t-test, and values were deemed significant if p < 0.05, and minimum fold change: 1.5. The statistical significance of the respective ratios is plotted on the y-axis (n = 5 technical replicates). **f**, Heatmaps of quantified proteins showing their change in A375 parental cells that were re-feeded with roseoflavin or riboflavin (100 nM) followed 96 hours of riboflavin deprivation. Statistical significance was assessed by unpaired two-sided t-test, and values were deemed significant if p < 0.05, and minimum fold change: 1.5. The heatmap shows five technical replicates from one independent experiment.

# Reporting Summary

## Statistics

For all statistical analyses, confirm that the following items are present in the figure legend, table legend, main text, or Methods section.

| n/a | Confirmed | |
|---|---|---|
| ☐ | ☒ | The exact sample size (*n*) for each experimental group/condition, given as a discrete number and unit of measurement |
| ☐ | ☒ | A statement on whether measurements were taken from distinct samples or whether the same sample was measured repeatedly |
| ☐ | ☒ | The statistical test(s) used AND whether they are one- or two-sided<br>*Only common tests should be described solely by name; describe more complex techniques in the Methods section.* |
| ☒ | ☐ | A description of all covariates tested |
| ☐ | ☒ | A description of any assumptions or corrections, such as tests of normality and adjustment for multiple comparisons |
| ☐ | ☒ | A full description of the statistical parameters including central tendency (e.g. means) or other basic estimates (e.g. regression coefficient) AND variation (e.g. standard deviation) or associated estimates of uncertainty (e.g. confidence intervals) |
| ☐ | ☒ | For null hypothesis testing, the test statistic (e.g. *F*, *t*, *r*) with confidence intervals, effect sizes, degrees of freedom and *P* value noted<br>*Give P values as exact values whenever suitable.* |
| ☒ | ☐ | For Bayesian analysis, information on the choice of priors and Markov chain Monte Carlo settings |
| ☒ | ☐ | For hierarchical and complex designs, identification of the appropriate level for tests and full reporting of outcomes |
| ☒ | ☐ | Estimates of effect sizes (e.g. Cohen's *d*, Pearson's *r*), indicating how they were calculated |

*Our web collection on statistics for biologists contains articles on many of the points above.*

## Software and code

Policy information about availability of computer code

| Data collection | Spark Control v3.2, BD FACSDiva Software v6.1.3, Amersham ImageQuant 800 v2.0.0, SnapGene v8.0.1, Incucyte v2023A Rev2, TraceFinder v5.1, Realplex v2.2.2, CFX Maestro v2.0, Schrödinger v.2024.2, AmberTools23, Gaussian 09, GROMACS v2024.2 and NanoDrop One v2.8.0.25 |
|---|---|
| Data analysis | GraphPad Prism v10.4.1, Excel v24.11, MAGeCK v0.5.9, R v4.2.0, RStudio v2022.02.0, Python v3.10.2, LPPtiger2.0, Skyline v24.1.0.199, MetaboAnalyst v6.0 (online platform: https://www.metaboanalyst.ca), Spectronaut v19.1.240724.62635, ImageJ 1.54m and FlowJo v10.10.0 |

For manuscripts utilizing custom algorithms or software that are central to the research but not yet described in published literature, software must be made available to editors and reviewers. We strongly encourage code deposition in a community repository (e.g. GitHub). See the Nature Portfolio guidelines for submitting code & software for further information.

## Data

Policy information about availability of data

All manuscripts must include a data availability statement. This statement should provide the following information, where applicable:

- Accession codes, unique identifiers, or web links for publicly available datasets
- A description of any restrictions on data availability
- For clinical datasets or third party data, please ensure that the statement adheres to our policy

Mass spectrometry data have been deposited in ProteomeXchange with the primary accession code PXD061038 (https://www.ebi.ac.uk/pride/archive/projects/PXD061038). All other data supporting the findings of this study are available from the corresponding author on reasonable request.

# Research involving human participants, their data, or biological material

Policy information about studies with human participants or human data. See also policy information about sex, gender (identity/presentation), and sexual orientation and race, ethnicity and racism.

| | |
|---|---|
| Reporting on sex and gender | *Use the terms sex (biological attribute) and gender (shaped by social and cultural circumstances) carefully in order to avoid confusing both terms. Indicate if findings apply to only one sex or gender; describe whether sex and gender were considered in study design; whether sex and/or gender was determined based on self-reporting or assigned and methods used. Provide in the source data disaggregated sex and gender data, where this information has been collected, and if consent has been obtained for sharing of individual-level data; provide overall numbers in this Reporting Summary. Please state if this information has not been collected. Report sex- and gender-based analyses where performed, justify reasons for lack of sex- and gender-based analysis.* |
| Reporting on race, ethnicity, or other socially relevant groupings | *Please specify the socially constructed or socially relevant categorization variable(s) used in your manuscript and explain why they were used. Please note that such variables should not be used as proxies for other socially constructed/relevant variables (for example, race or ethnicity should not be used as a proxy for socioeconomic status). Provide clear definitions of the relevant terms used, how they were provided (by the participants/respondents, the researchers, or third parties), and the method(s) used to classify people into the different categories (e.g. self-report, census or administrative data, social media data, etc.) Please provide details about how you controlled for confounding variables in your analyses.* |
| Population characteristics | *Describe the covariate-relevant population characteristics of the human research participants (e.g. age, genotypic information, past and current diagnosis and treatment categories). If you filled out the behavioural & social sciences study design questions and have nothing to add here, write "See above."* |
| Recruitment | *Describe how participants were recruited. Outline any potential self-selection bias or other biases that may be present and how these are likely to impact results.* |
| Ethics oversight | *Identify the organization(s) that approved the study protocol.* |

Note that full information on the approval of the study protocol must also be provided in the manuscript.

# Field-specific reporting

Please select the one below that is the best fit for your research. If you are not sure, read the appropriate sections before making your selection.

☒ Life sciences      ☐ Behavioural & social sciences      ☐ Ecological, evolutionary & environmental sciences

For a reference copy of the document with all sections, see nature.com/documents/nr-reporting-summary-flat.pdf

# Life sciences study design

All studies must disclose on these points even when the disclosure is negative.

| | |
|---|---|
| Sample size | No samples size calculation was performed. Preliminary cell viability experiments showed small variations between biological replicates, so we chose n = 3 for reproducibility. |
| Data exclusions | In very rare cases single values of biological triplicates were excluded from the analysis due to te cell clumps/uneven plating. |
| Replication | Most attempts to replicate experiments were successful, demonstrating the robustness of the results. To ensure reliable replication, all sera used were pretested for their suitability in ferroptosis research. It is well-established that variations in vitamin content, lipid composition, and selenium concentrations between serum batches profoundly impact on the outcome of ferroptosis inducing/inhibiting conditions. |
| Randomization | All sample handling for proteomics analysis (sample preparation and LC-MS/MS analysis) have been performed in a block randomization order. |
| Blinding | Blinding was not performed in this study as the experimental design required direct observation of cell phenotypes and treatment conditions were distinguishable. However, data analysis was conducted using objective, quantifiable measurements to minimize bias. |

# Reporting for specific materials, systems and methods

We require information from authors about some types of materials, experimental systems and methods used in many studies. Here, indicate whether each material, system or method listed is relevant to your study. If you are not sure if a list item applies to your research, read the appropriate section before selecting a response.

## Materials & experimental systems

| n/a | Involved in the study |
|---|---|
| ☐ | ☒ Antibodies |
| ☐ | ☒ Eukaryotic cell lines |
| ☒ | ☐ Palaeontology and archaeology |
| ☒ | ☐ Animals and other organisms |
| ☒ | ☐ Clinical data |
| ☒ | ☐ Dual use research of concern |
| ☒ | ☐ Plants |

## Methods

| n/a | Involved in the study |
|---|---|
| ☒ | ☐ ChIP-seq |
| ☐ | ☒ Flow cytometry |
| ☒ | ☐ MRI-based neuroimaging |

## Antibodies

| | |
|---|---|
| Antibodies used | GPX4 (Abcam, cat. no. ab125066 or Proteintech, cat. no. 67763-1-Ig), ACSL4 (Santa Cruz, cat. no. sc-271800), β-actin (Sigma-Aldrich, cat. no. A5441), vinculin (Santa Cruz, sc-73614), Flag-tag (Sigma-Aldrich, cat. no. F3165), FADS (Santa Cruz, cat. no. sc-376819), RFK (Santa Cruz, cat. no. sc-398830 or Abbexa cat. no. abx124688) and NQO1 (Santa Cruz, cat. no. sc-32793). The FSP1 antibody was developed in Helmholtz Zentrum München against recombinant human FSP1 protein (clone 6D8-11). |
| Validation | Antibodies against GPX4 (Abcam, cat. no. ab125066) were validated for Western blotting in a previous publication (PMID: 25402683). The GPX4 antibody from Proteintech (cat. no. 67763-1-Ig) was validated by the manufacturer using knockdown models (www.ptglab.com) and in our lab using knockout models. <br> The ACSL4 antibody (Santa Cruz, cat. no. sc-271800) was validated for Western blotting in a prior publication (PMID: 27842070). <br> The β-actin antibody (Sigma, cat. no. A5441) was validated as a loading control for Western blotting in a previous study (PMID: 15809369). <br> The vinculin antibody (Santa Cruz, cat. no. sc-73614) was validated as a loading control for Western blotting in another publication (PMID: 1478968). <br> The Flag-tag antibody (Sigma-Aldrich, cat. no. F3165) was validated in this study for Western Blotting, where we observed an increased signal in cells overexpressing Flag-FLAD1 compared to wild-type controls. <br> The FADS (Santa Cruz, cat. no. sc-376819), RFK (Santa Cruz, cat. no. sc-398830 or Abbexa, cat. no. abx124688), and NQO1 (Santa Cruz, cat. no. sc-32793) antibodies have not been fully validated according to all recommended criteria for Western blotting (e.g., validation with knockout models). For the FADS antibody, our Western blot analysis showed increased signal intensity in cells overexpressing FADS, suggesting specificity. For the RFK antibody, a clear reduction in RFK signal was observed in cells transduced with three RFK-targeting sgRNAs compared to wild-type controls, with the detected RFK band corresponding to the reported molecular weight, indicating specificity. For the NQO1 antibody, our Western blot analysis revealed a band at the expected molecular weight, and the observed expression pattern was consistent with proteomics data obtained under the same experimental conditions, as well as with previous publications (e.g., PMID: 32895367), supporting its specificity. However, these antibodies also showed non-specific bands at varying molecular weights under the tested conditions. <br> The FSP1 antibody was validated for Western blotting in a previous publication (PMID: 31634899). |

## Eukaryotic cell lines

Policy information about cell lines and Sex and Gender in Research

| | |
|---|---|
| Cell line source(s) | Human cancer cell lines HT1080, A375, MDA-MB-231, MDA-MB-436, A549 and H460 were purchased from ATCC (cat. no. CCL-121, CRL-1619, CRM-HTB-26, HTB-130, CRM-CCL-185 and HTB-177, respectively) and PC9 was purchased from Sigma (cat. no. 90071810) |
| Authentication | None of the cell lines used in this study were authenticated. |
| Mycoplasma contamination | Cells are tested at least once a year for Mycoplasm contamination by qPCR at Eurofins Genomics. |
| Commonly misidentified lines (See ICLAC register) | N/A |

## Plants

| | |
|---|---|
| Seed stocks | N/A |
| Novel plant genotypes | N/A |
| Authentication | N/A |

# Flow Cytometry

## Plots

Confirm that:

☒ The axis labels state the marker and fluorochrome used (e.g. CD4-FITC).

☒ The axis scales are clearly visible. Include numbers along axes only for bottom left plot of group (a 'group' is an analysis of identical markers).

☐ All plots are contour plots with outliers or pseudocolor plots.

☐ A numerical value for number of cells or percentage (with statistics) is provided.

## Methodology

| | |
|---|---|
| Sample preparation | 50,000 cells per well were seeded on 6-well plates one day prior to the experiment. The next day, cells were treated with the following conditions (i) DMSO, (ii) RSL3 (200 nM) and (iii) RSL3 (200 nM) + Lip-1 (500 nM) for 6 hours. After treatment, cells were washed with PBS and incubated with C11-BODIPY (1 µM) for 30 minutes at 37 °C before they were harvested by trypsinization. Subsequently, cells were resuspended in 400 µL of PBS supplemented with 2% FBS followed by analysis using a flow cytometer (FACS Canto II, BD Biosciences). For riboflavin deprivation experiments, cells were pre-incubated in riboflavin-deficient medium or medium supplemented with riboflavin (1 µM) for 72 hours before undergoing the treatments described above. |
| Instrument | FACS Canto II (BD Biosciences). Data was collected from the FITC detector (for the oxidized form of BODIPY) with a 502LP and 530/30 BP filter and from the PE detector (for the reduced form of BODIPY) with 556 LP and 585/42 BP filter. At least 10,000 events were analyzed per sample. |
| Software | For data collection the BD FACSDiva Software v6.1.3 was used. <br> For data analysis FlowJo v10.10.0 software was used. |
| Cell population abundance | The abundance of the desired cell population in post-sort fractions was generally > 85% of the total post-sort population. The ratio FITC/PE (oxidized/reduced ratio) was calculated as follows: (median FITC-A fluorescence−median FITC-A fluorescence of unstained samples)/(median PE-A fluorescence−median PE-A fluorescence of unstained samples). |
| Gating strategy | Live cell populations were separated from cellular debris and dead cells using SSC-A vs. FSC-A gating strategy. |

☐ Tick this box to confirm that a figure exemplifying the gating strategy is provided in the Supplementary Information.

