## [Peer Review File · Nature Cell Biology]

Riboflavin metabolism shapes FSP1-driven ferroptosis resistance

Corresponding Author: Professor José Pedro Friedmann Angeli

Version 0:

Decision Letter:

*Please delete the link to your author homepage if you wish to forward this email to co-authors.

Dear Professor Friedmann Angeli,

Your manuscript, "Riboflavin metabolism shapes FSP1-driven ferroptosis resistance", has now been seen by 3 referees, who are experts in ferroptosis (referee 1); ferroptosis, metabolism (referee 2); and vitamin biology and metabolism (referee 3). As you will see from their comments (attached below) they find this work of potential interest, but have raised substantial concerns, which in our view would need to be addressed with considerable revisions before we can consider publication in Nature Cell Biology.

Nature Cell Biology editors discuss the referee reports in detail within the editorial team, including the chief editor, to identify key referee points that should be addressed with priority, and requests that are overruled as being beyond the scope of the current study. To guide the scope of the revisions, I have listed these points below. We are committed to providing a fair and constructive peer-review process, so please feel free to contact me if you would like to discuss any of the referee comments further.

In particular, it would be essential to:

A) Strengthen the proposed working mechanism as questioned by the reviewers:

Reviewer 1

"What is the mechanisms of action of roFAD? Does it have a much higher affinity to FSP1 than FAD (hence competition), and is roFAD not able to undergo 6-hydroxylation upon FSP1-binding and thus gives rise to an inactive FSP1? Answering these questions can modestly improve the novelty."

Reviewer 2

"The observation that FLAD1 knockout had no impact on non-GPX4 inhibitors (e.g., Erastin/BSO) suggests distinct mechanisms of Erastin/BSO and GPX4 inhibition. So, what is the different about these ferroptosis-inducer? Moreover, FLAD1 ablation impairs both FSP1 activity and stability, seemingly conflicting with prior reports that FSP1 overexpression inhibits Erastin-induced ferroptosis. This paradox warrants deeper mechanistic discussion. Additionally, does Rbf-FMN-FAD metabolic axis affect the level of CoQ?"

Reviewer 3

"...however I suggest a much simpler, but I believe important experiment that should clarify, if FAD leads to a general stabilisation, or if cellular stresses play a role in this process. The authors have shown clear evidence that the loss of, for example, FLAD1 leads to loss of FSP-1. Cells under FLAD1 loss will experience a significant amount of oxidative stress. Testing if by relieving this stress, for example by treating with Lip1 leads to a stabilisation of FSP-1, will give clearer insights into the type of stabilisation we are looking at. Is it a physical/mechanical one, or is it a protection against, for example, exaggerated protein oxidation followed by proteasomal turnover? The latter could also be tested with a proteasomal inhibitor."

"...On that note, having established the system with the SCD1 inhibitor, it could be a quick and easy additional experiment to check for FSP-1 stability under SCD1 inhibition."

B) Clarify the rationale for the screening strategy as raised by Reviewer 2:

"The focused screening approach demonstrates ingenuity in enhancing threshold sensitivity for FSP1 regulators. Intriguingly, SCD1—a known ferroptosis regulator—emerged prominently in the screening targeting FSP1. Should other critical ferroptosis resistance factors exist in the library but remain undetected, investigating potential FSP1-dependent specificity of SCD1's regulatory role would merit future exploration.

Some questions:

- Why focus on a drug-targeted library rather than a genome-wide screening?
- Why are there no corresponding drugs for the RTKs identified in the drug-targeted library?
- Why were other enzymes in the FAD pathway not identified? Is it because this library does not include these genes?"

C) Address the potential discrepancies as raised by the reviewers:

Reviewer 2

"Would the lower physiological Rbf concentrations compared to cell culture conditions imply attenuated FSP1 functionality in vivo theoretically? This highlights potential discrepancies between in vitro models (influenced by selenium, cysteine, transferrin, VK/VE, etc.)

and physiological contexts—a critical consideration for translational research."

"In fig2k, while FLAD1 single KO and FLAD1/FSP1 double KO show comparable lipid ROS levels, FSP1 single KO exhibits markedly elevated ROS. This disparity may need further validation to confirm."

D) All other referee concerns pertaining to strengthening existing data, providing controls, methodological details, clarifications and textual changes if applicable should also be addressed.

E) Finally please pay close attention to our guidelines on statistical and methodological reporting (listed below) as failure to do so may delay the reconsideration of the revised manuscript. In particular please provide:

We would be happy to consider a revised manuscript that would satisfactorily address these points, unless a similar paper is published elsewhere, or is accepted for publication in Nature Cell Biology in the meantime.

- ensure that it conforms to our format instructions and publication policies (see below and www.nature.com/nature/authors/).

- provide a point-by-point rebuttal to the full referee reports verbatim, as provided at the end of this letter.

- provide the completed Editorial Policy Checklist (found here <https://www.nature.com/authors/policies/Policy.pdf>), and Reporting Summary (found here <https://www.nature.com/authors/policies/ReportingSummary.pdf>). This is essential for reconsideration of the manuscript and these documents will be available to editors and referees in the event of peer review. For more information see <http://www.nature.com/authors/policies/availability.html> or contact me.

Nature Cell Biology is committed to improving transparency in authorship. As part of our efforts in this direction, we are now requesting that all authors identified as 'corresponding author' on published papers create and link their Open Researcher and Contributor Identifier (ORCID) with their account on the Manuscript Tracking System (MTS), prior to acceptance. ORCID helps the scientific community achieve unambiguous attribution of all scholarly contributions. You can create and link your ORCID from the home page of the MTS by clicking on 'Modify my Springer Nature account'. For more information please visit <http://www.springernature.com/orcid>.

Link Redacted

We would like to receive a revised submission within six months. We would be happy to consider a revision even after this timeframe, however if the resubmission deadline is missed and the paper is eventually published, the submission date will be the date when the revised manuscript was received.

We hope that you will find our referees' comments, and editorial guidance helpful. Please do not hesitate to contact me if there is anything you would like to discuss.

Best wishes,

Zhe Wang

Zhe Wang, PhD
Senior Editor
Nature Cell Biology

Tel: +44 (0) 207 843 4924
email: zhe.wang@nature.com

Reviewers' Comments:

Reviewer #1 (Remarks to the Author):

Summary:

This manuscript focuses on FSP1, also known as AMID and AIFM2, a crucial suppressor of the iron-dependent form of cell death known as ferroptosis. By using a cellular model in which cell survival relies on FSP1 overexpression, the authors conducted a CRISPR screening and identified the riboflavin kinase RFK as a required factor for the ferroptosis-suppressing activity of FSP1. Through a series of subsequent experiments, the authors confirmed the established knowledge that FAD is an essential cofactor of FSP1. They also discovered that an analog of FAD, roFAD, can abrogate the ferroptosis-inhibitory activity of FSP1, likely by competing with FAD for FSP1-binding.

Originality and significance:

While technically this is overall an excellent study, the main conclusion of the study is unfortunately not new, as it has already been documented that FAD metabolites, particularly, 6-hydroxy-FAD (FAD binds to FSP1/AMID and is then converted to 6-hydroxy-FAD within the protein), is a bona fide cofactor of FSP1. This knowledge has been annotated in the UniProt database (<https://www.uniprot.org/uniprotkb/Q9BRQ8/entry>) and was reported long before the field of ferroptosis (PMID: 15958387). So, in essence the current manuscript confirmed these early discoveries but only by using another functional analysis of FSP1, inhibition of ferroptosis. With that said, a novel aspect of the paper is the discovery of roFAD as a potent inhibitor of FSP1, likely through competing with FAD for FSP1 binding. If the authors can extend this aspect by clearly defining the underlying mechanisms and exploring the therapeutic potential, the significance of the study can be improved. The authors are recommended to conduct additional experiments to address following questions:

1. What is the mechanisms of action of roFAD? Does it have a much higher affinity to FSP1 than FAD (hence competition), and is roFAD not able to undergo 6-hydroxylation upon FSP1-binding and thus gives rise to an inactive FSP1? Answering these questions can modestly improve the novelty.
2. In terms of significance: is roFAD a potential therapeutic agent in xenograft models, and how is it behaving in comparison with other "conventional" FSP1 inhibitors (potency, specificity, toxicity)?

Reviewer #2 (Remarks to the Author):

This study systematically investigates the upstream regulatory mechanisms of FSP1 and elucidates the specific pathway through which riboflavin (VB2) metabolism inhibits ferroptosis via non-RTA activity. Through an innovative FSP1-dependent focused pool screening, the authors first identified RTK as a modulator of FSP1-dependent ferroptosis. Further exploration revealed that the endogenous riboflavin-FMN-FAD metabolic axis suppresses ferroptosis in an FSP1-dependent manner. Mechanistically, this axis regulates both the enzymatic activity and protein stability of FSP1. Notably, the authors discovered that Rof, a structural analog of Rbf, disrupts the ferroptosis-protective effects of the riboflavin-FMN-FAD axis, thereby proposing a novel targeting strategy for FSP1 modulation.

Collectively, this work presents conceptual advances supported by rigorous experimental design and robust datasets. In ferroptosis research, it uncovers a protein-level regulatory mechanism in FSP1's upstream pathway—analogue to the selenium metabolism governing GPX4—thereby expanding the understanding of how endogenous metabolites determine ferroptosis susceptibility. Additionally, it provides new insights into the antioxidant mechanisms of vitamin B2 through the lens of membrane phospholipid oxidation homeostasis.

Comments:

1. Screening Strategy

The focused screening approach demonstrates ingenuity in enhancing threshold sensitivity for FSP1 regulators. Intriguingly, SCD1—a known ferroptosis regulator—emerged prominently in the screening targeting FSP1. Should other critical ferroptosis resistance factors exist in the library but remain undetected, investigating potential FSP1-dependent specificity of SCD1's regulatory role would merit future exploration.

Some questions:

- Why focus on a drug-targeted library rather than a genome-wide screening?
- Why are there no corresponding drugs for the RTKs identified in the drug-targeted library?
- Why were other enzymes in the FAD pathway not identified? Is it because this library does not include these genes?

2. FLAD1 Knockout Phenotype

The observation that FLAD1 knockout had no impact on non-GPX4 inhibitors (e.g., Erastin/BSO) suggests distinct mechanisms of Erastin/BSO and GPX4 inhibition. So, what is the different about these ferroptosis-inducer? Moreover, FLAD1 ablation impairs both FSP1 activity and stability, seemingly conflicting with prior reports that FSP1 overexpression inhibits Erastin-induced ferroptosis. This paradox warrants deeper mechanistic discussion. Additionally, does Rbf-FMN-FAD metabolic axis affect the level of CoQ?

3. Physiological Relevance of Rbf Levels

Would the lower physiological Rbf concentrations compared to cell culture conditions imply attenuated FSP1 functionality in vivo theoretically? This highlights potential discrepancies between in vitro models (influenced by selenium, cysteine, transferrin, VK/VE, etc.) and physiological contexts—a critical consideration for translational research.

4. Lipid ROS Validation

In fig2k, while FLAD1 single KO and FLAD1/FSP1 double KO show comparable lipid ROS levels, FSP1 single KO exhibits markedly elevated ROS. This disparity may need further validation to confirm.

Minor Points:

1. Line 219: Evidence for FAD's regulation of FSP1 activity remains experimentally unsubstantiated.
2. ED4j: GPX4 upregulation and ACSL4 downregulation with Rbf dose escalation in HT1080/A375 require clarification.
3. Line 307: "Fig4e" likely references Fig4d.
4. ED Fig2b: Missing FLAD1 WB bands; rationale for sg2's lack of ferroptosis phenotype?
5. Cell viability assays (e.g., ED Fig1d): Statistical annotations (asterisks) absent.
6. ED Fig1C: Mismatch text descriptions.

Reviewer #3 (Remarks to the Author):

Dear editor,

I reviewed now the study entitled Riboflavin metabolism shapes FSP1-driven ferroptosis resistance by Skafar et al. and in the following paragraphs I will share my assessment.

The authors study the phenomenon of ferroptosis and the FSP-1 driven protection from it. In previous work they had shown that FSP-1 could rescue cells from GPX4 inhibition, and in this manuscript they unravel further mechanistic details. Namely, they deploy a Crispr-screen that uncovers the enzyme RFK, part of the FAD biosynthetic pathway, as a crucial player in mediating FSP-1 dependent ferroptosis resistance in a GPX4 deplete environment. This leads to the finding that Riboflavin, as a precursor of FAD, is crucial for FSP-1 activity, that FAD stabilises FSP-1 and that a Riboflavin analogue, Roseoflavin, can counteract this activity.

Overall the manuscript is novel and studies an as yet unknown mechanism. The data is strong and the impact of the findings are in line with the scope of Nature Cell Biology. I will now detail a few further considerations.

The main aspect that, in my opinion, needs to be addressed is the role of FAD in stabilising FSP-1. This is a strong claim, which seems reasonable and the RMSD analysis of the FSP1-NAD-FAD complex in Ex. Fig. 2 tries to address it by modelling. As a disclaimer, I am not able to judge this particular experiment, as it is outside my field of expertise, however I suggest a much simpler, but I believe important experiment that should clarify, if FAD leads to a general stabilisation, or if cellular stresses play a role in this process. The authors have shown clear evidence that the loss of, for example, FLAD1 leads to loss of FSP-1. Cells under FLAD1 loss will experience a significant amount of oxidative stress. Testing if by relieving this stress, for example by treating with Lip1 leads to a stabilisation of FSP-1, will give clearer insights into the type of stabilisation we are looking at. Is it a physical/mechanical one, or is it a protection against, for example, exaggerated protein oxidation followed by proteasomal turnover? The latter could also be tested with a proteasomal inhibitor.

This ties in with the second point. The authors find a second prominent hit SCD1, which they spent a little time on. Here they claim that the experiments they do show that 'SCD1 depletion did not appear to directly impair FSP-1 function (line 116)'. I am not sure that the data in Ex. Fig. 1a,b allow for that conclusion. Personally, I think that the authors wouldn't even need to discuss the second hit, there will always be several hits in a screen and one is under no obligation of chasing all of them. However, if they do mention it, the claims they do should be backed by the data. On that note, having established the system with the SCD1 inhibitor, it could be a quick and easy additional experiment to check for FSP-1 stability under SCD1 inhibition.

Further small point:

- I cannot find Table 1 in my material, so I cannot comment on it

- Line 153: 'We obtained similar results using HT1080GPX4KO/FSP1OE cells, where we find that the absence of RFK impairs viability and induces ferroptosis (Extend Data 1 c-f)'. Looking at the respective Figure, it appears that these are A375 cells

- Fig. 3a shows prominent FSP-1 reduction after 96h of Riboflavin deprivation, however FSP-1 does not seem to feature in the respective heatmap in Ex. Fig. 3c. (strangely there is SCD as a prominent hit). Please explain this discrepancy.

- Lines 222-223, please check the panel reference. In part they do not seem to lead to the right panels.

- Line 234 'FSP1 emerged again as the most downregulated protein in riboflavin-deficient conditions'. This appears like a bit of an oversell, as indeed in the time course data (Ex. Fig 3), FSP-1 is the most prominent down regulated protein only in the 96h time point shown in Fig. 3. The wording needs to be adjusted.

- Line 307, it should be Fig. 4d

- Ex. Fig 2j. This panel should be rearranged to an earlier spot according to the flow of the text.

I hope these comments help.

Peter Kreuzaler

ABSTRACT AND MAIN TEXT – please follow the guidelines that are specific to the format of your manuscript, as listed in our Guide to

Authors (http://www.nature.com/ncb/pdf/ncb_gta.pdf) Briefly, Nature Cell Biology Articles, Resources and Technical Reports have 3500 words, including a 150 word abstract, and the main text is subdivided in Introduction, Results, and Discussion sections. Nature Cell Biology Letters have up to 2500 words, including a 180 word introductory paragraph (abstract), and the text is not subdivided in sections.

Methods should be written concisely, but should contain all elements necessary to allow interpretation and replication of the results. As a guideline, Methods sections typically do not exceed 3,000 words. The Methods should be divided into subsections listing reagents and techniques. When citing previous methods, accurate references should be provided and any alterations should be noted. Information must be provided about: antibody dilutions, company names, catalogue numbers and clone numbers for monoclonal antibodies; sequences of RNAi and cDNA probes/primers or company names and catalogue numbers if reagents are commercial; cell line names, sources and information on cell line identity and authentication. Animal studies and experiments involving human subjects must be reported in detail, identifying the committees approving the protocols. For studies involving human subjects/samples, a statement must be included confirming that informed consent was obtained. Statistical analyses and information on the reproducibility of experimental results should be provided in a section titled "Statistics and Reproducibility".

All Nature Cell Biology manuscripts submitted on or after March 21 2016 must include a Data availability statement at the end of the Methods section. For Springer Nature policies on data availability see <http://www.nature.com/authors/policies/availability.html>; for more information on this particular policy see <http://www.nature.com/authors/policies/data/data-availability-statements-data-citations.pdf>. The Data availability statement should include:

- Accession codes for primary datasets (generated during the study under consideration and designated as "primary accessions") and secondary datasets (published datasets reanalysed during the study under consideration, designated as "referenced accessions"). For primary accessions data should be made public to coincide with publication of the manuscript. A list of data types for which submission to community-endorsed public repositories is mandated (including sequence, structure, microarray, deep sequencing data) can be found here <http://www.nature.com/authors/policies/availability.html#data>.
- Unique identifiers (accession codes, DOIs or other unique persistent identifier) and hyperlinks for datasets deposited in an approved repository, but for which data deposition is not mandated (see here for details <http://www.nature.com/sdata/data-policies/repositories>).
- At a minimum, please include a statement confirming that all relevant data are available from the authors, and/or are included with the manuscript (e.g. as source data or supplementary information), listing which data are included (e.g. by figure panels and data types) and mentioning any restrictions on availability.
- If a dataset has a Digital Object Identifier (DOI) as its unique identifier, we strongly encourage including this in the Reference list and citing the dataset in the Methods.

We recommend that you upload the step-by-step protocols used in this manuscript to [protocols.io](https://www.protocols.io). More details can be found at <https://www.protocols.io/help/publish-articles>.

All imaging data should be accompanied by scale bars, which should be defined in the legend.

Cropped images of gels/blots are acceptable, but need to be accompanied by size markers, and to retain visible background signal within the linear range (i.e. should not be saturated). The boundaries of panels with low background have to be demarked with black lines. Splicing of panels should only be considered if unavoidable, and must be clearly marked on the figure, and noted in the legend with a statement on whether the samples were obtained and processed simultaneously. Quantitative comparisons between samples on different gels/blots are discouraged; if this is unavoidable, it should only be performed for samples derived from the same experiment with gels/blots were processed in parallel, which needs to be stated in the legend.

The total number of Supplementary Figures (not including the "unprocessed scans" Supplementary Figure) should not exceed the number of main display items (figures and/or tables (see our Guide to Authors and March 2012 editorial <http://www.nature.com/ncb/authors/submit/index.html#suppinfo>; <http://www.nature.com/ncb/journal/v14/n3/index.html#ed>). No restrictions apply to Supplementary Tables or Videos, but we advise authors to be selective in including supplemental data.

GUIDELINES FOR EXPERIMENTAL AND STATISTICAL REPORTING

REPORTING REQUIREMENTS – To improve the quality of methods and statistics reporting in our papers we have recently revised the

reporting checklist we introduced in 2013. We are now asking all life sciences authors to complete two items: an Editorial Policy Checklist (found here <https://www.nature.com/authors/policies/Policy.pdf>) that verifies compliance with all required editorial policies and a reporting summary (found here <https://www.nature.com/authors/policies/ReportingSummary.pdf>) that collects information on experimental design and reagents. These documents are available to referees to aid the evaluation of the manuscript. Please note that these forms are dynamic 'smart pdfs' and must therefore be downloaded and completed in Adobe Reader. We will then flatten them for ease of use by the reviewers. If you would like to reference the guidance text as you complete the template, please access these flattened versions at <http://www.nature.com/authors/policies/availability.html>.

Version 1:

Decision Letter:

Our ref: NCB-LE57142A

14th October 2025

Dear Dr. Friedmann Angeli,

Thank you for submitting your revised manuscript "Riboflavin metabolism shapes FSP1-driven ferroptosis resistance" (NCB-LE57142A). It has now been seen by the original referees and their comments are below. The reviewers find that the paper has improved in revision, and therefore we'll be happy in principle to publish it in Nature Cell Biology, pending minor revisions to comply with our editorial and formatting guidelines.

Meanwhile, please amend the manuscript files following the guidelines below:

1. Please restructure the main text as Article with Introduction, Results and Discussions sections;
2. Please remove all embedded figures, and only provide them separately, one figure per file;
3. For better legibility, please ensure that all figures adhere to a maximum page size of roughly 180mm wide x 200mm high to fit standard page format and use a font size of no smaller than 7pt Arial or Helvetica throughout the figures.

Thank you again for your interest in Nature Cell Biology Please do not hesitate to contact me if you have any questions.

Sincerely,

Zhe Wang, PhD
Senior Editor
Nature Cell Biology

Tel: +44 (0) 207 843 4924
email: zhe.wang@nature.com

Reviewer #1 (Remarks to the Author):

The authors have addressed comments from this reviewer satisfactorily and with carefully designed experiments and reasoning. Congratulations on the excellent work! Recommend to accept.

Reviewer #2 (Remarks to the Author):

I am satisfied with the responses. And the revision of the manuscript improves what was already an important contribution.

Reviewer #3 (Remarks to the Author):

I have now conducted the revision of the amended manuscript Riboflavin metabolism shapes FSP1-driven ferroptosis resistance by Skafar et al.

I feel like this was a very positive revision and the study is now stronger and more conclusive. The rebuttal figures 13-15 answer my most pressing points and consequently my most pressing questions have been answered. Looking at my peers' comments, their questions appear to also have been mostly answered. I do not see the need for further revisions. I hope that my suggestions have been seen as helpful to make the study more complete.

Version 2:

Decision Letter:

Dear Dr Friedmann Angeli,

I am pleased to inform you that your manuscript, "Riboflavin metabolism shapes FSP1-driven ferroptosis resistance", has now been accepted for publication in Nature Cell Biology.

Please note that *Nature Cell Biology* is a Transformative Journal (TJ). Authors may publish their research with us through the traditional subscription access route or make their paper immediately open access through payment of an article-processing charge (APC). Authors will not be required to make a final decision about access to their article until it has been accepted. [Find out more about Transformative Journals](https://www.springernature.com/gp/open-research/transformative-journals)

Authors may need to take specific actions to achieve compliance with funder and institutional open access mandates. If your research is supported by a funder that requires immediate open access (e.g. according to [Plan S principles](https://www.springernature.com/gp/open-science/plan-s-compliance) or the [NIH public access policy](https://www.springernature.com/gp/open-science/us-federal-agency-compliance)) then you should

select the gold OA route, and we will direct you to the compliant route where possible. Because authors warrant under our subscription licensing terms that they haven't committed to licensing any version of their article under a licence inconsistent with the terms of our agreement – including the applicable embargo period – publication under the subscription model isn't suitable for authors whose funders require no embargo.

If you have not already done so, we strongly recommend that you upload the step-by-step protocols used in this manuscript to protocols.io (<https://protocols.io>), an open online resource that allows researchers to share their detailed experimental know-how. All uploaded protocols are made freely available and are assigned DOIs for ease of citation. Protocols and Nature Portfolio journal papers in which they are used can be linked to one another, and this link is clearly and prominently visible in the online versions of both. Authors who performed the specific experiments can act as primary authors for the Protocol as they will be best placed to share the methodology details, but the Corresponding Author of the present research paper should be included as one of the authors. By uploading your Protocols onto protocols.io, you are enabling researchers to more readily reproduce or adapt the methodology you use, as well as increasing the visibility of your protocols and papers. You can also establish a dedicated workspace to collect your lab Protocols. Further information can be found at <https://www.protocols.io/help/publish-articles>.

Nature Cell Biology encourages authors presenting evidence for cell, biological, molecular, and genetic interactions to consider communicating these findings using Biofactoid (<https://biofactoid.org/>). This tool helps users share a searchable representation of interactions (e.g. binding, gene expression, post-translational modification) between genes, gene products, or chemicals. Information added to Biofactoid, with author attribution, is shared on social media and public databases, such as Pathway Commons, where it can be discovered and analyzed in the context of a large and growing corpus of knowledge.

With kind regards,

Zhe Wang, PhD
Senior Editor
Nature Cell Biology

Tel: +44 (0) 207 843 4924
email: zhe.wang@nature.com

** Visit the Springer Nature Editorial and Publishing website at http://editorial-jobs.springernature.com?utm_source=ejp_NCB_email&utm_medium=ejp_NCB_email&utm_campaign=ejp_NCB for more information about our career opportunities. If you have any questions please click [here](mailto:editorial.publishing.jobs@springernature.com).

Reviewers' Comments:

**Reviewer #1 (Remarks to the Author):**

Summary:

This manuscript focuses on FSP1, also known as AMID and AIFM2, a crucial suppressor of
the iron-dependent form of cell death known as ferroptosis. By using a cellular model in which
cell survival relies on FSP1 overexpression, the authors conducted a CRISPR screening and
identified the riboflavin kinase RFK as a required factor for the ferroptosis-suppressing activity
of FSP1. Through a series of subsequent experiments, the authors confirmed the established
knowledge that FAD is an essential cofactor of FSP1. They also discovered that an analog of
FAD, roFAD, can abrogate the ferroptosis-inhibitory activity of FSP1, likely by competing with
FAD for FSP1-binding.

Originality and significance:

While technically this is overall an excellent study, the main conclusion of the study is
unfortunately not new, as it has already been documented that FAD metabolites, particularly,
6-hydroxy-FAD (FAD binds to FSP1/AMID and is then converted to 6-hydroxy-FAD within the
protein), is a bona fide cofactor of FSP1. This knowledge has been annotated in the UniProt
database (<https://www.uniprot.org/uniprotkb/Q9BRQ8/entry>) and was reported long before the
field of ferroptosis (PMID: 15958387). So, in essence the current manuscript confirmed these
early discoveries but only by using another functional analysis of FSP1, inhibition of ferroptosis.
With that said, a novel aspect of the paper is the discovery of roFAD as a potent inhibitor of
FSP1, likely through competing with FAD for FSP1 binding. If the authors can extend this
aspect by clearly defining the underlying mechanisms and exploring the therapeutic potential,
the significance of the study can be improved. The authors are recommended to conduct
additional experiments to address following questions:

We thank the reviewer for their assessment and greatly appreciate the critical comments aimed
at enhancing the novelty and overall impact of our study. In response, we have conducted a
series of new experiments to address the reviewer's questions and comments. In the revised
manuscript, we now include additional data that support a more detailed mechanism of action
for roseoflavin. These new data significantly strengthen the manuscript.

We also want to clarify that the main goal of our work is not to re-demonstrate that FSP1 is a
flavoprotein and consequently requires flavin as a cofactor, which is already well-established,
but to emphasize that riboflavin metabolism is an actionable metabolic node whose activity
can be specifically modulated or exploited in the context of antimetabolites such as roseoflavin.
By identifying and characterizing roseoflavin as an inhibitor of FSP1, we expand the
understanding of its role in ferroptosis and explore new possibilities for therapy.

A point-by-point response follows.

What is the mechanisms of action of roFAD? Does it have a much higher affinity to FSP1 than
FAD (hence competition), and is roFAD not able to undergo 6-hydroxylation upon FSP1-
binding and thus gives rise to an inactive FSP1? Answering these questions can modestly
improve the novelty.

Per the reviewer's suggestions, we performed additional experiments to investigate the
mechanism of action of roseoflavin. These results are now presented in Fig. 5 and Extended
Data Fig. 5d–f.

We began with **molecular dynamics simulations** of FSP1 with FAD, 6-OH-FAD, and roFAD.
These simulations suggested that roFAD can bind within the FSP1 active site and partially
stabilize the protein, but less effectively than FAD and 6-OH-FAD (Extended Data Fig. 5d; see
also the flavin-free simulation in Extended Data Fig. 2n; also presented here as **Rebuttal Fig.**
**1** for convenience).

**Rebuttal Fig. 1.** Root mean square deviation (RMSD) of FSP1 backbone in FSP1-NAD (a), FSP1-NAD-FAD (b),
FSP1-NAD-6-OH-FAD (c) and FSP1-NAD-roFAD (d) complexes for three 500 ns-MD simulations.

Guided by these insights, we set to address the biochemical and functional consequences.
Attempts to produce recombinant FSP1–roFAD complexes in modified *E. coli* strains did not
yield sufficient material for flavin analysis, unlike with riboflavin (see **Appendix** at the end of
this letter). We therefore generated A375 cells stably expressing FSP1-Flag, purified the
protein by immunoprecipitation (IP), and tested its enzymatic activity using NADH as an

electron donor. We could demonstrate that the assay is specific (inactive in mock and blocked
 by iFSP1) (Fig. 5a,b; also presented here as **Rebuttal Fig. 2** for convenience).

**Rebuttal Fig. 2. a**, Immunoblot (IB) analysis of FSP1, flag and vinculin in input (IN), flow-through (FT) and elution
 (E) fractions from a representative immunoprecipitation (IP) assay. **b**, Time courses following resazurin reduction
 (100 μ M) by fluorescence in the presence of NADH (200 μ M) and elution fractions from IP (mock and FSP1-flag)
 in the absence or presence of iFSP1 (3 μ M) in phosphate buffer saline (PBS). RFU, relative fluorescence units.

Riboflavin starvation destabilized FSP1, but supplementation with either riboflavin or
 roseoflavin restored protein stability, consistent with incorporation of FAD or roFAD. Strikingly,
 however, enzymatic assays revealed that FSP1 stabilized by roFAD was catalytically inactive,
 in contrast to the restored activity seen with riboflavin (Fig. 5c,d; also presented here as
 **Rebuttal Fig. 3** for convenience).

**Rebuttal Fig. 3. a**, Immunoblot (IB) analysis of FSP1, flag and vinculin in input (IN), flow-through (FT) and elution
 (E) fractions from IP. Cells were cultured in riboflavin-deficient media for 72 h and subsequently supplemented with
 either riboflavin (100 nM RbF) or roseoflavin (100 nM RoF) for an additional 24 h. **b**, Time courses following
 resazurin reduction (100 μ M) by fluorescence in the presence of NADH (200 μ M) and E fractions shown in a in the
 absence or presence of iFSP1 (3 μ M) in PBS. RFU, relative fluorescence units.

We extended these observations with **proteome analysis** after 96 h of riboflavin deprivation
 followed by refeeding. Both riboflavin and roseoflavin rescued the expression of FSP1 and
 other flavoproteins (Fig. 5e,f; also presented here as **Rebuttal Fig. 4** for convenience). Yet,
 stress-response proteins (NRF2/ATF4 targets such as HMOX1, ATF3, OSGIN1, SESN2,
 SLC7A11) remained elevated in roseoflavin-treated cells (Extended Data Fig. 5e,f; also

presented here as **Rebuttal Fig. 4 and 5** for convenience), suggesting that roFAD supports
 protein stability but fails to restore enzymatic function and overall redox/metabolic
 homeostasis.

**Rebuttal Fig. 4. a.** Volcano plot of quantified proteins showing their change in A375 parental cells cultured in media
 supplemented with 100 nM riboflavin for 24 h relative to riboflavin-deficient media. Proteins are plotted based on
 their fold change (FC: riboflavin/no riboflavin). **b.** Volcano plot of quantified proteins showing their change in A375
 parental cells cultured in media supplemented with 100 nM roseoflavin for 24 h relative to riboflavin-deficient media.
 Proteins are plotted based on their fold change (FC: roseoflavin/no riboflavin). **c.** Volcano plots of quantified proteins
 showing their change in A375 parental cells that were re-fed with roseoflavin or riboflavin (100 nM) followed 96
 90 hours of riboflavin deprivation. Proteins are plotted based on their fold change (FC: roseoflavin/riboflavin). The
 91 statistical significance of the respective ratios is plotted on the y-axis (n = 5 technical replicates).

**Rebuttal Fig. 5.** Heatmap of quantified proteins showing their change in A375 parental cells that were re-fed with roseoflavin or riboflavin (100 nM) following 96 hours of riboflavin deprivation. The heatmap shows five technical replicates from one independent experiment.

In parallel, we confirmed that roseoflavin restores FSP1-GFP expression in A375 cells cultured under riboflavin-deficient conditions by fluorescence analysis. Notably, inhibition of the proteasome with MG-132 prevented the loss of FSP1 in the absence of flavin cofactors (Fig. 98 5h; also presented here as **Rebuttal Fig. 6** for convenience).

**Rebuttal Fig. 6.** Fluorescence histograms of A375 overexpressing FSP1-GFP cultured in riboflavin-deficient or supplemented with 1 μ M of riboflavin medium for 96 h, followed by treatment with DMSO, MG132 (1 μ M) or roseoflavin (RoF, 100 nM) for 24 h.

Mechanistically, these results align with known effects of isoalloxazine-ring substitutions on redox potential¹. The C8 dimethylamino group in roseoflavin is predicted to increase redox potential and impair electron transfer. Consistently, in solution assays, riboflavin but not roseoflavin oxidized NADH (Fig. 5i ; also presented here as **Rebuttal Fig. 7a** for convenience)

and promoted resazurin reduction (Fig. 5j; also presented here as **Rebuttal Fig. 7b** for
 convenience). While these assays must be interpreted with caution given the absence of a
 protein environment, they support a model in which roFAD incorporates into FSP1 but cannot
 accept electrons from NAD(P)H, rendering the enzyme inactive.

**Rebuttal Fig. 7. a**, Time courses following NADH oxidation (200 μ M) by absorbance at 340 nm in the presence of
 either riboflavin (25 μ M) or roseoflavin (25 μ M). **b**, Representative time courses following resazurin (RZ)
 (25 μ M) by fluorescence in the presence of the indicated reagents (initial concentrations: 200 μ M NADH, 25 μ M
 riboflavin, 25 μ M roseoflavin).

**In summary**, our modeling and functional data converge on a mechanism in which roseoflavin-
 derived cofactors occupy the FSP1 flavin binding site, stabilize the protein, but abolish its
 catalytic activity. These findings highlight riboflavin antimetabolites as effective inhibitors of
 FSP1 and provide a framework for pharmacologically targeting ferroptosis protection in cancer
 cells (Fig. 5k; also presented here as **Rebuttal Fig. 8** for convenience).

**Rebuttal Fig. 8.** Riboflavin metabolism supports FSP1 function and promotes ferroptosis resistance. This is
 therefore a process that riboflavin analogs, such as roseoflavin can modulate.

Considering these new results, we accordingly amended the text, which now reads as follows:

**(Line 345)** Finally, we explored the mechanism of action of roseoflavin. To this end, we first
generated A375 cells stably expressing either mock (control) or FSP1-Flag and performed
immunoprecipitation (IP) assays to isolate and purify the flag-tagged protein (Fig. 5a). Upon
successful purification, we measured enzymatic activity in a cell-free system using NADH as
the electron donor and monitored FSP1 activity by following resazurin reduction through
fluorescence (Fig. 5b). Importantly, there was no activity in the mock condition, and the signal
was abolished by iFSP1, thus confirming the specificity of the assay.

**(Line 352)** To assess whether roseoflavin can be metabolized, incorporated into FSP1, and
modulate its activity, we combined modeling and functional assays.

**(Line 354)** Molecular dynamics (MD) simulations of FSP1 and the cofactors FAD, 6-OH-FAD
and roFAD (Extended Data Fig. 5d) suggested that roFAD binds to FSP1, although the
resulting complex appears to be less stable.

**(Line 357)** We next examined the functional outcome by starving cells of riboflavin for 72 hours
and then refeeding the cells with riboflavin or roseoflavin. As expected, flavin starvation
destabilized FSP1, while supplementation with either compound stabilized the protein,
confirming their conversion to FAD or roFAD and proper incorporation into the enzyme.
Strikingly, IP assays showed that although roseoflavin supplementation stabilized FSP1, the
enzyme was catalytically inactive (Fig. 5c,d), in contrast to the activity restored by riboflavin.

**(Line 363)** In addition, we extended our proteomics analysis and confirmed that the expression
of FSP1 and other flavoproteins is restored upon refeeding cells with either riboflavin (Fig. 5e)
or roseoflavin (Fig. 5f) following 96 hours of riboflavin deprivation. Notably, some proteins
associated with stress response (NRF2 targets) remained upregulated in roseoflavin-treated
cells (Extended data Fig. 5e,f), suggesting that roseoflavin-derived metabolites can stabilize
some flavoproteins (Fig. 5g) but fail to restore their full functionality and flavin homeostasis.

**(Line 369)** In parallel, we confirmed that roseoflavin restores FSP1-GFP expression in A375
cells cultured under riboflavin-deficient conditions by cell sorting and fluorescence analysis.
Notably, inhibition of the proteasome with MG-132 prevented the loss of FSP1 in the absence
of flavin cofactors (Fig. 5h).

**(Line 372)** Substitutions on the isoalloxazine ring are known to change flavin redox properties
significantly¹⁹. These changes in the electronic density of the ring offer a plausible explanation
for why roFAD stabilizes, but inactivates FSP1.

**(Line 375)** Based on this, we hypothesized that the electron flow within roFAD would be less
favorable.

As roFAD is not commercially available, we could not directly measure its reactivity, but we
analyzed the reactivity of the roFAD precursor, roseoflavin, which retains the isoalloxazine ring.
Monitoring NADH oxidation at 340 nm revealed that riboflavin, but not roseoflavin, oxidized
NADH (Fig. 5i). Similarly, only riboflavin facilitated resazurin reduction (Fig. 5j).

**(Line 380)** While these assays were conducted in aqueous solution and must be interpreted
with caution, as the protein environment can further modulate the redox potential of flavins,
this combined evidence supports a mechanism in which roseoflavin-derived metabolites
incorporate into FSP1 but cannot accept electrons from NAD(P)H, thereby inactivating the
enzyme.

**(Line 384)** Altogether, these results demonstrate that FSP1 activity can be effectively inhibited
by riboflavin antimetabolites, offering additional opportunities to target ferroptosis protective
mechanisms in cancer cells (Fig. 5k).

In terms of significance: is roFAD a potential therapeutic agent in xenograft models, and how
is it behaving in comparison with other "conventional" FSP1 inhibitors (potency, specificity,
toxicity)?

Our results show that roFAD impairs FSP1 function, but its mode of action is less specific than
"conventional" small-molecule inhibitors such as iFSP1, icFSP1, or FSEN1, which display
clearer target selectivity. Because roseoflavin is metabolized into cofactors that can bind
multiple flavoproteins², its use is associated with off-target effects. This lack of specificity can
be seen as a disadvantage in terms of potency and safety. Nonetheless, riboflavin
antimetabolites offer a potential advantage over direct inhibitors: resistance is less likely to
develop, since mutations that weaken roFAD incorporation would also reduce utilization of
riboflavin itself, ultimately limiting cofactor availability and impairing flavoprotein activity more
broadly.

Roseoflavin has been reported to be suitable for *in vivo* use³, but given its broad activity, we
did not pursue xenograft experiments here. Instead, we view direct inhibitors and
antimetabolites as complementary approaches: direct inhibitors provide higher specificity,
whereas antimetabolites have been shown to be more potent and exploit pathway-level
vulnerabilities that may reduce the likelihood of resistance. Our findings highlight both the
limitations and the opportunities of antimetabolite strategies; these aspects are discussed in
the text.

**Reviewer #2 (Remarks to the Author):**

This study systematically investigates the upstream regulatory mechanisms of FSP1 and
elucidates the specific pathway through which riboflavin (VB2) metabolism inhibits ferroptosis
via non-RTA activity. Through an innovative FSP1-dependent focused pool screening, the
authors first identified RTK as a modulator of FSP1-dependent ferroptosis. Further exploration
revealed that the endogenous riboflavin-FMN-FAD metabolic axis suppresses ferroptosis in an
FSP1-dependent manner. Mechanistically, this axis regulates both the enzymatic activity and
protein stability of FSP1. Notably, the authors discovered that Rof, a structural analog of Rbf,
disrupts the ferroptosis-protective effects of the riboflavin-FMN-FAD axis, thereby proposing a
novel targeting strategy for FSP1 modulation.

Collectively, this work presents conceptual advances supported by rigorous experimental
design and robust datasets. In ferroptosis research, it uncovers a protein-level regulatory
mechanism in FSP1's upstream pathway—analogue to the selenium metabolism governing
GPX4—thereby expanding the understanding of how endogenous metabolites determine
ferroptosis susceptibility. Additionally, it provides new insights into the antioxidant mechanisms
of vitamin B2 through the lens of membrane phospholipid oxidation homeostasis.

We greatly appreciate the reviewer's positive evaluation.

**Comments:**

**1. Screening Strategy**

The focused screening approach demonstrates ingenuity in enhancing threshold sensitivity for
FSP1 regulators. Intriguingly, SCD1—a known ferroptosis regulator—emerged prominently in
the screening targeting FSP1. Should other critical ferroptosis resistance factors exist in the
library but remain undetected, investigating potential FSP1-dependent specificity of SCD1's
regulatory role would merit future exploration.

**Some questions:**

- Why focus on a drug-targeted library rather than a genome-wide screening?

The screening was conducted using libraries that Ulrich Elling's group developed at IMP-
Vienna. Their team designed eight distinct sub-pooled libraries, each targeting different
biological pathways. When combined, these libraries provide genome-wide coverage.
Additionally, most sgRNAs targeting a single gene are included in at least two sub-pools,
ensuring redundancy and enhancing reliability.

The rationale behind using these libraries was to identify overlapping gene hits across different
screens, thereby increasing confidence. We initially focused on the druggable gene set to

prioritize genes with therapeutic potential. During this stage, we already identified a high-
confidence hit that stood out as particularly interesting. Given its strong effect and lack of
reported role in ferroptosis, we chose to proceed with further investigation rather than
expanding to a genome-wide screen. We have provided as a supplementary file the complete
list of genes contained in the druggable gene set.

- Why are there no corresponding drugs for the RTKs identified in the drug-targeted library?

RFK was included in the drug-targeted library because of its kinase activity, making it
druggable. However, no direct inhibitors for this kinase have been identified. Since we haven't
specifically explored RFK's druggability, the reason for the lack of inhibitors remains unclear.
Nonetheless, our findings, along with other research showing that cancers like AML have
higher dependence on RFK, suggest that developing inhibitors for this kinase could be
promising⁴.

- Why were other enzymes in the FAD pathway not identified? Is it because this library does
not include these genes?

Aside from RFK, the only other FAD pathway component in the tested library was the riboflavin
transporter SLC52A2. Guides targeting SLC52A2 showed a trend toward depletion when
liproxstatin-1 was absent, but this trend wasn't statistically significant. In follow-up experiments
presented in the manuscript, we found that SLC52A2 knockout cells could only be created
when cultures were supplemented with uridine. Since the initial screen didn't include uridine,
SLC52A2-deficient cells were likely lost in both control and Lip-1 conditions, which would
explain why the gene didn't score as significant.

2. FLAD1 Knockout Phenotype

The observation that FLAD1 knockout had no impact on non-GPX4 inhibitors (e.g.,
Erastin/BSO) suggests distinct mechanisms of Erastin/BSO and GPX4 inhibition. So, what is
the different about these ferroptosis-inducer? Moreover, FLAD1 ablation impairs both FSP1
activity and stability, seemingly conflicting with prior reports that FSP1 overexpression inhibits
Erastin-induced ferroptosis. This paradox warrants deeper mechanistic discussion.

We thank the reviewer for raising this interesting point. While direct GPX4 inhibitors are potent
inducers of ferroptosis, GSH-depleting strategies – such as treatment with BSO or erastin –
have proved to be less effective in inducing ferroptosis, likely due to the presence of
compensatory mechanisms⁵ and pleiotropic effects of other pathways that are activated.
Remarkably, GPX4 can exploit alternative substrates once GSH concentration becomes
limiting⁶. In addition, in the context of erastin treatment (and system xc- inhibition), a more
active transsulfuration pathway can provide an alternative source of cysteine for GSH synthesis.

We have observed that the ability of erastin and BSO to induce ferroptosis is cancer cell line
 dependent. In our experiments, and consistent with previous reports⁷, erastin was less
 effective in inducing ferroptosis in A375 cells, as Liproxstatin-1 (Lip-1) only partially rescued
 erastin-induced cell death. In contrast, in HT1080 cells, erastin-induced cell death was strongly
 rescued by Lip-1, confirming a clear ferroptotic mechanism (**Rebuttal Fig. 9a**, see below).

To further investigate this, we re-tested erastin sensitivity in control (NT) and FLAD1^{KO} cells in
 both A375 and HT1080 cell lines. While no strongly remarkable difference was observed in
 A375 cells, HT1080 FLAD1^{KO} cells showed a marked sensitivity to erastin compared to the
 control (**Rebuttal Fig. 9b**, see below).

Finally, we generated HT1080 cells stably expressing either mock or FSP1-Flag and
 challenged them with erastin. Consistent with previous reports, FSP1-overexpressing cells are
 resistant to erastin treatment. Notably, deletion of FLAD1 in this background restores sensitivity
 to erastin (**Rebuttal Fig. 9c**, see below), supporting a role of FLAD1 in modulating FSP1
 function.

**Rebuttal Fig. 9. a**, Dose-dependent toxicity of erastin in A375 and HT1080 parental cell lines in the absence or
 presence of Lip-1 (500 nM). **b**, Dose-dependent toxicity of erastin in A375 and HT1080 cells transduced with either
 a non-target (NT) or FLAD1 targeting sgRNA. **c**, Immunoblot (IB) analysis (left) and dose dependent toxicity of
 erastin (right) in HT1080 cells overexpressing either an empty vector (mock) or FSP1-Flag that were transduced
 with either non-target (NT) or FLAD1 targeting sgRNAs. Cell viability was monitored using Alamar blue after 48 h
 of treatment.

In summary, the efficacy of ferroptosis inducers is strongly shaped by cellular buffering
 mechanisms. For compounds such as erastin or BSO, these compensatory pathways can dampen ferroptotic activity,
 so that much of the measured cell death is non-ferroptotic and therefore not prevented by FSP1 or influenced by
 FLAD1 loss. This explains the noted discrepancy with prior reports and underscores that the FLAD1–FSP1 axis provides
 robust protection specifically when inducers act through a “clean” ferroptotic mechanism.

Additionally, does Rbf-FMN-FAD metabolic axis affect the level of CoQ?

To address this point, we measured total CoQ levels in A375 cells, parental and FLAD1^{KO} cells
stably expressing either mock or Flag-FLAD1 (**Rebuttal Fig. 10**, see below). CoQ levels were
similar between parental and FLAD1^{KO} cells, suggesting that changes in total CoQ content are
unlikely to account for the observed phenotype.

Interestingly, FLAD1 overexpression in a FLAD1^{KO} background led to an increase in total CoQ
levels. In these cells, FLAD1 is approximately 5-fold higher than endogenous levels and
accompanied by a 2-fold increase in FAD content. This elevated availability of flavin cofactors
may support the activity of flavoproteins involved in CoQ biosynthesis (such as COQ6). An
increase in CoQ levels may also reflect an adaptation response to enhanced mitochondrial
function in FLAD1 overexpressing cells.

We acknowledge that total CoQ measurements do not directly reflect the pool relevant to
FSP1, which utilizes extra-mitochondrial CoQ to prevent ferroptosis. Quantifying the extra-
mitochondrial CoQ pool and measuring its redox status would more precisely assess substrate
availability and FSP1 function. For example, an increase in the ubiquinone/ubiquinol ratio is
expected to be in the plasma membrane of FLAD1^{KO} cells, as a result of FSP1 inhibition, and
the extent of this increase should be similar in FSP1^{KO} cells.

**Rebuttal Fig. 10.** Relative quantification of total coenzyme Q in A375 parental and A375 FLAD1^{KO} C1 cells stably
overexpressing either an empty vector (mock) or Flag-FLAD1.

3. Physiological Relevance of Rbf Levels

Would the lower physiological Rbf concentrations compared to cell culture conditions imply
attenuated FSP1 functionality in vivo theoretically? This highlights potential discrepancies
between in vitro models (influenced by selenium, cysteine, transferrin, VK/VE, etc.) and
physiological contexts—a critical consideration for translational research.

Traditional cell culture media often fail to recapitulate physiological conditions, particularly with
respect to vitamin composition. They typically contain supraphysiological levels of certain

vitamins, such as riboflavin, while others, like ascorbate (vitamin C), are entirely absent from
basic formulations. This represents a significant limitation of *in vitro* models when interpreting
mechanisms that depend on vitamin metabolism.

In the case of riboflavin, human plasma concentrations are lower than those present in
standard culture media, which could potentially result in reduced FSP1 protein abundance *in*
*vivo*. However, FSP1 functionality is not solely determined by its protein levels, but also by the
availability, compartmentalization, and redox state of its substrates, such as NAD(P)H and
ubiquinone. Moreover, a more active riboflavin metabolic pathway *in vivo* could compensate
for limited riboflavin availability, maintaining FSP1 functionality despite lower plasma
concentrations.

Taken together, these considerations highlight an important caveat: discrepancies between *in*
*vitro* and physiological riboflavin levels may influence the apparent strength of FSP1-mediated
ferroptosis protection. This underlines the need to contextualize cell culture findings carefully
and to validate them in more physiologically relevant models for robust translational
conclusions.

Regarding this comment, we amended the discussion, which now reads as follows:

**(Line 435)** *The discrepancy between riboflavin concentrations in vitro and in human plasma is therefore*
*highly relevant for translational research, as lower availability of riboflavin in vivo may attenuate FSP1-*
*mediated protection against ferroptosis.*

4. Lipid ROS Validation

In fig2k, while FLAD1 single KO and FLAD1/FSP1 double KO show comparable lipid ROS
levels, FSP1 single KO exhibits markedly elevated ROS. This disparity may need further
validation to confirm.

We thank the reviewer for bringing this to our attention. This inconsistency is likely attributed
to differences in the genetic backgrounds of the cell lines. FLAD1^{KO} was isolated as a single
clone, whereas the FSP1^{KO} cells were used as a multiclonal population selected after lentiviral
transduction of Cas9 and a FSP1-targeting sgRNAs. We acknowledge that it was not a fair
comparison, as these cell lines are not isogenic.

To address this point, we used the FLAD1/FSP1 double KO cell line. We stably expressed
either an empty vector (mock) or FLAD1 (**Rebuttal Fig. 11**, see below), thereby generating an
isogenic pair to measure the contribution of FSP1 deletion to lipid peroxidation. In these cell
lines, we did not observe any noticeable difference in BODIPY oxidation.

**Rebuttal Fig. 11. a**, Immunoblot (IB) analysis of FLAD1, FSP1, NQO1 and vinculin in the indicated cell lines. **b**,
Lipid peroxidation evaluated by C11-BODIPY 581/591 staining in the indicated cell lines. Cells were treated with
DMSO, RSL3 (200 nM), or RSL3 (200 nM) + Lip-1 (500 nM) for 6 h.

In the revised manuscript, we have updated Fig. 2k (also presented here as **Rebuttal Fig. 12**
for convenience) and now shows only the BODIPY oxidation comparison between FLAD1
single KO (+ non-target control, NT) and the FLAD1/FSP1 double KO, representing an isogenic
pair and the most relevant comparison for our study. The data demonstrate that the
FLAD1/FSP1 double KO cells did not exhibit a further increase in lipid peroxidation relative to
the FLAD1 single KO cells when assessed in an isogenic background.

**Rebuttal Fig. 12.** Lipid peroxidation evaluated by C11-BODIPY 581/591 staining of A375 FLAD1^{KO} C1 cells
transduced with either a non-targeting control (NT) or a FSP1-targeting sgRNAs (FSP1^{KO}). Cells were treated with
DMSO, RSL3 (200 nM), or RSL3 (200 nM) + Lip-1 (500 nM) for 6 h.

**Minor Points:**

1. Line 219: Evidence for FAD's regulation of FSP1 activity remains experimentally
unsubstantiated.

In the revised manuscript, we have included new experiments where we measured the
enzymatic activity of FSP1 purified from A375 cells cultured in media supplemented with either
riboflavin or roseoflavin (Fig. 5c,d; also presented here as **Rebuttal Fig. 3** for convenience,

see above). We have confirmed that FSP1 is enzymatically active when bound to FAD,
whereas its activity is abolished when the cofactor is roFAD.

2. ED4j: GPX4 upregulation and ACSL4 downregulation with Rbf dose escalation in
HT1080/A375 require clarification.

We have observed upregulation of the selenoprotein TXNRD1 in A375 cells cultured in
riboflavin deficient media, most likely as a result of NRF2 activation. A plausible explanation is
that the upregulation of this selenoprotein in HT1080 and A375 cells limits the availability of
selenium required for GPX4 translation. Consistently, GPX4 downregulation was not observed
in NRF2-mutant cell lines such as A549 and H460. Another manuscript is in preparation to
details this interesting phenotype. ACSL4 levels differences might be an adaptive response to
an increased ferroptosis susceptibility at low riboflavin concentrations.

3. Line 307: "Fig4e" likely references Fig4d.

We have modified the text accordingly.

4. ED Fig2b: Missing FLAD1 WB bands; rationale for sg2's lack of ferroptosis phenotype?

We attempted to detect FLAD1 using the available antibodies; however, despite multiple
optimization efforts, we were unable to detect endogenous protein levels. Therefore, we
confirmed the deletion of *FLAD1* by genotyping (Extended Data Fig. 2 f,g) and by measuring
relative FAD levels (Fig. 2d and Extended Data Fig. 2 c,e,i).

5. Cell viability assays (e.g., ED Fig1d): Statistical annotations (asterisks) absent.

We did not perform statistical tests on the viability assays because we have presented one
representative experiment out of 2–3 independent biological replicates, with the data from the
remaining replicates provided in the source data.

6. ED Fig1C: Mismatch text descriptions.

We have modified the text accordingly.

**Reviewer #3 (Remarks to the Author):**

Dear editor,

I reviewed now the study entitled Riboflavin metabolism shapes FSP1-driven ferroptosis
resistance by Skafar et al. and in the following paragraphs I will share my assessment.

The authors study the phenomenon of ferroptosis and the FSP-1 driven protection from it. In
previous work they had shown that FSP-1 could rescue cells from GPX4 inhibition, and in this
manuscript they unravel further mechanistic details. Namely, they deploy a CRISPR-screen that
uncovers the enzyme RFX, part of the FAD biosynthetic pathway, as a crucial player in
mediating FSP-1 dependent ferroptosis resistance in a GPX4 depleted environment. This leads
to the finding that Riboflavin, as a precursor of FAD, is crucial for FSP-1 activity, that FAD
stabilises FSP-1 and that a Riboflavin analogue, Roseoflavin, can counteract this activity.

Overall the manuscript is novel and studies an as yet unknown mechanism. The data is strong
and the impact of the findings are in line with the scope of Nature Cell Biology. I will now detail
a few further considerations.

We are most grateful for the reviewer's positive evaluation.

The main aspect that, in my opinion, needs to be addressed is the role of FAD in stabilising
FSP-1. This is a strong claim, which seems reasonable and the RMSD analysis of the FSP1-
NAD-FAD complex in Ex. Fig. 2 tries to address it by modelling. As a disclaimer, I am not able
to judge this particular experiment, as it is outside my field of expertise, however I suggest a
much simpler, but I believe important experiment that should clarify, if FAD leads to a general
stabilisation, or if cellular stresses play a role in this process. The authors have shown clear
evidence that the loss of, for example, FLAD1 leads to loss of FSP-1. Cells under FLAD1 loss
will experience a significant amount of oxidative stress. Testing if by relieving this stress, for
example by treating with Lip1 leads to a stabilisation of FSP-1, will give clearer insights into
the type of stabilisation we are looking at. Is it a physical/mechanical one, or is it a protection
against, for example, exaggerated protein oxidation followed by proteasomal turnover? The
latter could also be tested with a proteasomal inhibitor.

Thank you for this excellent comment. We have now conducted new experiments and included
new data in Fig. 5h and amended the text and the discussion accordingly.

For maintenance, FLAD1^{KO} cells are cultured in medium supplemented with Lip-1, which does
not restore FSP1 expression.

Following the reviewer's suggestion, we have now tested FSP1 expression by FACS analysis
of FSP1-GFP overexpressing cells (**Rebuttal Fig. 13a**, see below) and western blot analysis
of endogenous levels (**Rebuttal Fig. 13b**, see below) upon 24 hours of co-treatment with α -

tocopherol (α -toc) and N-acetylcysteine (NAC) in WT (non-target control, NT) and *FLAD1*^{KO}
 backgrounds. These antioxidants fail to rescue FSP1 expression in the tested conditions.
 Similarly, neither Lip-1 nor co-treatment with NAC and α -toc restored FSP1-GFP expression
 under riboflavin-deficient conditions (**Rebuttal Fig. 13c**, see below).

Notably, treatment with the proteasome inhibitor MG-132 for 24 hours rescued FSP1 protein
 levels in *FLAD1*^{KO} cells and WT cells cultured under riboflavin-deficient conditions (**Rebuttal**
 **Fig. 13a-c**, see below).

**Rebuttal Fig. 13. a**, Fluorescence histograms of A375 overexpressing FSP1-GFP cells transduced with either a
 non-target (NT) or *FLAD1*-targeting sgRNAs after treatment with DMSO, α -tocopherol (α -Toc, 10 μ M), N-
 acetylcysteine (NAC, 2.5 mM) or MG132 (1 μ M) for 24 h. **b**, Immunoblot (IB) analysis of NRF2, FSP1 and vinculin
 in A375 cells transduced with either non-target (NT) or *FLAD1* targeting sgRNAs treated with α -Toc (10 μ M), NAC
 (2.5 mM) and MG132 (1 μ M) for 24 h. NRF2 expression was checked in these experiments as a positive control for
 the proteasome inhibitor treatment. **c**, Fluorescence histograms of A375 overexpressing FSP1-GFP cultured in
 riboflavin-deficient or supplemented with 1 μ M of riboflavin medium for 96 h, followed by treatment with DMSO,
 Liproxstatin-1 (Lip-1, 500 nM), NAC (2.5 mM), α -Toc (10 μ M), MG132 (1 μ M) or roseoflavin (RoF, 100 nM) for 24 h.

We have also observed that refeeding A375 cells with roseoflavin upon 96 hours of riboflavin
 deprivation restored FSP1 expression, but did not ameliorate oxidative stress, as indicated by
 a NRF2 activation signature still present (Fig. 5 e,f and Extended data Fig. 5 e,f; also presented
 here as **Rebuttal Fig. 14** for convenience).

**Rebuttal Fig. 14. a**, Volcano plot of quantified proteins showing their change in A375 parental cells cultured in
 media supplemented with 100 nM riboflavin for 24 h relative to riboflavin-deficient media. Proteins are plotted based
 on their fold change (FC: riboflavin/no riboflavin). **b**, Volcano plot of quantified proteins showing their change in
 A375 parental cells cultured in media supplemented with 100 nM roseoflavin for 24 h relative to riboflavin-deficient
 media. Proteins are plotted based on their fold change (FC: roseoflavin/no riboflavin). **c**, Volcano plots of quantified
 proteins showing their change in A375 parental cells that were re-fed with roseoflavin or riboflavin (100 nM)
 followed 96 hours of riboflavin deprivation. Proteins are plotted based on their fold change (FC:
 roseoflavin/riboflavin). The statistical significance of the respective ratios is plotted on the y-axis (n = 5 technical
 replicates).

We have now gathered evidence supporting the conclusion that the cofactor is critical for the
 physical/mechanical stabilization of FSP1, with its absence likely leading to misfolding and
 proteasomal degradation of the protein. Notably, during the revision of our manuscript, a
 preprint from the Olzmann lab was published, reporting similar findings for the role of FAD on
 the structural stabilization of FSP1⁸, and a comparable phenomenon was previously observed
 for another flavoprotein, NQO1⁹.

In response to these additional results, we have added a new plot in Fig. 5h (also presented
 here as **Rebuttal Fig. 15** for convenience) and amended the text, which now reads as follows:

**(Line 369)** *In parallel, we confirmed that roseoflavin restores FSP1-GFP expression in A375 cells*
 *cultured under riboflavin-deficient conditions. Notably, inhibition of the proteasome with MG-132*
 *prevented the loss of FSP1 in the absence of flavin cofactors (Fig. 5h).*

**(Line 424)** *The FAD cofactor is essential for structural stabilization of FSP1, and its absence leads to*
 *misfolding and ultimately degradation via the proteasomal pathway.*

**Rebuttal Fig. 15.** Fluorescence histograms of A375 overexpressing FSP1-GFP cultured in riboflavin-deficient or
 supplemented with 1 μ M of riboflavin medium for 96 h, followed by treatment with DMSO, MG-132 (1 μ M) or
 roseoflavin (RoF, 100 nM) for 24 h.

This ties in with the second point. The authors find a second prominent hit SCD1, which they
 spent a little time on. Here they claim that the experiments they do show that ‘SCD1 depletion
 did not appear to directly impair FSP-1 function(line 116)’. I am not sure that the data in Ex. Fig.
 1a,b allow for that conclusion. Personally, I think that the authors wouldn’t even need to discuss
 the second hit, there will always be several hits in a screen and one is under no obligation of
 chasing all of them. However, if they do mention it, the claims they do should be backed by the
 data. On that note, having established the system with the SCD1 inhibitor, it could be a quick
 and easy additional experiment to check for FSP-1 stability under SCD1 inhibition.

In the revised manuscript, we have removed the sentence in line 116 as we fully agree that the
 data do not properly back it. We hypothesize that SCD1 inhibition indirectly impairs FSP1
 function, as we observed cell death in HT1080^{GPX4KO/FSP1OE} cells under SCD1 inhibition only at
 very low density and after 144 hours of treatment. In contrast, treatment with a direct FSP1
 inhibitor (iFSP1) induces cell death within a few hours in these cells. Following the reviewer’s
 suggestion, we have now conducted additional experiments to assess whether SCD1 inhibition
 affects FSP1 protein levels or subcellular localization.

First, we tested FSP1 stability under SCD1 inhibition in HT1080^{GPX4KO/FSP1OE} and A375 cells
 after 48 and 96 hours of treatment (0 - 10 μ M) (**Rebuttal Fig. 16a,b**, see below). We found no
 marked difference in FSP1 protein abundance under SCD1 inhibition by immunoblotting under
 these conditions. Next, to investigate whether SCD1 inhibition affects FSP1 subcellular
 localization, we generated A375 cells stably expressing FSP1-GFP, treated them with the
 SCD1 inhibitor for 72 hours, and visualized them by confocal microscopy. No substantial
 changes in FSP1 localization under SCD1 inhibition were observed (**Rebuttal Fig. 16c**, see
 below).

Finally, we used A375 FSP1-deficient cells available in our laboratory. These cells were pre-
 incubated with the SCD1 inhibitor for 24 hours, followed by treatment with the GPX4 inhibitor

RSL3 for an additional 48 hours. Sensitization to ferroptosis was still observed in FSP1-
 deficient cells, albeit to a lesser extent than in control cells, suggesting that SCD1-mediated
 ferroptosis induction is only partially dependent on FSP1 function (**Rebuttal Fig. 16d**, see
 below). Taken together, these results suggest that loss or inhibition of SCD1 increases the pool
 of oxidizable substrates by reducing monounsaturated fatty acids, which are less prone to lipid
 peroxidation and normally help to protect against ferroptosis. As a result, the excess of
 oxidizable lipids may deplete membrane-embedded antioxidants faster than FSP1 can
 regenerate them, thereby triggering ferroptosis in the absence of GPX4.

 **Rebuttal Fig. 16. a**, Immunoblot (IB) analysis of FSP1 and vinculin in HT1080GPX4KO/FSP1OE cells treated with
 the indicated concentrations of the SCD1 inhibitor CAY10566 for 48 and 96 hours. **b**, IB analysis of FSP1, GPX4
 and vinculin in A375 cells treated with the indicated concentrations of the SCD1 inhibitor CAY10566 for 48 and 96
 507 hours. **c**, Confocal microscopy images of A375 cells that express GFP-tagged FSP1 incubated with a SCD1 inhibitor
 (CAY10566 0.5 and 1 μ M) for 72 hours. **d**, Dose-dependent toxicity of RSL3 in A375 parental and A375 FSP1^{KO}
 cells that were pre-treated with a SCD1 inhibitor (CAY10566 0.5 and 1 μ M) for 24 hours. Cell viability was monitored
 using Alamar blue after 48 h of RSL3 treatment.

**Further small point:**

**-I cannot find Table 1 in my material, so I cannot comment on it**

**-Line 153: 'We obtained similar results using HT1080GPX4KO/FSP1OE cells, where we find**
 **that the absence of RFK impairs viability and induces ferroptosis (Extend Data 1 c-f)'. Looking**
 **at the respective Figure, it appears that these are A375 cells**

We have modified the text accordingly

-Fig. 3a shows prominent FSP-1 reduction after 96h of Riboflavin deprivation, however FSP-1
does not seem to feature in the respective heatmap in Ex. Fig. 3c. (strangely there is SCD as
a prominent hit). Please explain this discrepancy,

We believe we have this discrepancy because the proteomics raw data at 96 hours contains
an outlier value (**Rebuttal Fig. 17**, see below), and therefore, FSP1 was not one of the top hits
in this particular analysis.

**Rebuttal Fig. 17.** FSP1 protein levels (log₁₀LFQ intensity) at different time points (24, 48, 96 and 144 h) in A375
cells cultured in riboflavin-deficient (No RbF) or supplemented with 1 μM of riboflavin (RbF) media. ND, non
detected.

-Lines 222-223, please check the panel reference. In part they do not seem to lead to the right
panels.

We have modified the text accordingly.

-Line 234 'FSP1 emerged again as the most downregulated protein in riboflavin-deficient
conditions'. This appears like a bit of an oversell, as indeed in the time course data (Ex. Fig 3),
FSP-1 is the most prominent down regulated protein only in the 96h time point shown in Fig.
3. The wording needs to be adjusted.

We have adjusted the wording accordingly.

-Line 307, it should be Fig. 4d

We have modified the text accordingly.

-Ex. Fig 2j. This panel should be rearranged to an earlier spot according to the flow of the text.

We have modified the text accordingly.

**Appendix**

**Recombinant expression of FSP1 in *E. coli***

We attempted to produce the recombinant hFSP1 loaded with the cofactors FAD or roFAD in
 *E. coli*. A modified strain from CmpX13 was used (CpXFAD: BL21(DE3) Δ *manX*::*ribM*,
 Δ *ribE*::*ribCF*, see figure for the reviewer, below)¹⁰. In this strain, there is an introduction of the
 heterologous riboflavin transporter (*ribM*) to efficiently import riboflavin and its analog
 roseoflavin. It is also auxotrophic for riboflavin (*rib*-) due to the chromosomal deletion of *ribE*
 (riboflavin synthase, EC 2.5.1.9). In addition, contains a copy of *ribCF* (replacing *ribE*),
 encoding the endogenous bifunctional flavokinase/FAD synthetase which produces both FMN
 and FAD. In the presence of roseoflavin, this enzyme also catalyses the production of roFAD.
 Riboflavin auxotrophy is important to be able to regulate cofactor loading of recombinant
 flavoproteins.

**Appendix Fig. 1.** A schematic view of the *Escherichia coli* strains CpXFMN and CpXFAD employed for the *in vivo*
 generation of flavoproteins loaded with the FMN/FAD-cofactor analogs RoFMN or RoFAD. The genes of 40 different
 recombinant flavoproteins (FP; grey ovals) are expressed using the expression plasmid pCA24N¹¹. Upon induction
 of protein synthesis, riboflavin (RbF) or roseoflavin (RoF) are added to the growth medium. Both flavins are taken
 up via RibM (the corresponding gene *ribM* from *Corynebacterium glutamicum* replaces *manX* in the *E. coli*
 chromosome). *E. coli* naturally does not produce a flavin transporter. The His6-tagged recombinant flavoproteins
 combine with FMN, FAD, RoFMN or RoFAD, are purified using affinity chromatography and are analysed with
 regard to their cofactor content employing HPLC/MS. CpXFMN contains the gene *FMN1* from
 *Schizosaccharomyces pombe*¹². The gene product FMN1 produces FMN from riboflavin and ATP and RoFMN from
 roseoflavin and ATP. FMN1 (replacing *ribE*) was inserted into the chromosome to enhance intracellular synthesis of
 FMN analogs and to stimulate loading of FMN dependent flavoproteins with FMN or RoFMN. FMN1 is under control
 of the *ribE* promoter Pribep5. Analogously, *E. coli* CpXFAD is used for the analysis of FAD-dependent flavoproteins.
 CpXFAD contains an additional copy of *E. coli* *ribCF* (replacing *ribE*) encoding the endogenous bifunctional
 flavokinase/FAD synthetase which produces both FMN (from riboflavin and ATP) and FAD (from FMN and ATP).
 The genes *ribA*, *ribDG*, *ribH* and *ribB* encoding riboflavin biosynthetic enzymes are shown. The gene *ribB* is
 controlled by the FMN riboswitch sroG.

hFSP1 was expressed from a pET-SUMO plasmid upon IPTG induction in the presence of
either riboflavin or roseoflavin (55 μ M).

Induction in the presence of riboflavin yielded soluble FSP1, which was subsequently purified,
and incorporation of FAD was verified by HPLC analysis against a FAD standard (see figure
for the reviewer, below).

**Appendix Fig. 2.** Overproduction and purification of recombinant FSP1 from cell-free extracts of *E. coli* CpXribF
treated with riboflavin (50 μ M). SDS-PAGE analysis of different steps during purification. Protein production was
performed for 24 h at 18°C at 200 rpm in 1 L flasks in a shaking incubator in the presence of riboflavin. Protein
purification was performed using affinity chromatography and a HisTrap HP column connected to an Äkta purifier
system. Lane 1: Cell-free extract after disruption using a French press and ultracentrifugation. Lane 2: Flow through.
Lane 4: Wash fraction. Lane 6: Pellet fraction after cell disruption and ultracentrifugation. Lane M: PageRuler Plus
Prestained Protein Ladder, molecular masses are shown in kDa. Lanes 8-10: Fractions (fractions 13 to 15) following
elution with histidine. Apparently, the FSP1 preparations were over 95% pure. FSP1 is boxed red. Some FSP1 is
present also in the pellet fraction indicating that some FSP1 protein molecules are not folded correctly.

**Appendix Fig. 3.** HPLC analysis of flavins. A Poroshell 120 EC-C18 column (2.7 μ m particle size, 50 mm x 3 mm;
Agilent, Santa Clara, USA) was employed. The following solvent system was used at a flow rate of 5 ml/min: 18%
(vol/vol) methanol-20 mM formic acid-20 mM ammonium formate (pH 3.7). Detection of riboflavin, FMN and FAD
was carried out photometrically at 445 nm and detection of roseoflavin, RoFMN and RoFAD was carried out
photometrically at 503 nm. The left panel shows a signal which corresponds to FAD released from recombinant
FSP1 isolated from *E. coli* CpXribF. The right panel shows the spectrum of the peak (retention time 12.774 min)
shown in panel A with the typical FAD maxima at approximately 222 nm, 266 nm, 373 nm (UV range) and 445-450
592 nm (visible range). In particular, the peak at approximately 445 nm in visible light is characteristic of the yellow
colour of FAD.

We next repeated the experiment with expression induced in the presence of roseoflavin.

Differing from the riboflavin condition, the protein was only collected in one fraction (fraction
13, see figure for the reviewer, below). When the presence of flavins was analysed by HPLC,

neither FAD nor RoFAD were found, most likely because of low yield. Analysis of the pellet
from the centrifuged cell-free extract revealed that large amounts of apparently water-insoluble
recombinant FSP1 were present. This result suggests that FSP1 does not fold correctly in the
presence of RoFAD, and therefore is not soluble in the employed standard buffer. The pellet
fraction was analysed by HPLC and RoFAD was found, however, we cannot rule out that other
proteins are responsible for the release of the cofactor (the fraction was not pure).

**Appendix Fig. 4.** Overproduction and purification of recombinant FSP1 from cell-free extracts of *E. coli* CpXribF
treated with roseoflavin (20 μ M). SDS-PAGE analysis of different steps during purification. Protein production was
performed for 24 h at 18°C at 200 rpm in 1 L flasks in a shaking incubator in the presence of roseoflavin (20 μ M).
Purification was performed using a HisTrap HP column connected to an Äkta purifier. Lane M: PageRuler Plus
Prestained Protein Ladder, molecular masses are shown in kDa. Lane 1: Cell-free extract after disruption in French
press and ultracentrifugation, Lane 3: Flow through. Lane 5: Wash fraction. Lane, 7: Pellet fraction in which FSP1
apparently is insoluble (boxed red) after cell disruption and ultracentrifugation. Lane M: PageRuler Plus. Lanes 10
and 13: Fractions 1 and 2 following elution of FSP1 (boxed red) with histidine. Finally, the expression was induced
in the absence of additional riboflavin or roseoflavin. FSP1 was found again in the insoluble fraction, suggesting a
problem with the correct folding in the absence of cofactor.

Altogether, these data suggest that RoFAD either fails to bind FSP1 or its binding interferes
with proper folding in this system in *E. coli*.

**References**

- 1. Hasford, J.J. & Rizzo, C.J. Linear Free Energy Substituent Effect on Flavin Redox
Chemistry. *Journal of the American Chemical Society* **120**, 2251-2255 (1998).
- 2. Langer, S., Hashimoto, M., Hobl, B., Mathes, T. & Mack, M. Flavoproteins are potential
targets for the antibiotic roseoflavin in Escherichia coli. *J Bacteriol* **195**, 4037-45 (2013).
- 3. Ooshio, T. et al. Identifying simultaneous inhibition of riboflavin metabolism and MEK
as a novel therapeutic strategy for pancreatic cancer. *bioRxiv*, 2024.10.23.619807
(2024).
- 4. Bjelosevic, S. et al. Riboflavin drives nucleotide biosynthesis and iron-sulfur
metabolism to promote acute myeloid leukemia. *bioRxiv* (2025).
- 5. Yang, W.S. et al. Regulation of ferroptotic cancer cell death by GPX4. *Cell* **156**, 317-
331 (2014).
- 6. Zheng, J. et al. N-acetyl-l-cysteine averts ferroptosis by fostering glutathione
peroxidase 4. *Cell Chem Biol* **32**, 767-775 e5 (2025).
- 7. Nakamura, T. et al. Phase separation of FSP1 promotes ferroptosis. *Nature* **619**, 371-
377 (2023).
- 8. Deol, K.K. et al. Vitamin B2 metabolism promotes FSP1 stability to prevent ferroptosis.
*bioRxiv* (2025).
- 9. Martínez-Limón, A., Calloni, G., Ernst, R. & Vabulas, R.M. Flavin dependency
undermines proteome stability, lipid metabolism and cellular proliferation during vitamin
B2 deficiency. *Cell Death Dis* **11**, 725 (2020).
- 10. Mathes, T., Vogl, C., Stolz, J. & Hegemann, P. In vivo generation of flavoproteins with
modified cofactors. *J Mol Biol* **385**, 1511-8 (2009).
- 11. Kitagawa, M. et al. Complete set of ORF clones of Escherichia coli ASKA library (a
complete set of E. coli K-12 ORF archive): unique resources for biological research.
*DNA Res* **12**, 291-9 (2005).
- 12. Plumbridge, J. Control of the expression of the manXYZ operon in Escherichia coli: Mlc
is a negative regulator of the mannose PTS. *Molecular Microbiology* **27**, 369-380
(1998).

Reviewer #1:

Remarks to the Author:

The authors have addressed comments from this reviewer satisfactorily and with carefully designed experiments and reasoning. Congratulations on the excellent work! Recommend to accept.

Reviewer #2:

Remarks to the Author:

I am satisfied with the responses. And the revision of the manuscript improves what was already an important contribution.

Reviewer #3:

Remarks to the Author:

I have now conducted the revision of the amended manuscript Riboflavin metabolism shapes FSP1-driven ferroptosis resistance by Skafar et al.

I feel like this was a very positive revision and the study is now stronger and more conclusive. The rebuttal figures 13-15 answer my most pressing points and consequently my most pressing questions have been answered. Looking at my peers' comments, their questions appear to also have been mostly answered. I do not see the need for further revisions. I hope that my suggestions have been seen as helpful to make the study more complete.

R- Given that no additional aspect requires clarification we take the opportunity for thanking the reviewers for their time and suggestions. Thank you very much!